# Landing with the Score:
# Riemannian Optimization through Denoising

**Andrey Kharitenko**
ETH Zurich,
Institute for Machine Learning
akharitenko@ethz.ch

**Zebang Shen**
ETH Zurich,
Institute for Machine Learning
zebang.shen@inf.ethz.ch

**Riccardo De Santi**
ETH Zurich,
ETH AI Center
rdesanti@ethz.ch

**Niao He**
ETH Zurich,
Institute for Machine Learning
niao.he@inf.ethz.ch

**Florian Dörler**
ETH Zurich,
Automatic Control Laboratory
doerfler@control.ee.ethz.ch

## Abstract

Under the *data manifold hypothesis*, high-dimensional data concentrate near a low-dimensional manifold. We study Riemannian optimization when this manifold is only given implicitly through the data distribution, and standard geometric operations are unavailable. This formulation captures a broad class of data-driven design problems that are central to modern generative AI. Our key idea is a *link function* that ties the data distribution to the geometric quantities needed for optimization: its gradient and Hessian recover the projection onto the manifold and its tangent space in the small-noise regime. This construction is directly connected to the score function in diffusion models, allowing us to leverage well-studied parameterizations, efficient training procedures, and even pretrained score networks from the diffusion model literature to perform optimization. On top of this foundation, we develop two efficient inference-time algorithms for optimization over data manifolds: *Denoising Landing Flow* (DLF) and *Denoising Riemannian Gradient Descent* (DRGD). We provide theoretical guarantees for approximate feasibility (manifold adherence) and optimality (small Riemannian gradient norm). We demonstrate the effectiveness of our approach on finite-horizon reference tracking tasks in data-driven control, illustrating their potential for practical generative and design applications.

## 1 Introduction

Riemannian optimization Boumal (2023); Absil et al. (2008); Hu et al. (2020) considers minimizing an objective function $f : \mathbb{R}^d \to \mathbb{R}$ over an *explicitly known* embedded submanifold $\mathcal{M} \subseteq \mathbb{R}^d$,

$$\min_{x \in \mathcal{M}} f(x). \tag{1}$$

Problem (1) is ubiquitous in fields of machine learning and control and encompasses problems such as independent component analysis Nishimori (1999), low-rank matrix completion Vandereycken (2013), training of orthogonally normalized neural networks Bansal et al. (2018), the control of rigid bodies Duong et al. (2024), as well as sensor network localization Patwari & Hero (2004) and many others. Compared to general constrained optimization, Riemannian optimization promises the advantage of exploiting the natural geometry of the problem, producing feasible iterates and increased numerical robustness Boumal (2023), thus allowing for early termination and making it more suitable for real-time implementation.

In contrast to the above classical setup, in this work we focus on the setting where the manifold $\mathcal{M}$ is given *implicitly* through a *finite set of samples* from an underlying *population data distribution* $\mu_{\mathrm{data}}$ supported on $\mathcal{M}$. This perspective is especially relevant in view of the data *manifold hypothesis* (Loaiza-Ganem et al., 2024), which posits that many real-world data sets lie (approximately)

on a manifold with dimension much smaller than that of the ambient space Fefferman et al. (2016). Importantly, such data manifolds are not just low-dimensional geometric structures – they also capture rich *semantic meaning*. For instance, the image manifold corresponds to photo-realistic images Pope et al., the system behavior manifold to dynamically feasible input-output trajectories Willems & Polderman (1997), while the manifold of airfoils represents aerodynamically viable shapes Zheng et al. (2025). The optimization problem (1) in this implicit setting thus encompasses a broad class of modern tasks, including airfoil- and ship hull design (Chen et al., 2025; Bagazinski & Ahmed, 2023), additive manufacturing Peng et al. (2023), reinforcement learning (Lee & Choi, 2025), and Bayesian inverse problems (Chung et al., 2022b). In this data-driven regime, methods from classical smooth Riemannian optimization *cannot* be applied directly, since they rely on explicit manifold operations such as tangent-space projection, retraction, or exponential maps (Boumal et al., 2019; Boumal, 2023). Graph-based Riemannian optimization has recently been proposed in (Wang et al., 2025), but the setup is quite different, as to be elaborated in Section 1.2. Furthermore, while there has been extensive research on manifold learning Meilă & Zhang (2024); Lin & Zha (2008); Cayton et al. (2005); Belkin & Niyogi (2005) in the past, none of these works addressed (1) from an optimization point of view and focused instead on learning the manifold geometry, rather than representation models that are suitable to be included as a constraint in an optimization problem.

Instead, in this work, we propose a novel data-driven approach to recover the fundamental operations needed for optimization on manifolds: Starting first from the population data distribution $\mu_{\text{data}}$, we smooth it with a Gaussian kernel to obtain

$$p_\sigma = \mathcal{N}(0, \sigma^2 I) * \mu_{\text{data}}, \tag{2}$$

and define the associated *link function*

$$\ell_\sigma(x) = \frac{1}{2}\|x\|^2 + \sigma^2 \log p_\sigma(x), \quad x \in \mathbb{R}^d. \tag{3}$$

We show that, as the smoothing parameter $\sigma$ decreases, the gradient $\nabla \ell_\sigma$ recovers the projection back to the manifold, while the Hessian $\nabla^2 \ell_\sigma$ recovers the projection onto its tangent space. These results reveal that core ingredients of Riemannian optimization – such as retraction and gradient computation – can be implemented directly from the derivative information of $\ell_\sigma$, which itself can be constructed from $\mu_{\text{data}}$.

Given the above novel theoretical findings, a key practical challenge is how to access (even approximately) the gradient $\nabla \ell_\sigma$ and Hessian $\nabla^2 \ell_\sigma$, when, in real-world applications, only samples from $\mu_{\text{data}}$ are available. A crucial observation is that $p_\sigma$ coincides with the marginal distribution of the Variance-Exploding SDE (VE-SDE)[1]. Note that the gradient of $\log p_\sigma$, i.e. the *score function*, is central to diffusion models. It is parameterized by a neural network and learned directly from samples of $\mu_{\text{data}}$, typically via *denoising score matching* (Vincent, 2011). We can hence exploit this well-developed toolkit for a practical implementation of our idea: Let $\mathrm{s}(x, \sigma)$ be a pretrained score network. We use $\mathrm{v}(x) = x + \sigma^2 \mathrm{s}(x, \sigma)$ to approximately represent $\nabla \ell_\sigma$, while the Hessian $\nabla^2 \ell_\sigma$ can be recovered by computing its Jacobian. This approach offers two key advantages: (i) strong inductive biases can be incorporated through the neural network parameterization, and (ii) efficient, well-established training techniques from the diffusion model literature can be directly leveraged. Taken together, the theoretical link between $\nabla \ell_\sigma$ and manifold geometry, and the practical machinery for learning the score, form the foundation of the paradigm shift: from classical smooth Riemannian optimization with explicit manifold knowledge to a data-driven framework where geometry is recovered from samples—thus enabling principled manifold optimization in generative and design-driven applications.

Building on these insights, we propose *the first score-based framework for optimization over data manifolds*. We derive two algorithms: denoising landing flow (DLF) and denoising Riemannian gradient descent (DRGD). Both rely on an approximate score network $\mathrm{s}(x, \sigma) \approx \nabla \log p_\sigma$ learned from samples of $\mu_{\text{data}}$ using standard diffusion-model training. DRGD uses the learned gradient and its Jacobian (for a small, fixed $\sigma$) to mimic Riemannian gradient descent with retraction, while DLF performs gradient flow on a penalized objective, relaxing feasibility at intermediate iterates. We establish non-asymptotic convergence to approximate stationary points as $\sigma \to 0$. Importantly, our methods require only inference of the neural network and gradients with respect to its inputs—not

---

[1]Analogous extensions to VP-SDE and DDPM can also be similarly derived.

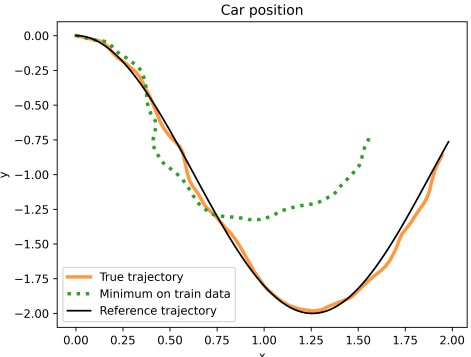

Figure 1: Optimized trajectory (orange) on the system trajectory manifold for the unicycle car model that is desired to track the reference trajectory (black, thin). The closest tracking trajectory in the set of available manifold samples is given in (green, dotted). See Section 6 for more details.

with respect to the network parameters. Thus, if a pretrained score network is already available for a given task, no additional training is required to enable Riemannian optimization on the corresponding data manifold. Viewed from this perspective, our approach can also be interpreted as an *inference-time algorithm*, aligning with a growing trend in modern machine learning research.

## 1.1 OUR CONTRIBUTIONS

We next summarize our main contributions and outline the structure of the paper.

- **Link function and data-driven manifold operations.** Building on the smoothed data distribution $p_\sigma$ and the associated link function $\ell_\sigma$ introduced above, we show in Section 3 that its gradient and Hessian recover, in the small-$\sigma$ regime, the projection onto the manifold and its tangent space. Further, with the help of score learning in diffusion model training, we bridge the gap between the requirements of classical Riemannian optimization (which assumes explicit manifold knowledge) and the emerging need to optimize over data manifolds that are only implicitly available.

- **First score-based algorithms for optimization over data manifolds.** We propose two algorithms – denoising landing flow (DLF) in Section 4 and denoising Riemannian gradient descent (DRGD) in Section 5 – to the best of our knowledge, *the first in the literature* that exploit operations enabled by a pretrained score function. Our methods require only inexpensive inference-time queries and back-propagation with respect to the input of the neural network, making them computationally efficient and readily applicable when pretrained scores are available.

- **Non-asymptotic guaranties.** We establish non-asymptotic convergence guarantees for both algorithms, showing approximate feasibility (outputs close to the data manifold) and approximate optimality (small Riemannian gradient norm) as $\sigma \to 0$ (Theorems 3 and 5). A key technical ingredient is a uniform control of how $\nabla \ell_\sigma$ and $\nabla^2 \ell_\sigma$ approximate the ideal manifold operations at finite $\sigma$ (Theorem 1).

Finally, in Section 6 we validate our approach on classical Riemannian optimization benchmarks and a data-driven optimal control problem for reference trajectory tracking. We demonstrate that our methods can generate feasible points on the manifold with objective values significantly lower than those observed in the training data (Figure 1), illustrating how strong inductive biases of modern deep networks can be harnessed for constrained optimization.

## 1.2 RELATED WORK

In this section we briefly give an overview over the related work. More connections of our approach to previous work can be found in Appendix A.

**Smooth Riemannian Optimization.**  Riemannian optimization (RO) was originally developed under the assumption that the constraint manifold is explicitly known, either through closed-form descriptions such as matrix manifolds or via nonlinear equality constraints (Sato, 2021; Boumal, 2023). A prototypical algorithm in this setting is *Riemannian gradient descent*, which updates by taking a gradient step along the tangent space and then retracting back to the manifold (Boumal et al., 2019). Such algorithms guarantee that the iterates remain on the manifold at every iteration and are thus referred to as *feasible methods*. In contrast, *infeasible methods*, where the iterates are not constrained to stay on $\mathcal{M}$, have also been studied – for example, augmented Lagrangian approaches (Xie & Wright, 2021). However, these typically involve solving complicated optimization subproblems in each inner loop, making them computationally expensive in practice. More recently, a new class of *landing-type algorithms* has emerged: instead of enforcing feasibility at every step, they regularize the objective function with the distance to the manifold and then perform gradient flow or descent on this regularized objective. Such methods have demonstrated strong empirical performance, offering a promising alternative to classical approaches (Ablin & Peyré, 2022; Schechtman et al., 2023).

**Optimization over graph**  Recently, Wang et al. (2025) propose a graph-based strategy for manifold denoising, i.e. given observations sampled from the data manifold, for a new query point, find the closest point on the data manifold. The mixed-order optimization algorithm in this work consists of two types of updates: (1) By extending the concept of tangent space in classical smooth manifolds to the graph setting, first-order steps can be taken over the tangent bundle graph; (2) By enumerating the points connected by the "zero-order edge", zero-th order steps can be taken to avoid local minima and achieve a strong result of convergence to global minima. We highlight that their setup is quite different from ours: (i) Their method is purely non-parametric and hence cannot exploit the recent development of deep learning; (ii) The proposed mix-order algorithm is more like a discrete search method. As an example, their strong global convergence result is built on the existence of zero-order edges, which "teleport" the iterate out of the local minima automatically. This is *not* the typical behavior of continuous optimization method, either deterministic or stochastic; (iii) Their method is purely designed for the denoising task while our focus is for minimizing general smooth objectives over data manifold. We highlight that due to the requirement of their analysis, for different objectives, the zero-order edges need to be reconstructed to avoid local minima.

**Manifold Learning.**  High-dimensional data in modern machine learning often exhibit an intrinsic lower-dimensional structure. Such structure is of central importance: it enables more efficient representation and compression of data, facilitates interpretability by revealing meaningful semantic organization, and provides a foundation for designing algorithms that exploit geometry rather than ambient dimensionality. The task of uncovering this structure is commonly referred to as *manifold estimation* or *manifold learning*, with a large body of work devoted to this goal.

- **Non-parametric** approaches include Isomap (Tenenbaum et al., 2000), Laplacian Eigenmaps (Belkin & Niyogi, 2003), and Locally Linear Embedding (LLE) (Roweis & Saul, 2000), Diffusion Map (Coifman & Lafon, 2006), among many subsequent developments, e.g. (Zhou et al., 2020; Yao et al., 2023).

- **Parametric manifold learning** techniques such as generative adversarial networks (GAN) and various types of autoencoders (AE) are used to estimate data manifolds $\mathcal{M}$ by learning a map $\psi : \mathcal{Z} \to \mathcal{M}$ that parametrizes $\mathcal{M}$ through some lower-dimensional "latent space" $\mathcal{Z}$ and which can be interpreted as a learned coordinate chart on the former Goodfellow et al. (2014); Berahmand et al. (2024). In the case of GAN this map corresponds to the generator, while for AE it corresponds to the decoder. For both GAN and AE, it has been noted that $\psi$ is not always a local chart in the differential-geometric sense Lee et al. (2022) and an additional line of work considered enforcing the latter property for the decoder through regularization or architectural constraints Sorrenson et al. (2023); Kumar et al. (2020). Manifold learning flows (M-flows) Brehmer & Cranmer (2020) address this problem by parametrizing $\psi$ in such a way that $\psi$ is injective and $\psi \circ \psi^{-1}$ can be efficiently computed and have been used for manifold learning and density estimation. Having obtained a chart $\psi$, one can locally trivialize the problem equation 1 to $\min_{\mathcal{Z}} f \circ \psi$, variants of which have also been known under the name of latent space optimization Tripp et al. (2020); Notin et al. (2021). We provide a detailed comparison of this paragraph to our work in Appendix A.

**Comparison with Pull-back Gradient Flow.** When the map $\psi$ is learned, it induces a pull-back metric on the latent space $\mathcal{Z}$. Smooth Riemannian optimization algorithms on $\mathcal{M}$ can then be equivalently formulated in $\mathcal{Z}$; for instance, Riemannian gradient flow on $\mathcal{M}$ corresponds to a pull-back gradient flow on $\mathcal{Z}$. While we are not aware of prior work that explicitly exploits this idea to address optimization over data manifolds, we provide a comparison with this approach in Appendix A.2 and highlight several nontrivial challenges that should not be overlooked.

**Diffusion models.** Diffusion models have achieved remarkable success in generative modeling, where the goal is to generate new samples consistent with an underlying data distribution $\mu_{\mathrm{data}}$. A central ingredient of these methods is the learning of the *score function* – the gradient of the log-density of the diffused distribution $p_\sigma$ Tang & Zhao (2025); Song et al. (2020). In practice, the score is parameterized by a neural network whose architectural design has been extensively studied, and its training is carried out using well-established techniques such as denoising score matching (Song et al., 2020). This combination of principled theory and mature practice has made diffusion models one of the most effective tools for data-driven generative modeling.

When the data distribution resides on a manifold, a recent observation in the diffusion model community is that the score is asymptotically orthogonal to the manifold surface Stanczuk et al. (2024). This observation has been exploited for the estimation of the manifold dimension. See (Kamkari et al., 2024) for the same task. Furthermore, Ventura et al. (2024) has shown that in the case of linear manifolds (i.e. affine subspaces), the Jacobian of the score scaled by the diffusion temperature asymptotically approximates the projection of the manifold onto the normal space and used it to study the geometric phases of diffusion models.

Recent work on the statistical complexity of diffusion models under the manifold hypothesis provides indirect evidence that these models capture geometric information about the underlying data manifold (Oko et al., 2023; Tang & Yang, 2024). In particular, the sample complexity required to learn the data distribution $\mu_{\mathrm{data}}$ depends only on the intrinsic dimension of the manifold, rather than on the ambient dimension.

**Two Formulations: Optimization and Posterior Sampling.** To avoid possible confusion, we stress the distinction between our optimization formulation and the posterior sampling literature. In short, our optimization formulation in eq. (1) enforces the manifold constraint directly. This ensures (approximate) feasibility at the final step and guarantees that the optimization process remains semantically meaningful, which is not guaranteed by the sampling formulation, as discussed below.

Across the literatures Classifier(-Free) Guidance in Diffusion Models (Dhariwal & Nichol, 2021; Ho & Salimans, 2022) and the Plug-and-Play Framework in Bayesian Inverse Problems (Venkatakrishnan et al., 2013; Laumont et al., 2022; Pesme et al., 2025; Graikos et al., 2022; Chung et al., 2022a), a unifying perspective is the task of sampling from a posterior distribution of the form

$$p_{\mathrm{post}} \propto p_{\mathrm{pre}} \exp\left(-\tfrac{r}{\alpha}\right),$$

where $r$ denotes a cost function to be minimized, corresponding to our objective $f$, and $\alpha > 0$ is a temperature-like parameter (Domingo-Enrich et al., 2024). In Classifier(-Free) Guidance, $r$ encodes a classifier signal (e.g., specified by a prompt) and in Bayesian inverse problems, $r$ is the negative log-likelihood of observations. Meanwhile, $p_{\mathrm{pre}}$ serves as a prior distribution, typically derived from a large-scale pretrained generative model, which is expected to capture the semantic structure of the data manifold. Sampling from $p_{\mathrm{post}}$ thus aims to balance semantic plausibility with low cost.

If $p_{\mathrm{pre}}$ were *exactly* supported on the data manifold $\mathcal{M}$ and $\alpha \to 0$, this formulation would reduce to the constrained optimization problem eq. (1). In practice, however, the situation is very different: the distribution induced by a pretrained diffusion model is not concentrated on a low-dimensional manifold but rather has support of non-zero Lebesgue measure in the ambient space—indeed, in many cases, essentially the full space (due to the noisy generation process of $p_{\mathrm{pre}}$). As a result, when $\alpha$ is set too small, the posterior $p_{\mathrm{post}}$ becomes dominated by the exponential tilt $\exp(-r/\alpha)$, pushing samples into regions far from the true data manifold $\mathcal{M}$ and thereby *losing* semantic meaning. Consequently, these sampling-based frameworks must carefully tune $\alpha$ to trade off semantic fidelity (staying close to $\mathcal{M}$) against optimization quality (achieving low $r$).

## 2 PRELIMINARIES

Here we introduce some preliminary notation and concepts to state our results. We refer the reader to the Appendices B and D for more details.

### 2.1 MANIFOLDS AND DISTANCE FUNCTIONS

Let $\mathcal{M} \subseteq \mathbb{R}^d$ be a $k$-dimensional embedded compact $C^2$-submanifold without boundary. For any point $p \in \mathcal{M}$ we denote by $T_p \mathcal{M} \subseteq \mathbb{R}^d$ and $N_p \mathcal{M} \subseteq \mathbb{R}^d$ the tangent and normal spaces of $\mathcal{M}$ at $p$, respectively, and their orthogonal projections by $P_{T_p \mathcal{M}}$ and $P_{N_p \mathcal{M}}$. For a $C^2$-submanifold, then there exists a radius $\tau_{\mathcal{M}} > 0$ such that every point in $x \in \mathcal{T}(\tau_{\mathcal{M}})$, where

$$\mathcal{T}(\tau) := \{x \in \mathbb{R}^d \mid \mathrm{dist}(x, \mathcal{M}) < \tau\}, \quad \mathrm{dist}(x, \mathcal{M}) = \inf_{p \in \mathcal{M}} \|x - p\|,$$

has a unique projection $\pi(x) \in \mathcal{M}$. Sets of the form $\mathcal{T}(\tau)$ with $\tau \in (0, \tau_{\mathcal{M}})$ are called tubular neighborhoods of $\mathcal{M}$. The squared distance function is defined by $\mathrm{d}(x) = \frac{1}{2} \mathrm{dist}(x, \mathcal{M})^2$ and $\mathrm{d}$ and $\pi$ are both differentiable on $\mathcal{T}(\tau_{\mathcal{M}})$ with $\frac{1}{2} \|x - \pi(x)\|^2 = \mathrm{d}(x)$ and $x - \pi(x) \in N_{\pi(x)} \mathcal{M}$ and $\mathrm{d}'(x) = x - \pi(x)$ and $\pi'(x)$ given explicitly in Appendix B. Finally, for a function $f : \mathcal{M} \to \mathbb{R}$ we denote the Riemannian gradient by $\mathrm{grad}_{\mathcal{M}} f(p)$, which in the case of $f : \mathbb{R}^d \to \mathbb{R}$ is given by $\mathrm{grad}_{\mathcal{M}} f(p) = P_{T_p \mathcal{M}} \nabla f(p)$.

### 2.2 THE STEIN SCORE FUNCTION AND SCORE-BASED DIFFUSION MODELS

For a Borel probability measure we denote its Gaussian blurring by $p_\sigma = \mathcal{N}(0, \sigma^2 I) * \mu$ for $\sigma > 0$ with $p_0 = \mu$. The score function $\nabla \log p_\sigma$ of $p_\sigma$ and its Jacobian allow for the following interpretation (see Jaffer & Gupta (1972), also Appendix C):

$$x + \sigma^2 \nabla \log p_\sigma(x) = \nabla \ell_\sigma(x) = \mathbb{E}\nu_{x,\sigma}, \quad I + \sigma^2 \nabla^2 \log p_\sigma(x) = \nabla^2 \ell_\sigma(x) = \frac{1}{\sigma^2} \mathrm{Cov}(\nu_{x,\sigma}), \quad (4)$$

where $\ell_\sigma$ is the link function (3) and $\nu_{x,\sigma}$ is the posterior distribution observing $x$ under the noise model $p_\sigma$ and prior $\mu$. The representation for $\mathbb{E}\nu_{x,\sigma}$ has also been known under the name of Tweedie's formula Robbins (1992); Efron (2011). The score function has gained recent attention due to its use in score-based diffusion models in the field of generative modelling. Specifically in the so-called *variance exploding* (VE) diffusion scheme one seeks to learn $\nabla \log p_\sigma$ for different noise scales $\sigma$ via a neural network $s(\cdot, \sigma)$ by minimizing the conditional score matching loss $L_{\mathrm{CSM}}(s(\cdot, \sigma))$, which attains its unique minimum in $s(\cdot, \sigma) = \nabla \log p_\sigma$. Then $s(\cdot, \sigma)$ is used for sampling from $\mu$ by following a particular reverse-time SDE or ODE flow in the noise scale $\sigma$ (see Appendix D for more details).

## 3 SCORE AS RETRACTION AND TANGENT SPACE PROJECTION

In this section we give another interpretation to the score function $\nabla \log p_\sigma(x)$ and its Jacobian $\nabla^2 \log p_\sigma(x)$. Namely, it has been already observed in Stanczuk et al. (2024) that the score function is for $\sigma \to 0$ asymptotically orthogonal to the tangent space of the data manifold. Our first contribution is showing that when the support of the distribution $\mu$ is a manifold $\mathcal{M}$ and $\mu$ is absolutely continuous w.r.t. its volume measure, then both quantities (4) approximate the projection operator $\pi(x)$ and its Jacobian $\pi'(x)$ *uniformly* on tubular neighborhoods of $\mathcal{M}$. Formally we establish the following

**Theorem 1** (Main). *Let $\mathcal{M} \subseteq \mathbb{R}^d$ be a compact, embedded $C^3$-submanifold and $\mu \in \mathcal{P}(\mathbb{R}^d)$ a Borel probability measure with $\mathrm{supp}\, \mu = \mathcal{M}$ and $\mu \ll \mathrm{Vol}_{\mathcal{M}}$ such that $\frac{\mathrm{d}\mu}{\mathrm{d}\mathrm{Vol}_{\mathcal{M}}} \in C^3(\mathcal{M})$. Then for any $\tau \in (0, \tau_{\mathcal{M}})$ there exist some constants $K = K(\tau, \mathcal{M}, \mu) > 0$ and $\overline{\sigma} = \overline{\sigma}(\tau, \mathcal{M}, \mu) > 0$ depending on $\tau$, $\mathcal{M}$ and $\mu$ such that*

$$\|\mathbb{E}\nu_{x,\sigma} - \pi(x)\| \leq K\sigma |\log(\sigma)|^3 \quad and \quad \left\| \frac{1}{\sigma^2} \mathrm{Cov}(\nu_{x,\sigma}) - \pi'(x) \right\| \leq K\sigma |\log(\sigma)|^3 \qquad (5)$$

*for all $\sigma \in (0, \overline{\sigma})$ and $x \in \mathcal{T}(\tau)$.*

The proof is deferred to Appendix E and is based on a careful non-asymptotic estimate of the Laplace integral method. As a consequence and together with fact that $\pi'(x)$ coincides with $\mathrm{P}_{\mathrm{T}_x \mathcal{M}}$ for $x \in \mathcal{M}$ (see Appendix B) we obtain the following result.

**Corollary 2.** *Let $\mathcal{M}$ and $\mu$ be as in Theorem 1 and suppose that $x \in \mathcal{M}$. Then*
$$\lim_{\sigma \to 0} I + \sigma^2 \nabla^2 \log p_\sigma(x) = \mathrm{P}_{\mathrm{T}_x \mathcal{M}}\,.$$

In view of Theorem 1 let us abbreviate (4) into
$$\mathrm{d}_\sigma(x) = -\sigma^2 \log p_\sigma(x)\,, \quad \pi_\sigma(x) = x + \sigma^2 \nabla \log p_\sigma(x)\,, \quad P_\sigma(x) = I + \sigma^2 \nabla^2 \log p_\sigma(x)\,, \quad (6)$$
with the limiting cases $\mathrm{d}_0 = \mathrm{d}$, $\pi_0 = \pi$ and $P_0(x) = \pi'(x)$. We stress that the expressions in (6) are defined for all $x \in \mathbb{R}^d$, whereas $\mathrm{d}$, $\pi$ and $\pi'$ in are only sensible in a tubular neighborhood $\mathcal{T}$ of $\mathcal{M}$. Thus, in theory, a well-trained diffusion model score $\mathrm{s}(\cdot, \sigma)$ and its Jacobian $\mathrm{s}'(\cdot, \sigma)$ allow us to approximate the closest-point projection $\pi(x)$ and the tangent space projection $\mathrm{P}_{\mathrm{T}_x \mathcal{M}}$ as $\sigma \to 0$ arbitrarily well via the operator $\mathrm{v}(x) = x + \sigma^2 \mathrm{s}(x, \sigma)$ and its Jacobian, respectively.

## 4 DENOISING RIEMANNIAN GRADIENT FLOW WITH LANDING

In this section we show how to use the score function from Section 3 for Riemannian optimization of (1) for some smooth $f \in C^1(\mathbb{R}^d)$. Assuming that we have access to a sufficiently accurate estimate of the score in form of a vector function $\mathrm{v} \in C^1(\mathbb{R}^d; \mathbb{R}^d)$ such that $\mathrm{v}(x) = x + \sigma^2 \mathrm{s}(x, \sigma)$ and
$$\|\mathrm{v}(x) - \pi_\sigma(x)\| \le \epsilon \text{ and } \|\mathrm{v}'(x) - P_\sigma(x)\| \le \epsilon \text{ for } x \in \mathcal{T}(\tau) \quad (7)$$
for some $\tau \in (0, \tau_\mathcal{M})$, we propose for $\eta \ge 0$ the *denoising landing flow* (DLF)
$$\dot{x} = -\mathrm{v}'(x)\nabla f(\mathrm{v}(x)) + \eta(\mathrm{v}(x) - x)\,. \quad (8)$$
In the exact case $\mathrm{v}(x) = \pi_\sigma(x)$ and $\mathrm{v}'(x) = P_\sigma(x)$ (i.e. $\epsilon = 0$ in (7)), flow (8) is the gradient flow
$$\dot{x} = -\nabla F_\sigma^\eta(x) = -P_\sigma(x)\nabla f(\pi_\sigma(x)) + \eta(\pi_\sigma(x) - x) \text{ with } F_\sigma^\eta(x) = f(\pi_\sigma(x)) + \eta \, \mathrm{d}_\sigma(x) \quad (9)$$
and the dynamics in (9) consists of two parts: An approximate projection $P_\sigma(x)\nabla f(\pi_\sigma(x))$ of the gradient $\nabla f(\pi_\sigma(x))$ and an approximate landing term $\eta(\pi_\sigma(x) - x)$ corresponding to the penalty function $\eta \, \mathrm{d}_\sigma(x)$. In the further case of $\sigma = 0$ and $x(0) \in \mathcal{M}$ the flow (9) reduces to the ordinary Riemannian gradient flow, which has been extensively studied Helmke & Moore (2012); Ambrosio et al. (2005). Interestingly, when $\sigma = 0$, but only $x(0) \in \mathcal{T}(\tau)$, then (9) reduces to
$$\dot{x} = -H_x^{-1} \operatorname{grad}_\mathcal{M} f(\pi(x)) + \eta(\pi(x) - x)\,, \quad (10)$$
with $H_x^{-1}$ a linear operator on $\mathrm{T}_{\pi(x)} \mathcal{M}$ given in Appendix B. In particular the two terms in (10) belong to $\mathrm{T}_{\pi(x)} \mathcal{M}$ and $\mathrm{N}_{\pi(x)} \mathcal{M}$, respectively, and are orthogonal to each other, implying that the distance between $x$ and $\mathcal{M}$ is non-increasing, which allows for perfect landing on the manifold via similar arguments as in Ablin & Peyré (2022); Schechtman et al. (2023), see Theorem 22 in Appendix F.3. For $\sigma > 0$ or $\epsilon > 0$, the two summands in (9) and (8) in general not perpendicular to each other and the landing is not exact. However, using Theorem 1 we show the following.

**Theorem 3.** *Consider the flow (8) and $\tau \in (0, \tau_\mathcal{M})$. Set $C = \|\nabla f|_{\mathcal{T}(\tau)}\|_\infty$ and $L = \operatorname{Lip}(\nabla f)$. Suppose that for some $\epsilon > 0$ and $\sigma \in (0, \overline{\sigma}(\tau, \mathcal{M}, \mu))$ with $\epsilon + K(\tau, \mathcal{M}, \mu)\sigma|\log(\sigma)|^3 \le \min\{\tau, \frac{2\tau}{1+C/\eta}\}$ the function $\mathrm{v}$ satisfies (7). Then for any $x(0) \in \mathcal{T}(\tau)$ the solution $x(t)$ to (8) exists for all $t \ge 0$ and is contained in $\mathcal{T}(\tau)$. Moreover, every accumulation point $x_*$ of this flow satisfies*
$$\operatorname{dist}_\mathcal{M}(x_*) \le \tau_0 := \frac{1}{2}\left(\frac{C}{\eta} + 1\right)(\epsilon + K(\tau, \mathcal{M}, \mu)\sigma|\log(\sigma)|^3)\,,$$
*and for the projection $p_* = \pi(x_*)$ it holds that*
$$\|\operatorname{grad}_\mathcal{M} f(p_*)\| \le \left(2(L + C + 2\eta) + \frac{(1 + C/\eta)/\tau_\mathcal{M}}{1 - \tau/\tau_\mathcal{M}} C\right)(\epsilon + K(\tau, \mathcal{M}, \mu)\sigma|\log(\sigma)|^3)\,. \quad (11)$$

Thus, Theorem 3 shows that one can still use the flow (8) for a fixed $\sigma > 0$ to converge to approximate critical points of the objective $f$, at which the approximation error and norm of the Riemannian gradient are both $\widetilde{O}(\sigma)$ plus the score error $\epsilon$.

**Remark 4.** *We can evaluate the right hand side of the flow (8) in a single forward-backward pass of the network $\mathrm{v}$. Namely, given an input $x$, we compute and store $p = \mathrm{v}(x)$ by a forward pass of $\mathrm{v}$, while keeping the computational graph of $\mathrm{v}(x)$. Then we evaluate $g = \nabla f(p)$ and build the computational graph of $y = \langle \mathrm{v}(x), g \rangle$, while detaching $g$. Finally we backpropagate on $x$ in $y$ to obtain $\mathrm{v}'(x)g = \mathrm{v}'(x)\nabla f(\mathrm{v}(x))$.*

## 5 DENOISING RIEMANNIAN GRADIENT DESCENT

In a practical implementation one has to consider a discretized version of the flow (8). A natural alternative is to study the following approximate version of the Riemannian gradient descent Absil et al. (2008); Boumal (2023)

$$x_{k+1} = \mathrm{v}(x_k - \gamma_k \mathrm{v}'(x_k)\nabla f(x_k)), \tag{12}$$

which we term the *denoising Riemannian gradient descent* (DRGD). Here $\mathrm{v}$ acts as an approximate retraction and $\mathrm{v}'$ as an approximate projection onto the tangent space. We obtain the following convergence result for this algorithm.

**Theorem 5.** *Let $\tau \in (0, \tau_{\mathcal{M}}/2)$ and set $C = \|\nabla f|_{\mathcal{T}(\tau)}\|_\infty$ and $L = \mathrm{Lip}(\nabla f)$, $D = \|f|_{\mathcal{T}(\tau)}\|_\infty$ and*

$$L_0 = 8C\left(2(\frac{3}{\tau_{\mathcal{M}}} + \tau M) + \frac{1}{\tau_{\mathcal{M}}}\right) + 2L.$$

*Suppose that for some $\sigma \in (0, \overline{\sigma}(\tau, \mathcal{M}, \mu))$ and $\epsilon > 0$ with $\epsilon' := \epsilon + K(\tau, \mathcal{M}, \mu)\sigma|\log(\sigma)|^3 \leq \tau/2$ the function $\mathrm{v}$ satisfies (7) and that the step-size $\gamma_k$ is constrained by $\gamma_k \in [\gamma_{\min}, \gamma_{\max}]$ with*

$$0 < \gamma_{\min} < \gamma_{\max} < \min\left\{\frac{2}{L_0}, \frac{\tau}{C(4 + \tau)}\right\}.$$

*Then for any $x_0 \in \mathcal{T}(\tau/2)$ the iterates $x_k$ of (12) satisfy $\{x_k\}_{k=1}^\infty \subseteq \mathcal{T}(\epsilon') \subseteq \mathcal{T}(\tau/2)$ and for the projection $p_k = \pi(x_k)$ the following average-of-gradient-norm condition holds:*

$$\frac{1}{N}\sum_{k=0}^N \|\mathrm{grad}_{\mathcal{M}} f(p_k)\|^2 \leq \frac{4D/N + (8C^2\epsilon'/\tau_{\mathcal{M}}^2 + 2(2C + L_0\gamma_{\max}(C + L)))\epsilon'}{\gamma_{\min}(1 - \frac{L_0}{2}\gamma_{\max})}.$$

*In particular there exists at least one accumulation point $x_* \in \overline{\mathcal{T}}(\epsilon)$ of $\{x_k\}_{k=0}^\infty$ such that its projection $p_* = \pi(x_*)$ satisfies*

$$\|\mathrm{grad}_{\mathcal{M}} f(p_*)\|^2 \leq \frac{(8C^2\epsilon'/\tau_{\mathcal{M}}^2 + 2(2C + L_0\gamma_k(C + L)))\epsilon'}{\gamma_{\min}(1 - \frac{L_0}{2}\gamma_{\max})}.$$

**Remark 6.** *If we set $\epsilon = 0$ and $\sigma = 0$, we recover, up to constants, the classical result on iterates of the Riemannian gradient descent with known manifold $\mathcal{M}$ and non-convex objective $f$ with Lipschitz gradient (Boumal, 2023, Corollary 4.9) (see Appendix A.3).*

**Remark 7.** *Note that both Theorem 3 and Theorem 5 require the rather strong $L^\infty$-approximation assumption (7) on $\mathrm{v}$ and its Jacobian. The analysis under a weaker $L^2$-bound is out of score for this paper and left for future work.*

## 6 NUMERICAL EXPERIMENTS AND APPLICATIONS

In this section we provide some numerical results for our proposed algorithms, namely the denoising landing flow (8) (more precisely, the discretized version (51)) and the denoising Riemannian gradient descent (12).

### 6.1 OPTIMIZATION ON ORTHOGONAL GROUP $O(n)$

In this section we evaluate flow (8) on a synthetic example. In order to compare the error of our method to classical Riemannian optimization techniques, we consider distributions supported on manifolds that are the focus of study in Absil et al. (2008); Boumal (2023). Specifically we consider the orthogonal group manifold $\mathcal{M} = O(n) \subseteq \mathbb{R}^{n \times n}$ with $\mu = \mathrm{Vol}_{\mathcal{M}}$ being the uniform volume measure and Brockett's cost function Helmke & Moore (2012) defined by

$$\min_{X \in O(n)} f(X) := \mathrm{tr}(AXQX^\top),$$

where $A, Q \in \mathbb{S}^{n \times n}$ are given. We consider the cases $n \in \{10, 20\}$ and assume that we are given a set $\mathcal{D}_{\mathrm{train}} \subseteq \mathcal{M}$ of $N_{\mathrm{data}} = 20000$ data points from $\mu$ and train the score function $\mathrm{s}$ with denoising score matching (see Appendix D for diffusion models and Appendix H.1.1 for implementation details). In Figure 2 (left) we compare the evolution of the objective value of our approximation of (8) to the exact landing flow (10) for different noise levels $\sigma > 0$ and dimension $n$. We observe that we can obtain objective values with cost lower than the best possible point in the training set and that the accuracy improves as $\sigma \to 0$.

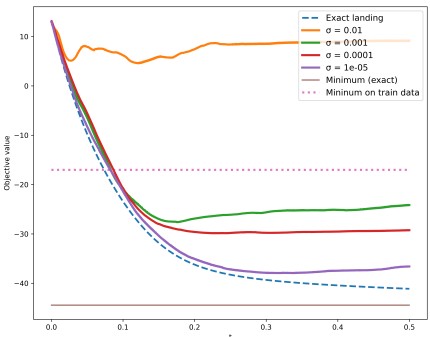 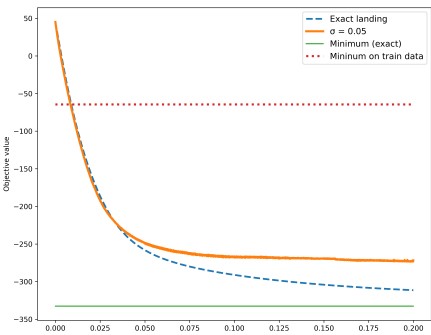

Figure 2: Objective value vs. flow time $t$ for the orthogonal manifold for $n = 10$ (left, different $\sigma > 0$) and for $n = 20$ (right, $\sigma = 0.05$). Here "exact landing" refers to equation 8 with exact operations $\mathrm{v} = \pi$ and $\mathrm{v}' = \pi'$.

## 6.2 Reference tracking via data-driven control

**Problem definition:** In this example we consider applying our method to the control of discrete-time dynamical systems on a finite horizon. Specifically we assume that we are given a discrete-time state-space system

$$x_{k+1} = \boldsymbol{f}(x_k, u_k), \quad y_k = \boldsymbol{g}(x_k, u_k), \quad k = 0, \ldots, N_h - 1, \tag{13}$$

on a finite time horizon $N_h$ with state $x_k \in \mathbb{R}^{n_x}$, input $u_k \in \mathbb{R}^{n_u}$, output $y_k \in \mathbb{R}^{n_y}$ and a fixed initial state $x_0 = 0 \in \mathbb{R}^{n_x}$. The task is to find inputs $\boldsymbol{u} = (u_0, \ldots, u_{N_h-1})$ such that the corresponding outputs $\boldsymbol{y} = (y_0, \ldots, y_{N_h})$ closely track a prespecified reference trajectory $\boldsymbol{r} = (r_0, \ldots, r_{N_h})$ by solving the optimal control problem

$$\min_{(\boldsymbol{u}, \boldsymbol{y}) \in \mathcal{M}_{\mathrm{IO}}} f(\boldsymbol{u}, \boldsymbol{y}) \tag{14}$$

with tracking objective

$$f(\boldsymbol{u}, \boldsymbol{y}) = \sum_{k=0}^{N_h-1} u_k^\top R u_k + (y_k - r_k)^\top Q (y_k - r_k) + (y_{N_h} - r_{N_h})^\top Q (y_{N_h} - r_{N_h}) \tag{15}$$

for positive-definite weight matrices $R \in \mathbb{S}^{n_u \times n_u}$ and $Q \in \mathbb{S}^{n_y \times n_y}$ and feasible input-output set

$$\mathcal{M}_{\mathrm{IO}} = \left\{ (\boldsymbol{u}, \boldsymbol{y}) \in (\mathbb{R}^{n_u})^{N_h} \times (\mathbb{R}^{n_y})^{N_h+1} \;\middle|\; \begin{array}{l} \text{exists } \boldsymbol{x} = (x_0, \ldots, x_{N_h}) \in (\mathbb{R}^{n_x})^{N_h+1} \\ \text{with } (\boldsymbol{u}, \boldsymbol{x}, \boldsymbol{y}) \text{ satisfying (13)} \end{array} \right\}.$$

Under smoothness assumptions on the dynamics $\boldsymbol{f}$ and $\boldsymbol{g}$, the set $\mathcal{M}_{\mathrm{IO}}$, being a graph of a smooth map, is a (non-compact) embedded smooth submanifold of $(\mathbb{R}^{n_u})^{N_h} \times (\mathbb{R}^{n_y})^{N_h+1}$. The problem (14) is ubiquitous in receding horizon control applications such as model predictive control (MPC) and used for e.g. autonomous driving Vu et al. (2021), motion planning Cohen et al. (2020), optimizing HVAC system energy efficiency Serale et al. (2018) and inventory control Kostić (2009). In many of these applications the dynamics (13) governing the system are *not known* explicitly. Instead, in data-driven control Dörfler (2023a;b); Markovsky et al. (2023) one assumes that (13) is given *implicitly* by a finite number of measured input-output trajectories

$$\mathcal{D}_{\mathrm{train}} = \{ (\boldsymbol{u}_i, \boldsymbol{y}_i) \mid i = 1, \ldots, N_{\mathrm{data}} \} \subseteq \mathcal{M}_{\mathrm{IO}},$$

where the input $\boldsymbol{u}$ is persistently exciting Willems et al. (2005), e.g. given by (white) noise Ljung (1999). In particular the so-called *system behavior* manifold $\mathcal{M}_{\mathrm{IO}}$ Willems & Polderman (1997) is given by samples from a distribution $\mu$ on its in- and outputs and fits precisely into our framework of data-driven Riemannian optimization (1). A similar setup with observable state $\boldsymbol{y} = \boldsymbol{x}$ has been considered in the domain of reinforcement learning Janner et al. (2022). We test our proposed denoising Riemannian gradient descent on two classical systems from the

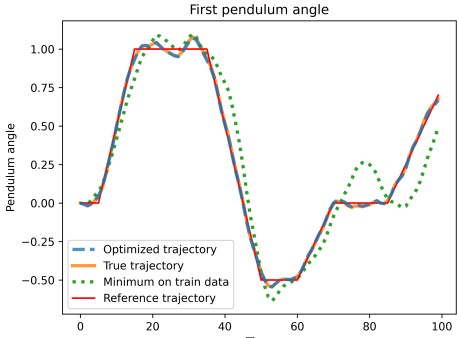 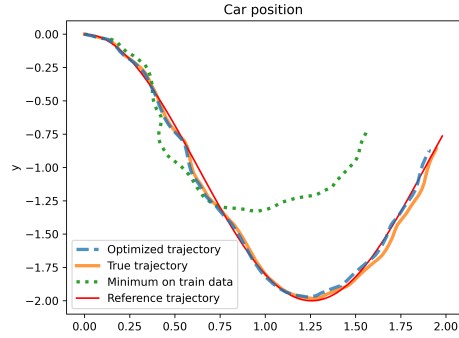

Figure 3: Denoising Riemannian gradient descent: Angle of the first pendulum (left) and unicycle car position (right) with the optimized output trajectory $\boldsymbol{y}^*$ (blue, dashed), the true system trajectory $\boldsymbol{y}^{\text{true}}$ (orange), the initial trajectory $\boldsymbol{y}_0$ (green, dotted) and the reference trajectory $\boldsymbol{r}$ (red)

control domain: The discretized double pendulum system and the unicycle car model LaValle (2006) (see Appendix H.2.1 detailed information on the systems and the particular choice of $\boldsymbol{r}$, $R$ and $Q$ in (15)), each on a horizon of $N_h = 100$. To apply our proposed methods, we train a diffusion model (see Appendix H.2.2 for implementation details) on the measured trajectories $\mathcal{D}_{\text{train}}$ and solve (14) via the denoising Riemannian gradient descent to obtain a solution $(\boldsymbol{u}^*, \boldsymbol{y}^*)$. As initial values we take the trajectories from the training set that minimize the objective cost $f$, i.e. $(\boldsymbol{u}_0, \boldsymbol{y}_0) = \arg\min_{(\boldsymbol{u}, \boldsymbol{y}) \in \mathcal{D}_{\text{train}}} f(\boldsymbol{u}, \boldsymbol{y})$. Note that, as seen in Section 5, in general the final iterate will not exactly lie on the input-output manifold, i.e. $(\boldsymbol{u}^*, \boldsymbol{y}^*) \notin \mathcal{M}_{\text{IO}}$. To account for this deviation we back-test our generated input trajectory $\boldsymbol{u}^*$ by implementing it on the true underlying system (13) to obtain the real output $\boldsymbol{y}^{\text{true}}$.

**Results and discussion:** In Figure 4 (in Appendix H.2.3) we depict the evolution of the objective value w.r.t. iteration count and in Figure 3 we depict the final optimizing trajectories $\boldsymbol{y}^*$ and $\boldsymbol{y}^{\text{true}}$. We can observe that the error $\|\boldsymbol{y}^* - \boldsymbol{y}^{\text{true}}\|$ is small, which shows that $(\boldsymbol{u}^*, \boldsymbol{y}^*)$ is close to the true system behavior $\mathcal{M}_{\text{IO}}$. Moreover, we can see that the trajectory $\boldsymbol{y}^{\text{true}}$ tracks the corresponding reference $\boldsymbol{r}$ much better than the train set minimum $\boldsymbol{y}_0$, which shows a generalization capability of our diffusion model. We from Figure 4 (left) that the current objective can depart from the true objective significantly. This is due to the iterates deviating from $\mathcal{M}_{\text{IO}}$. In this example the algorithm (DRGD) recovers and we have found it to be robust w.r.t. moderate deviations from the manifold. Note that we have set a iteration budget of $N_{\text{iter}} = 3000$ and $N_{\text{iter}} = 2500$, respectively, while the objective is still decreasing. Accelerating the convergence of DRGD is left for future work.

## 7 CONCLUSION AND FUTURE WORK

In this paper, we show that the denoising score and its Jacobian allow to perform manifold operations such as the closest-point and tangent space projection without the explicit knowledge of the manifold. We then propose a landing flow for the corresponding manifold-constrained Riemannian optimization problem and show that its limit points approximate critical points of the original problem. Moreover, we investigate the approximate version of the Riemannian gradient descent and provide a bound on the average-gradient-norm of its iterates, which converges to zero as the manifold operations become more exact. We apply both algorithms on known manifolds and to finite-horizon reference tracking in the domain of data-driven control. Future work will consist of deriving error bounds for this flow when the denoising score is trained with a non-zero $L^2$-error as well as the study of more sophisticated classical Riemannian optimization algorithms such as Newton and trust region methods when using the approximate manifold operations with the trained score to accelerate convergence.

ACKNOWLEDGMENTS AND DISCLOSURE OF FUNDING

We thank the reviewers for their constructive suggestions. The work is supported by ETH research grant, Swiss National Science Foundation (SNSF) Project Funding No. 200021-207343, and SNSF Starting Grant. Andrey Kharitenko is supported as a part of NCCR Automation, a National Centre of Competence in Research, funded by the Swiss National Science Foundation (grant number 51NF40_225155). Riccardo De Santi is supported by the ETH AI Center through an ETH AI Center doctoral fellowship.

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

## A    CONNECTIONS AND COMPARISONS TO PRIOR WORK

### A.1    CONNECTION TO DENOISING AUTOENCODERS

In the following we show how our results connect to learning denoising auto-encoders. The following theorem is shown in Alain & Bengio (2014):

**Theorem 8** (Theorem 1 in Alain & Bengio (2014)). *Let $\mu$ be a probability measure on $\mathbb{R}^d$ and consider for $\sigma > 0$ the reconstruction problem*

$$\min_r L_{\text{DAE}}(r) \quad for \quad L_{\text{DAE}}(r) = \mathbb{E}_{x \sim \mu, \, \epsilon \sim \mathcal{N}(0,I)} \|x - r(x + \sigma\epsilon)\|^2 .$$

*If $\mu$ is absolutely continuous with (Lebesgue) density $p$, the optimal reconstruction function $r_\sigma^*$ satisfies*

$$r_\sigma^*(x) = \frac{\mathbb{E}_{\epsilon \sim \mathcal{N}(0,I)} p(x - \sigma\epsilon)(x - \sigma\epsilon)}{\mathbb{E}_{\epsilon \sim \mathcal{N}(0,I)} p(x - \sigma\epsilon)} \overset{\sigma \to 0}{=} x + \sigma^2 \nabla \log p(x) + o(\sigma^2)$$

*at any $x \in \mathbb{R}^d$ with $p(x) > 0$.*

To see how this result relates to Theorem 1, note that function $r_\sigma^*$ here is precisely our "approximate projection" $\pi_\sigma$, since the first above expression corresponds to the posterior of observing $x$ under the convolution of the prior $p$ with $\mathcal{N}(0, \sigma^2 I)$, i.e. in our notation

$$r_\sigma^*(x) = \mathbb{E}\nu_{x,\sigma} = \pi_\sigma(x).$$

Therefore, if $\mu$ has the Lebesgue-density $p$ (i.e. the data manifold would have full dimension $k = d$), then Alain & Bengio (2014) show that

$$x + \sigma^2 \nabla \log p_\sigma(x) \overset{\sigma \to 0}{=} x + \sigma^2 \nabla \log p(x) + o(\sigma^2).$$

In particular, if $p(x) > 0$, then the last equation states that

$$\lim_{\sigma \to 0} x + \sigma^2 \nabla \log p_\sigma(x) = x.$$

Theorem 1 on the other hand deals with the case $k < d$, i.e. when the manifold is truly lower-dimensional. In this case, $\mu$ has no density $p$ w.r.t. the Lebesgue measure on $\mathbb{R}^d$, and it holds that

$$\lim_{\sigma \to 0} x + \sigma^2 \nabla \log p_\sigma(x) = \pi(x),$$

where $\pi$ denotes the projection operation onto the data manifold $\mathcal{M}$. While Alain & Bengio (2014) does consider manifold support in their discussion (see (Alain & Bengio, 2014, Section 3.4, Figure 5)), note that the population data measure in this case is **not** absolute continuous w.r.t. the Lebesgue measure on $\mathbb{R}^d$, and hence $\log p(x)$ has no meaning. Consequently, results of Alain & Bengio (2014), e.g. the expansion above regarding $r_\sigma^*$, are not applicable. Theorem 1 can therefore be seen a (uniform) generalization of Alain & Bengio (2014) to an order of $\widetilde{O}(\sigma)$, establishing the asymptotic convergence of $r_\sigma^*$ and its Jacobian for $\sigma \to 0$ as

$$r_\sigma^*(x) = \pi(x) + \widetilde{O}(\sigma), \quad (r_\sigma^*)'(x) = \pi'(x) + \widetilde{O}(\sigma).$$

### A.2 TRIVIALIZATIONS AND NATURAL GRADIENTS USING PARAMETRIC MANIFOLD LEARNING

In this section we compare our approach to latent space optimization. For this purpose, we first describe the latter problem in terms of trivializations of (1) using local parametrizations: Given a finite atlas $\mathcal{A} = (\psi_i, \mathcal{U}_i, \mathcal{V}_i)_{i=1}^m$ of $\mathcal{M}$ with open $\mathcal{V}_i \subseteq \mathbb{R}^k$ and $\mathcal{U}_i \subseteq \mathcal{M}$ and charts $\psi_i : \mathcal{U}_i \to \mathcal{V}_i$ it clearly holds

$$\min_{x \in \mathcal{M}} f(x) = \min_{i=1,\dots,m} \min_{z \in \mathcal{V}_i} f(\psi_i(z)).$$

In the following we will focus on one inner subproblem by fixing a single chart $(\psi, \mathcal{U}, \mathcal{V})$ and consider

$$\min_{x \in \mathcal{U}} f(x) = \min_{z \in \mathcal{V}} f(\psi(z)). \tag{16}$$

While for the above equality we only need $\psi$ to be surjective (i.e. $\psi(\mathcal{V}) = \mathcal{U}$), for equivalence of first-order stationary points of both problems we need $\psi$ to be a submersion: To see this, note that stationarity of $x_* \in \mathcal{U}$ for the left hand side in equation 16 is equivalent to $\text{grad}_\mathcal{M} f(x_*) = P_{T_{x_*}\mathcal{M}} \nabla f(x_*) = 0$ and stationarity of any $z_* \in \mathcal{V} \cap \psi^{-1}(\{x_*\})$ for the right

hand side is equivalent to $\psi'(z_*)^\top \nabla f(\psi(z_*)) = \psi'(z_*)^\top \nabla f(x_*) = 0$. In particular, for these two notions to be equivalent a-priori for any $f$ and $x_*$, the columns of $\psi'(z_*)$ must span $\mathrm{T}_{x_*}\mathcal{M}$, which implies that $\psi$ is a submersion. In particular, if $\psi$ is a chart, then the stationary points of both sides in equation 16 are in a one-to-one correspondence and *only then* both problems are equivalent from an first-order optimization point of view.

Two possible ways to solve the right hand side in (16) are the latent space gradient flow (LSGF) and the pull-back Riemannian gradient flow (PBRGF) given by

$$\dot z = -\nabla(f \circ \psi)(z) = -\psi'(z)^\top \nabla f(\psi(z)), \tag{17}$$

$$\dot z = -G_\psi(z)\psi'(z)^\top \nabla f(\psi(z)), \tag{18}$$

respectively, where $G_\psi(z) = \psi'(z)^\top \psi'(z)$ is the metric tensor of $\psi$ at $z \in V$. Here PBRGF[2] is obtained from LSGF via a change in inner product on the tangent space $\mathrm{T}_{\psi(z)}\mathcal{M}$ and is equivalent to the Riemannian gradient flow in the sense that for $x = \psi(z)$ it holds that

$$\dot x = -\operatorname{grad} f(x).$$

While (17) and (18) are both possible approaches to solving (1) and have been applied in the past Pegios et al.; Tripp et al. (2020), there are a few challenges present these methods:

- **Parameterizing the full atlas of a manifold is hard**: The atlas $\mathcal{A}$ of a manifold typically consists of multiple charts $(\psi_i, U_i, V_i)$. The number of charts as well as their span across the manifold is typically unknown and hard to estimate a-priori.

- **Intrinsic dimension of the manifold unknown**: The dimension of the latent space $\mathcal{Z}$ must correspond to the manifold dimension, the estimation of which is not a trivial task.

- **Generative model training**: The implementation of (18) or (17) require $\psi$ to be invertible and the inverse coordinate map $\psi^{-1}$ to be well approximated by a neural network. It is not straightforward to align this requirement with the standard training objectives commonly used in generative models, such as the likelihood-based loss in VAEs, the denoising score-matching loss in diffusion models, or the adversarial loss in GANs. Further approaches such as M-flows Brehmer & Cranmer (2020) require $\psi$ to be essentially a normalizing flow, which is fundamentally incompatible with common architectures such as U-Nets or transformers.

- **Expensive update rules for PBRGF**: The update rule of PBRGF requires (i) computing the matrix $G_\psi(z)$, which in turn involves the explicit evaluation of the Jacobian $\psi'(z)$, and (ii) inverting $G_\psi(z)$. Both operations become computationally expensive when $\psi$ is implemented by a neural network even for a moderate intrinsic dimension $k$.

- **Limited latent space validity region**: Since $\mathcal{Z}$ is representing the local coordinates of a single chart, the validity of the model $\psi$ is only given in a bounded domain of $\mathcal{Z}$. Optimizing outside of this domain requires changing charts, since otherwise the obtained latent points will have no meaning in the data-space. Estimating the validity domain of $\psi$ is difficult a-priori Notin et al. (2021).

In the above light, the advantages of the algorithms DLF and DRGD described in Sections 4 and 5 are as follows:

- **Easy to parameterize with a neural network**: The central object in our approach is the projection operator onto the data manifold $\mathcal{M}$, which is provably $C^2$ in the tubular neighborhood of any $C^3$ manifold. Assuming $\mathcal{M}$ is compact, standard universal approximation results ensure that the target map $\pi$ can be well-approximated by a single neural network

- **Dimension-agnostic algorithms**: Both DLF and DRGD require no knowledge about the manifold dimension $k$.

- **Alignment with denoising score matching**: DLF and DRGD use an approximation of $\pi$ via the score function of $p_\sigma$ (up to scaling and a residual term of $x$), so the denoising score-matching loss used to train diffusion models is directly aligned with our target quantity.

---

[2]The Euler discretization of the continuous dynamics (18) is also known as natural gradient descent (Amari, 1998, Theorem 1) in the classical Riemannian optimization literature

Our approach imposes no architectural constraints: in our experiments, for example, we successfully use both MLPs and U-Nets.

- **Simple and efficient update rule** The update rules (8) and (12) are highly compatible with modern hardware and software stacks and requires only a single forward–backward pass of the network per iteration (see Remark 4)

- **Convergence guarantees in the embedding space**: Theorem 3 and 5 provide explicit convergence analysis with guarantees on both optimality and feasibility, stated directly for the original data manifold in the data space, where convergence is semantically meaningful.

## A.3 RECOVERY OF ITERATE COMPLEXITY OF RIEMANNIAN GRADIENT DESCENT

In this section we show that in the case of $\sigma = \epsilon = 0$, Theorem 5 recovers the corresponding result for the Riemannian gradient descent. To see this, let us recall the standard complexity guarantee for Riemannian gradient descent under a general smooth non-convex objective $f$: In (Boumal, 2023, Corollary 4.9) it is stated that (taking $R_x(v) = \pi(x + v)$ as the retraction defined on some ball $B_r(x) \cap T_x \mathcal{M}$, where for us $r > 0$ can be picked uniformly over $x \in \mathcal{M}$) if $f \circ \pi$ satisfies the pullback $L$-Lipschitz condition

$$f(\pi(x + v)) - f(x) \leq v^\top \nabla f(x) + \frac{L}{2}\|v\|^2 \text{ for all } v \in B_r(x) \cap T_x \mathcal{M}.$$

then for step-size $\gamma_k = \frac{1}{L}$ the RGD generates iterates $(x_k, v_k = -\gamma_k \pi'(x_k)\nabla f(x_k))$ that satisfy

$$\min_{k=0,\ldots,N-1} \|\operatorname{grad} f(x_k)\| \leq \frac{\sqrt{2L(f(x_0) - f_*)}}{\sqrt{N}}.$$

provided that $v_k \in B_r(x) \cap T_x \mathcal{M}$ for all $k = 0, \ldots, N-1$.

Now we set the following in Theorem 5 in our paper: $s = \pi$, $\sigma = 0$. The pullback $L_0$-Lipschitz condition is satisfied under our assumption $L = \operatorname{Lip}(\nabla f)$ for a different constant $L_0 = \operatorname{Lip}(\nabla(f \circ \pi))$, which can be estimated in terms of $L$, $C$, $\tau$ and $M$ (see the proof of Theorem 5). The rate obtained in Theorem 5 reads then

$$\min_{k=0,\ldots,N} \|\operatorname{grad} f(x_k)\|^2 \leq \frac{4D}{N\gamma_{\min}(1 - L_0\gamma_{\max}/2)},$$

where $\gamma_{\min}(1 - L_0\gamma_{\max}/2)$ can be lower-bounded by a constant constructed from $\tau$ and $L_0$. Moreover, $2D$ is simply an estimation of $f(x_0) - f_*$ via the triangle inequality (see the proof of Theorem 5). Thus in this case both theorems guarantee an $O(1/\sqrt{N})$ best-norm-of-gradient convergence rate with similar constants.

## B MORE ON MANIFOLDS AND DISTANCE FUNCTIONS

In addition to the notation introduced in Section 2.1, we note that $\tau_{\mathcal{M}}$ is also the largest $\tau \geq 0$ such that the map $\{(p, v) \in \mathrm{N}\,\mathcal{M} \mid \|v\| < \tau\} \to \mathbb{R}^d : (p, v) \mapsto p + v$ is a diffeomorphism. By a tubular neighborhood of radius $\tau \in (0, \tau_{\mathcal{M}}]$ we mean a set of the form $\mathcal{T}(\tau) = \{p + v \mid p \in \mathcal{M}, v \in \mathrm{N}_p\,\mathcal{M}, \|v\| < \tau\}$. Moreover, for $x \in \mathbb{R}^d$ let $\operatorname{dist}_{\mathcal{M}}(x) = \inf_{p \in \mathcal{M}}\|x - p\|$ denote the distance function so that $\mathrm{d}(x) = \frac{1}{2}\operatorname{dist}_{\mathcal{M}}(x)^2$. The second fundamental form of $\mathcal{M}$ at a point $p \in \mathcal{M}$ will be denoted by $\mathbb{I}_p$ and is a symmetric bilinear map $\mathbb{I}_p : \mathrm{T}_p\,\mathcal{M} \times \mathrm{T}_p\,\mathcal{M} \to \mathrm{N}_p\,\mathcal{M}$ intrinsic to the manifold $\mathcal{M}$. Fixing some $u \in \mathrm{N}_p\,\mathcal{M}$ we also define the directed second fundamental form $\mathbb{I}_p^u : \mathrm{T}_p\,\mathcal{M} \times \mathrm{T}_p\,\mathcal{M} \to \mathbb{R} : (v, w) \mapsto \langle \mathbb{I}_p(v, w), u\rangle_{\mathrm{N}_p\,\mathcal{M}}$. The Weingarten map $S_p^u$ at a point $p \in \mathcal{M}$ in the direction $u \in \mathrm{N}_p\,\mathcal{M}$ is defined as the unique self-adjoint linear operator $S_p^u : \mathrm{T}_p\,\mathcal{M} \to \mathrm{T}_p\,\mathcal{M}$ such that $\langle w, S_p^u(v)\rangle_{\mathrm{T}_p\,\mathcal{M}} = \mathbb{I}_p^u(v, w)$ for all $v, w \in \mathrm{T}_p\,\mathcal{M}$. A useful operator that has been studied in Abatzoglou (1978); Breiding & Vannieuwenhoven (2021); Alvarez et al. (2004) is

$$H_x = I_{\mathrm{T}_{\pi(x)}\mathcal{M}} + S_{\pi(x)}^{\pi(x)-x} : \mathrm{T}_{\pi(x)}\,\mathcal{M} \to \mathrm{T}_{\pi(x)}\,\mathcal{M}. \tag{19}$$

In Breiding & Vannieuwenhoven (2021) it has been shown (see Lemma 10) that $H_x$ is invertible on $\mathcal{T}$. Now we have the following useful identities that hold for $x \in \mathcal{T}$

$$\pi'(x) = \mathrm{I}_{\mathrm{T}_{\pi(x)}\mathcal{M}} H_x^{-1} \mathrm{P}_{\mathrm{T}_{\pi(x)}\mathcal{M}}$$

$$\nabla\,\mathrm{d}(x) = x - \pi(x)$$

$$\nabla^2\,\mathrm{d}(x) = I - \pi'(x) = I - \mathrm{I}_{\mathrm{T}_{\pi(x)}\mathcal{M}} H_x^{-1} \mathrm{P}_{\mathrm{T}_{\pi(x)}\mathcal{M}}.$$

For any $p \in \mathcal{M}$ the map $\mathrm{pr}_p : \mathcal{M} \to \mathrm{T}_p\,\mathcal{M} : q \mapsto \mathrm{P}_{\mathrm{T}_p\,\mathcal{M}}(q - p)$ is a local diffeomorphism at $p$ with inverse $\psi_p$ defined on $B^{\mathrm{T}_p\,\mathcal{M}}_{\tau_{\mathcal{M}}/4}(0) := B_{\tau_{\mathcal{M}}/4}(0) \cap \mathrm{T}_p\,\mathcal{M}$. Following Divol (2022) we define $\mathcal{M}_k(\tau, M)$ as the set of all $C^k$-manifolds $\mathcal{M}$ as above such that $\tau_{\mathcal{M}} > \tau$ and $\sup_{p \in \mathcal{M}} \|\psi_p\|_{C^k} \leq M$. For the class $\mathcal{M}_k(\tau, M)$, a manifold $\mathcal{M} \in \mathcal{M}_k(\tau, M)$ and $p \in \mathcal{M}$, we denote by $\psi_p$ *always* the inverse of the orthogonal projection $\mathrm{pr}_p$ (also called *Monge* or *graph chart*) restricted to the particular neighborhood $B^{\mathrm{T}_p\,\mathcal{M}}_{\min\{\tau_{\mathcal{M}}, M\}/4}(0)$, which will be (isometrically) identified with the ball $B_{\min\{\tau_{\mathcal{M}}, M\}/4}(0) \subseteq \mathbb{R}^k$. We make frequent use of the following useful result from Divol (2022).

**Lemma 9** (Lemma A.1 in Divol (2022)). *Suppose $\mathcal{M} \in \mathcal{M}_k(\tau, M)$ and $p \in \mathcal{M}$. Then $\psi_p : B^{\mathrm{T}_p\,\mathcal{M}}_{\min\{\tau_{\mathcal{M}}, M\}/4}(0) \to \mathcal{M}$ is well-defined, $C^k$-smooth and the following holds:*

*(i)* *For all $r \leq \min\{\tau_{\mathcal{M}}, M\}/4$ it holds that $B_r(p) \cap \mathcal{M} \subseteq \psi_p(B^{\mathrm{T}_p\,\mathcal{M}}_r(0)) \subseteq B_{8r/7}(p) \cap \mathcal{M}$. For $z \in B^{\mathrm{T}_p\,\mathcal{M}}_{\min\{\tau_{\mathcal{M}}, M\}/4}(0)$ it holds that $\|z\| \leq \|\psi_p(z) - p\| \leq 8\|z\|/7$.*

*(ii)* *There exists a map $W_p : B^{\mathrm{T}_p\,\mathcal{M}}_{\min\{\tau_{\mathcal{M}}, M\}/4}(0) \to \mathrm{N}_p\,\mathcal{M}$ with $W_p'(0) = 0$ and such that $\psi_p(z) = p + z + W_p(z)$ and $\|W_p(z)\| \leq M\|z\|^2$ for all $z \in B^{\mathrm{T}_p\,\mathcal{M}}_{\min\{\tau_{\mathcal{M}}, M\}/4}(0)$.*

*(iii)* *For $G_{\psi_p} : B^{\mathrm{T}_p\,\mathcal{M}}_{\min\{\tau_{\mathcal{M}}, M\}/4}(0) \to \mathbb{R} : z \mapsto \sqrt{\det \psi_p'(z)^\top \psi_p'(z)}$ it holds that $G_{\psi_p}(0) = 1$ and $\nabla G_{\psi_p}(0) = 0$.*

Note that for the graph chart $\psi_p$ we always have

$$\psi_p'(0) = \mathrm{I}_{\mathrm{T}_p\,\mathcal{M}}\,, \quad \psi_p''(0)[\cdot, \cdot] = W_p''(0)[\cdot, \cdot] = \mathbb{I}_p(\cdot, \cdot)\,, \tag{20}$$

and hence $\|\psi_p'(0)\| \leq 1$ and $\|\psi_p''(0)\| = \|\mathbb{I}_p\| \leq 1/\tau_{\mathcal{M}}$.

### B.1 PROPERTIES OF (19)

We study the invertibility and boundedness of the operator (19). For this purpose, let us recall first the definition of the (normalized) curvature radius of $\mathcal{M}$ at $p$ in the direction of $u \in \mathrm{N}_p\,\mathcal{M}$:

$$\frac{1}{\rho(p, u)} = \max_{\substack{v \in \mathrm{T}_p\,\mathcal{M} \\ \mathbb{I}_p^u(v,v) \geq 0}} \frac{\mathbb{I}_p^{u/\|u\|}(v, v)}{\|v\|^2} = \max(\mathrm{eig}(S_p^{u/\|u\|}) \cup \{0\})\,.$$

If $S_p^u$ has only non-positive eigenvalues, then $\mathcal{M}$ is curved away from the unit vector $u$ and thus the curvature radius is infinite. Moreover, we define the (normalized) curvature of $\mathcal{M}$ to be

$$\kappa_p^{\mathcal{M}}(u) = \max\left|\mathrm{eig}(S_p^{u/\|u\|})\right| \text{ for } u \in \mathrm{N}_p\,\mathcal{M}\,,$$

and the maximal curvature by

$$\kappa_{\mathcal{M}} = \max_{(p, u) \in \mathrm{N}\,\mathcal{M}} \kappa_p^{\mathcal{M}}(u)\,. \tag{21}$$

We have the following

**Lemma 10.** *The operator (19) is invertible on $\mathcal{T}(\tau_{\mathcal{M}})$. Moreover, (19) satisfies*[3]

$$\|P_0(x)\| = \|H_x^{-1}\| = \left(1 - \frac{\|\pi(x) - x\|}{\rho(\pi(x), x - \pi(x))}\right)^{-1} \leq \frac{1}{1 - \|x - \pi(x)\|\kappa_{\mathcal{M}}} \leq \frac{1}{1 - \|x - \pi(x)\|/\tau_{\mathcal{M}}}\,,$$

*and if $\psi$ is a local parametrization of $\mathcal{M}$ with $\psi(0) = p$, then*

$$P_0(x) = \psi'(0)\left(\psi'(0)^\top \psi'(0) + \sum_{i=1}^{d} (p - x)_i \nabla^2 \psi_i(0)\right)^{-1} \psi'(0)^\top\,.$$

*In particular, for any $\tau \in (0, \tau_{\mathcal{M}})$, it holds*

$$\sup_{x \in \mathcal{T}(\tau)} \|H_x^{-1}\| < \infty\,.$$

---

[3]In (Breiding & Vannieuwenhoven, 2021, Theorem 4.3) the quantity $\|H_x^{-1}\|$ has been shown to equal the condition number of a certain critical point problem associated with $\mathcal{M}$.

*Proof.* In (Breiding & Vannieuwenhoven, 2021, Lemma A.2) the following condition has been established: Let $\mathcal{S} = \{(a, p) \in \mathbb{R}^n \times \mathcal{M} \mid a - p \in \mathrm{N}_p \mathcal{M}\}$. Then $\mathcal{S}$ is diffeomorphic to the normal bundle $\mathrm{N}\mathcal{M}$ via the diffeomorphism $\Phi : \mathrm{N}\mathcal{M} \to \mathcal{S} : (v, p) \mapsto (p + \mathrm{I}_{\mathrm{N}_p \mathcal{M}}(v), p)$. Consider the operator $\Pi : \mathcal{S} \to \mathbb{R}^n : (a, p) \mapsto a$ and the domain where its differential is invertible $\mathcal{W} = \{(a, p) \in \mathcal{S} \mid \Pi'(a, p) : \mathrm{T}_{(a,p)} \mathcal{S} \to \mathbb{R}^n \text{ invertible}\}$. Then $H_x$ is invertible iff $(x, \pi(x)) \in \mathcal{W}$. But $\Pi \circ \Phi : \mathrm{N}\mathcal{M} \mapsto \mathbb{R}^n : (v, p) \mapsto p + \mathrm{I}_{\mathrm{N}_p \mathcal{M}}(v)$ being a diffeomorphism (and thus having an inveritble differential) is precisely the condition in the definition of the tubular neighborhood $\mathcal{T}$. The formulas for $P_0(x)$ in local coordinates as well as $\|P_0(x)\|$ are given in (Abatzoglou, 1978, Theorem 4.1, Corollary 4.1). To see that $P_0(x)$ is bounded on $\mathcal{T}(\tau)$ for any $\tau \in (0, \tau_{\mathcal{M}})$ it sufficies to note that in the tubular neighborhood $\mathcal{T} = \mathcal{T}(\tau_{\mathcal{M}})$ we always have $\|\pi(x) - x\| < \rho(\pi(x), x - \pi(x))$ and that $\overline{\mathcal{T}(\tau)}$ is a compact subset thereof. The second inequality follows from $1/\rho(p, u) \le \kappa_{\mathcal{M}}$. □

Now let us derive bounds for the quantity $\|P_0(\pi(x)) - P_0(x)\|$ when $x \in \mathcal{T}(\tau_{\mathcal{M}})$.

**Lemma 11.** *If $x \in \mathcal{T}(\tau_{\mathcal{M}})$, then*

$$\|P_0(\pi(x)) - P_0(x)\| \le \kappa^{\mathcal{M}}_{\pi(x)}(x - \pi(x)) \left(1 - \frac{\|\pi(x) - x\|}{\rho(\pi(x), x - \pi(x))}\right)^{-1} \|x - \pi(x)\|,$$

$$\le \frac{\|x - \pi(x)\|\kappa_{\mathcal{M}}}{1 - \|x - \pi(x)\|\kappa_{\mathcal{M}}} \le \frac{\|x - \pi(x)\|/\tau_{\mathcal{M}}}{1 - \|x - \pi(x)\|/\tau_{\mathcal{M}}}.$$

*Proof.* We clearly have for $x \in \mathcal{T}$

$$P_0(\pi(x)) - P_0(x) = \mathrm{I}_{\mathrm{T}_{\pi(x)} \mathcal{M}}(I - H_x^{-1}) \mathrm{P}_{\mathrm{T}_{\pi(x)} \mathcal{M}}.$$

Moreover, $(I - H_x^{-1}) : \mathrm{T}_{\pi(x)} \mathcal{M} \to \mathrm{T}_{\pi(x)} \mathcal{M}$ is symmetric with eigenvalues

$$\mathrm{eig}(I - H_x^{-1}) = \left\{\frac{\zeta}{1 + \zeta} \mid \zeta \in \mathrm{eig}(S_{\pi(x)}^{\pi(x) - x})\right\} = \left\{\frac{-\|\pi(x) - x\|\zeta}{1 - \|\pi(x) - x\|\zeta} \mid \zeta \in \mathrm{eig}(S_{\pi(x)}^u)\right\},$$

where $u = \frac{x - \pi(x)}{\|x - \pi(x)\|}$. Thus

$$\|P_0(\pi(x)) - P_0(x)\| \le \max_{\zeta \in \mathrm{eig}(S_{\pi(x)}^u)} \left|\frac{\zeta}{1 - \|x - \pi(x)\|\zeta}\right| \|x - \pi(x)\|$$

$$\le \|S_{\pi(x)}^u\| \left(1 - \frac{\|\pi(x) - x\|}{\rho(\pi(x), x - \pi(x))}\right)^{-1} \|x - \pi(x)\|,$$

which shows the first inequality. The second and third inequalities follow from $1/\rho(p, u) \le \kappa_{\mathcal{M}} \le 1/\tau_{\mathcal{M}}$. □

The next lemma establishes a bound on the Lipschitz-constant of $P_0$ of some $\mathcal{M} \in \mathcal{M}_k(\tau, M)$ in terms of $M$ and $\tau_{\mathcal{M}}$.

**Lemma 12.** *If $\mathcal{M} \in \mathcal{M}_k(\tau, M)$, then*

$$\sup_{x \in \mathcal{T}(\tau)} \|P_0'(x)\| \le \left(\frac{1}{1 - \tau/\tau_{\mathcal{M}}}\right)^2 \left((\frac{3}{\tau_{\mathcal{M}}} + \tau M)(\frac{1}{1 - \tau/\tau_{\mathcal{M}}})^2 + \frac{2}{\tau_{\mathcal{M}}}\right)$$

*Proof.* Let $x \in \mathcal{T}(\tau)$ where $\tau \in (0, \tau_{\mathcal{M}})$. First let us relate the quantity $\pi(x)$ and a fixed chart $\psi : \mathcal{V} \to \mathcal{U}$ with $\pi(x) \in \mathcal{U}$ for all $x \in \mathcal{W}$ for some (small enough) open set $\mathcal{W} \subseteq \mathbb{R}^d$. The map $h_\psi(z) = \frac{1}{2}\|x - \psi(z)\|^2$ attains its minimum in some $z_\psi(x) \in \mathcal{V}$ and if $F^\psi(z, x) = h'_\psi(z) = \psi'(z)(\psi(z) - x)$, then $F^\psi(z_\psi(x), x) = 0$. Moreover, we have $\pi(x) = \psi(z_\psi(x))$ and hence for $w \in \mathbb{R}^d$

$$P_0'(x)[w, w] = \pi''(x)[w, w] = \psi''(z_\psi(x))[z'_\psi(x)[w], z'_\psi(x)[w]] + \psi'(z_\psi(x))[z''_\psi(x)[w, w]],$$

and hence

$$\|P_0'(x)\| \le \|\psi''(z_\psi(x))\|\|z'_\psi(x)\|^2 + \|\psi'(z_\psi(x))\|\|z''_\psi(x)\|$$

By the implicit function theorem we can bound the derivatives of $z_\psi$ in terms of derivatives of $F^\psi$. Indeed, we have (here again $w \in \mathbb{R}^d$ and $v \in \mathbb{R}^k$ are place-holder vectors to express the differentials)

$$
\begin{aligned}
0 &= F_z^\psi(z_\psi(x), x)[v, z_\psi'(x)[w]] + F_x^\psi(z_\psi(x), x)[v, w] \,, \\
0 &= F_z^\psi(z_\psi(x), x)[v, z_\psi''(x)[w, w]] + F_{zz}^\psi(z_\psi(x), x)[v, z_\psi'(x)[w], z_\psi'(x)[w]] \\
&\quad + 2 F_{xz}^\psi(z_\psi(x), x)[v, z_\psi(x)[w], w] + F_{xx}^\psi(z_\psi(x), x)[v, w, w] \,,
\end{aligned}
\tag{22}
$$

with

$$
\begin{aligned}
F_z^\psi(z, x)[v, v] &= \langle \psi'(z)[v], \psi'(z)[v] \rangle + \langle \psi(z) - x, \psi''(z)[v, v] \rangle \,, \\
F_x^\psi(z, x)[w, v] &= \langle \psi'(z)[v], w \rangle \,, \\
F_{zz}^\psi(z, x)[v, v, v] &= 3 \langle \psi''(z)[v, v], \psi'(z)[v] \rangle + \langle \psi(z) - x, \psi'''(z)[v, v, v] \rangle \,, \\
F_{xz}^\psi(z, x)[w, v, v] &= \langle \psi''(z)[v, v], w \rangle \,, \\
F_{xx}^\psi(z, x)[w, v, v] &= 0 \,.
\end{aligned}
$$

Now we set $\psi = \psi_p$ for the graph chart defined on $\mathcal{V} = B_{\min\{\tau, M\}/4}^{\mathrm{T}_p \mathcal{M}}(0)$, where $p = \pi(x)$. In this case $z = z_\psi(x) = 0$ and $\|F_z^\psi(0, x)^{-1}\| = \|P_0(x)\| \le (1 - \tau/\tau_\mathcal{M})^{-1}$ by Lemma 10. Solving for $z_\psi'(x)$ and $z_\psi''(x)$ in (22), using (20) with $\|\mathbb{I}_p\| \le 1/\tau_\mathcal{M}$ and taking norms yields then

$$
\|z_\psi'(x)\| \le \frac{1}{1 - \tau/\tau_\mathcal{M}} \,,
$$

$$
\|z_\psi''(x)\| \le \frac{1}{1 - \tau/\tau_\mathcal{M}} \left( (\frac{3}{\tau_\mathcal{M}} + \tau M) \|z_\psi'(x)\|^3 + \frac{1}{\tau_\mathcal{M}} \|z_\psi'(x)\| \right) \,.
$$

Plugging this back into the upper bound of $\|P_0'(x)\|$ and using once again (20) finishes the proof $\quad\square$

The next lemma establishes a lower bound on the distance between a point $x$ in a tubular neighborhood and any other point that is sufficiently away from $p = \pi(x)$ uniformly in $x$.

**Lemma 13.** *Let $\tau \in (0, \tau_\mathcal{M})$ and let $\eta > 0$. Then*

$$
\inf_{x \in \mathcal{T}(\tau)} \frac{1}{2} \operatorname{dist}_{\mathcal{M} \setminus B_\eta(\pi(x))}(x)^2 - \frac{1}{2} \operatorname{dist}_\mathcal{M}(x)^2 \ge \frac{1}{2} \frac{\tau_\mathcal{M} - \tau}{\tau_\mathcal{M} + \tau} \eta^2 > 0 \,.
$$

*Proof.* Let $\tau' = \frac{\tau + \tau_\mathcal{M}}{2} \in (0, \tau_\mathcal{M})$. Take $x \in \mathcal{T}(\tau)$ and set $p = \pi(x)$ and $y = p + \tau' \widehat{n}$ with $\widehat{n} = \frac{x - p}{\|x - p\|}$. Then $y \in \mathcal{T}(\tau_\mathcal{M})$ with $\operatorname{dist}_\mathcal{M}(y) = \tau'$ and thus $\overline{B}_{\tau'}(y) \cap \mathcal{M} = \{p\}$. Then, since $\mathcal{M} \setminus B_\eta(p) \subseteq \mathbb{R}^d \setminus (\overline{B}_{\tau'}(y) \cup B_\eta(p))$, it is sufficient to show

$$
\inf_{q \in \mathbb{R}^d \setminus (\overline{B}_{\tau'}(y) \cup B_\eta(p))} \|x - q\|^2 - \|x - p\|^2 \ge \frac{\tau_\mathcal{M} - \tau}{\tau_\mathcal{M} + \tau} \eta^2 \,.
$$

To see this, note that $q \in \mathbb{R}^d \setminus (\overline{B}_{\tau'}(y) \cup B_\eta(p))$ implies $\|y - q\| \ge \tau' = \|y - p\|$ and $\|q - p\| \ge \eta$, as well as $2 \langle p - q, y - q \rangle \ge \|p - q\|^2$. Then, abbreviating $t = \frac{\|x - p\|}{\|y - p\|} \in (0, 1)$, we have $x = ty + (1 - t)p = p + t(y - p)$ and

$$
\begin{aligned}
\|x - q\|^2 - \|x - p\|^2 &= \|ty + (1 - t)p - q\|^2 - t^2 \|y - p\|^2 \\
&\ge \|t(y - q) + (1 - t)(p - q)\|^2 - t^2 \|y - q\|^2 \\
&= (1 - t)^2 \|p - q\|^2 + 2t(1 - t) \langle p - q, y - q \rangle \\
&\ge (1 - t) \|p - q\|^2 \\
&\ge \frac{\tau_\mathcal{M} - \tau}{\tau_\mathcal{M} + \tau} \eta^2 \,.
\end{aligned}
$$

$\square$

## B.2 DENSITIES ON MANIFOLDS

Given a manifold $\mathcal{M}$ as in Section B and any chart $\psi : \mathcal{V} \to \mathcal{U} \subseteq \mathcal{M}$, the volume measure $\mathrm{Vol}_{\mathcal{M}}$ is uniquely defined by

$$\mathrm{Vol}_{\mathcal{M}}(E) = \int_{\psi^{-1}(E)} G_\psi(z) \, \mathrm{d}z \ \text{ for } E \subseteq \mathcal{U} \text{ Borel measurable.}$$

Here $G_\psi(z) = \sqrt{\det \psi'(z)^\top \psi'(z)}$ and $\mathrm{Vol}_{\mathcal{M}}$ is independent of the chart $\psi$. If $\mu \in \mathcal{P}(\mathbb{R}^d)$ is absolutely continuous w.r.t. $\mathrm{Vol}_{\mathcal{M}}$, then for the density $\mu(y) := \frac{\mathrm{d}\,\mu}{\mathrm{d}\,\mathrm{Vol}_{\mathcal{M}}}(y)$ on $\mathcal{M}$ we have the local representations

$$\mu(\psi(z)) = \frac{\mathrm{d}\,\lambda}{\mathrm{d}(\psi^{-1}\# \mathrm{Vol}_{\mathcal{M}})}(z), \quad \lambda(z) = G_\psi(z)\mu(\psi(z))$$

with $\lambda = \psi^{-1}\#\mu$ the pullback under the chart $\psi$ with density $\lambda(z) = \frac{\mathrm{d}\,\lambda}{\mathrm{d}\,m_{\mathcal{V}}}(z)$ w.r.t. the Lebesgue measure $m_{\mathcal{V}}$ on $\mathcal{V}$. In particular when $\mathcal{M} \in \mathcal{M}_k(\tau, M)$ with graph chart $\psi_p$, we have

$$\lambda(0) = \mu(p), \quad \nabla\lambda(0) = \mathrm{grad}_{\mathcal{M}}\,\mu(p) \in \mathrm{T}_p\mathcal{M} \cong \mathbb{R}^k. \tag{23}$$

## C   MORE ON THE STEIN SCORE FUNCTION

In this section we derive the representations (4). For $\sigma \geq 0$ let $\mathcal{N}(0, \sigma^2 I_d)$ denote the Gaussian distribution with mean zero and variance $\sigma^2$ and for $\sigma > 0$ the Gaussian (heat) kernel in the ambient space $\mathbb{R}^d$ by

$$\varphi_\sigma(x) = \frac{1}{Z_\sigma} e^{-\|x\|^2/2\sigma^2}, \quad Z_\sigma = \frac{1}{(2\pi\sigma^2)^{d/2}}.$$

Then $p_\sigma = \mathcal{N}(0, \sigma^2 I) * \mu$ has for $\sigma > 0$ a fully supported $C^\infty$-density, denoted by $p_\sigma$ as well, given by

$$p_\sigma(x) = \int_{\mathcal{M}} \varphi_\sigma(x - y) \, \mathrm{d}\mu(y), \quad x \in \mathbb{R}^d.$$

Let $\nu_{x,\sigma}$ be the posterior of observing $x$ under $p_\sigma$ with prior $\mu$, i.e.

$$\nu_{x,\sigma}(E) = \frac{1}{p_\sigma(x)} \int_E \varphi_\sigma(x - y) \, \mathrm{d}\mu(y), \quad E \subseteq \mathbb{R}^d \text{ Borel}.$$

We have the following representation

**Lemma 14.** *For each $\sigma > 0$ representations (4) hold.*

*Proof.* Clearly

$$\nabla p_\sigma(x) = -\frac{1}{\sigma^2} \int_{\mathcal{M}} (x - y)\varphi_\sigma(x - y) \, \mathrm{d}\mu(y), \quad x \in \mathbb{R}^d,$$

$$\nabla^2 p_\sigma(x) = -\frac{1}{\sigma^2} \left( p_\sigma(x) I - \frac{1}{\sigma^2} \int_{\mathcal{M}} (x - y)(x - y)^\top \varphi_\sigma(x - y) \, \mathrm{d}\mu(y) \right), \quad x \in \mathbb{R}^d.$$

and hence

$$\nabla \log p_\sigma(x) = \frac{\nabla p_\sigma(x)}{p_\sigma(x)} = -\frac{1}{\sigma^2}(x - \mathbb{E}\nu_{x,\sigma}),$$

which yields the first representation in (4). Further

$$\frac{\nabla^2 p_\sigma(x)}{p_\sigma(x)} = -\frac{1}{\sigma^2} \left( I - \frac{1}{\sigma^2} \int_{\mathcal{M}} (x - y)(x - y)^\top \, \mathrm{d}\nu_{x,\sigma}(y) \right)$$

and hence

$$
\begin{aligned}
\nabla^2 \log p_\sigma(x) &= \frac{\nabla^2 p_\sigma(x)}{p_\sigma(x)} - \frac{\nabla p_\sigma(x) \nabla p_\sigma(x)^\top}{p_\sigma(x)^2} \\
&= -\frac{1}{\sigma^2} \left( I - \frac{1}{\sigma^2} \int_{\mathcal{M}} (x-y)(x-y)^\top \, \mathrm{d}\nu_{x,\sigma}(y) \right) \\
&\quad - \frac{1}{\sigma^4}(x - \mathbb{E}\nu_{x,\sigma})(x - \mathbb{E}\nu_{x,\sigma})^\top \\
&= -\frac{1}{\sigma^2} I + \frac{1}{\sigma^4} \left( \int_{\mathcal{M}} yy^\top \, \mathrm{d}\nu_{x,\sigma}(y) - (\mathbb{E}\nu_{x,\sigma})(\mathbb{E}\nu_{x,\sigma})^\top \right) \\
&= -\frac{1}{\sigma^2} I + \frac{1}{\sigma^4} \operatorname{Cov}(\nu_{x,\sigma}) \,,
\end{aligned}
$$

which yields the second representation in (4). $\qquad\square$

## D  VARIANCE-EXPLODING DIFFUSION MODELS

In the variance-exploding scheme of score-based diffusion models Song et al. (2020); Tang & Zhao (2025), one considers the following stochastic differential equation on the finite interval $[0, T]$:

$$
\mathrm{d}X_t = \sqrt{2t} \, \mathrm{d}W_t \,, \quad t \in [0, T] \,, \quad X_0 \sim \mu \,. \tag{24}
$$

The solution $X_t$ of this SDE is distributed according to $X_t \sim p_{\sigma(t)}$, where $\sigma^2(t) = t^2$. To sample from $\mu$ one exploits the fact that the reverse SDE

$$
\mathrm{d}\overline{X}_t = 2(T-t)\nabla \log p_{\sigma(T-t)}(\overline{X}_t) \, \mathrm{d}t + \sqrt{2(T-t)} \, \mathrm{d}W_t \,, \quad t \in [0, T] \,, \quad \overline{X}_0 \sim p_{\sigma(T)} \,, \tag{25}
$$

satisfies $\overline{X}_t \sim p_{\sigma(T-t)}$ and in particular $\overline{X}_0 \sim p_0 = \mu$. Here the initial distribution is approximated by $p_{\sigma(T)} \approx \mathcal{N}(0, \sigma(T)^2 I)$ and unknown score $\nabla \log p_{\sigma(T-t)}$ is learned by minimizing the conditional score matching loss defined as

$$
L_{\mathrm{CSM}}(s) = \mathbb{E}_{t \sim \mathrm{Unif}[0,T]} \mathbb{E}_{x_0 \sim \mu} \mathbb{E}_{x \sim p_{\sigma(t)}(\cdot \mid x_0)} \sigma(t)^2 \| s_{\sigma(t)}(x) - \nabla \log p_{\sigma(t)}(x \mid x_0) \|^2 \,, \tag{26}
$$

and which admits the unique minimizer $s_\sigma(x) = \nabla \log p_\sigma(x)$. The loss $L_{\mathrm{CSM}}$ can be evaluated because $\nabla \log p_\sigma(x \mid x_0) = -\frac{1}{\sigma^2}(x - x_0)$ is known explicitly.

## E  PROOF OF THEOREM 1

### E.1  NONASYMPTOTIC LAPLACE METHOD

In this section we derive some non-asymptotic error estimates for the Laplace method Hashorva et al. (2015); Hwang (1980), which concerns itself with the asymptotic of integrals of the form

$$
\int_{\mathcal{V}} f(z) e^{-\frac{1}{\sigma^2} h(z)} \, \mathrm{d}z \quad (\sigma \to 0)
$$

for some functions $f, h : \mathcal{V} \to \mathbb{R}$ from an open set $\mathcal{V} \subseteq \mathbb{R}^k$, where $h$ is non-negative and attains a unique minimum in, say, $z = 0 \in \mathcal{V}$. While there have been results providing such an error estimation to the first order expansion Inglot & Majerski (2014); Majerski (2015); Lapinski (2019), we need an error estimate up to the second order expansion, as provided in Theorem 16. For the sake of completeness we also include the statement and proof for the first order expansion in Theorem 15. Before we state the results, we need to introduce some notation and conventions. For any $f \in C^0(\mathcal{V})$ and $h \in C^2(\mathcal{V})$, respectively we write

$$
\begin{aligned}
\zeta_f(z) &= f(z) - f(0) \overset{z \to 0}{=} o(1) \\
\chi_h(z) &= h(z) - h(0) - \nabla h(0)^\top z - \frac{1}{2} z^\top \nabla^2 h(0) z \overset{z \to 0}{=} o(\|z\|^2)
\end{aligned}
$$

for its first and second order remainder terms and abbreviate

$$\zeta_f^\sigma(w) = \zeta_f(\sigma w) \text{ and } \chi_h^\sigma(w) = \frac{1}{\sigma^2}\chi_h(\sigma w) \text{ for } \sigma > 0\,.$$

Let us note the following: If additionally $f \in C^1(\mathcal{V})$ and $g \in C^3(\mathcal{V})$, then $\zeta_f(z) = O(\|z\|)$ and $\chi_g(z) = O(\|z\|^3)$ for $z \to 0$, i.e. there exist some $\eta > 0$ and $C > 0$ such that $B_\eta(0) \subseteq \mathcal{V}$ and

$$|\zeta_f(z)| \le C\|z\| \text{ and } |\chi_h(z)| \le C\|z\|^3 \text{ for all } z \in B_\eta(0)\,. \tag{27}$$

In particular for $\sigma > 0$ and $\beta : [0,\infty) \to [0,\infty)$ it holds

$$\sup_{\|w\| \le \min\{\eta/\sigma, \beta(\sigma)\}} |\zeta_f^\sigma(w)| \le C\sigma\beta(\sigma) \text{ and } \sup_{\|w\| \le \min\{\eta/\sigma, \beta(\sigma)\}} |\chi_g^\sigma(w)| \le C\sigma\beta(\sigma)^3\,. \tag{28}$$

For a non-negative function $h$ with a global minimum at $z = 0$ we define

$$\gamma_h(\delta) = \inf_{\substack{z \in \mathcal{V} \\ \|z\| \ge \delta}} h(z) - h(0)\,.$$

We say that $h$ satisfies a *local quadratic growth condition* in $B_\eta(0) \subseteq \mathcal{V}$ if there exists a $c > 0$ with

$$h(z) - h(0) \ge c\|z\|^2 \text{ for all } z \in B_\eta(0)\,, \tag{29}$$

and that $h$ admits *minimum separation* outside of $B_\eta(0)$ in $\mathcal{V}$ if there exists a $\Delta > 0$ with

$$h(z) - h(0) \ge \Delta > 0 \text{ for all } z \in \mathcal{V} \setminus B_\eta(0)\,. \tag{30}$$

Clearly, if $h$ satisfies (27), then $h$ also satisfies (29) (for a potentially smaller $\eta$). On the other hand, the global assumption (30) cannot be inferred from properties of $h$ around $z = 0$ alone. Together, (29) and (30) imply that $\gamma_h$ can be bounded below by

$$\gamma_h(\delta) \ge \min\{c\delta^2, \Delta\}\,, \tag{31}$$

which is crucial to obtain a quantitative bound on the convergence of the Laplace method. While we state the following results in this full generality, in our application we have $B_\eta(0) = \mathcal{V}$ and thus (30) is not needed, with (27) and (29) holding globally on $\mathcal{V}$ and implying that

$$\gamma_h(\delta) \ge c\delta^2\,. \tag{32}$$

**Theorem 15** (First Order Laplace method). *Let $h \in C^3(\mathcal{V})$ with $\nabla h(0) = 0$ and $f \in C^1(\mathcal{V})$ for some open $B_r(0) \subseteq \mathcal{V} \subseteq \mathbb{R}^k$. Let $\eta \in (0, r]$ and $C, c, \Delta > 0$ be such that (27), (29) and (30) hold. Then for all $\sigma \in (0, \overline{\sigma})$ with $\overline{\sigma} = \min\{\eta^2, (\log(2)/(2C))^2\}$ with it holds*

$$\left| e^{\frac{1}{\sigma^2}h(0)} \frac{\sqrt{\det \Sigma}}{Z_\sigma} \int_\mathcal{V} f(z)e^{-\frac{1}{\sigma^2}h(z)}\,dz - f(0) \right| \le E_1(\sigma; f, h, r) \tag{33}$$

*with $\Sigma = \nabla^2 h(0)$ and $E_1(\sigma; f, h, r) = O(\sigma|\log \sigma|^3)$ for $\sigma \to 0$ given in (34).*

*Proof.* Without loss of generality we can assume $h(0) = 0$. For brevity we also write $Z_\sigma^\Sigma = \frac{Z_\sigma}{\sqrt{\det \Sigma}}$ for the normalizing constant. Let $\alpha : [0,\infty) \to [0,\infty)$ be any function and denote $B(\sigma) = B_{\alpha(\sigma)}(0)$. We can split

$$\frac{1}{Z_\sigma^\Sigma} \int_\mathcal{V} f(z)e^{-\frac{1}{\sigma^2}h(z)}\,dz = \frac{1}{Z_\sigma^\Sigma} \int_{\mathcal{V}\setminus B(\sigma)} f(z)e^{-\frac{1}{\sigma^2}h(z)}\,dz + \frac{1}{Z_\sigma^\Sigma} \int_{B(\sigma)} f(z)e^{-\frac{1}{\sigma^2}h(z)}\,dz\,.$$

We will craft $\alpha : [0,\infty) \to [0,\infty)$ in such a way that the first integral vanishes and the second converges to $f(0)$ for $\sigma \to 0$. For the first integral we have

$$\left| \frac{1}{Z_\sigma^\Sigma} \int_{\mathcal{V}\setminus B(\sigma)} f(z)e^{-\frac{1}{\sigma^2}h(z)}\,dz \right| \le \frac{1}{Z_\sigma^\Sigma} e^{-\frac{1}{\sigma^2}\gamma_h(\alpha(\sigma))} \|f\|_{L^1(\mathcal{V})}\,.$$

For the second integral let us write $f(z) = f(0) + \zeta_f(z)$ and split

$$\frac{1}{Z_\sigma^\Sigma} \int_{B(\sigma)} f(z)e^{-\frac{1}{\sigma^2}h(z)}\,dz = \frac{f(0)}{Z_\sigma^\Sigma} \int_{B(\sigma)} e^{-\frac{1}{\sigma^2}h(z)}\,dz + \frac{1}{Z_\sigma^\Sigma} \int_{B(\sigma)} \zeta_f(z)e^{-\frac{1}{\sigma^2}h(z)}\,dz$$

$$=: I(\sigma) + J(\sigma)\,.$$

First let us estimate the difference between $I(\sigma)$ and $f(0)$. We have, using the substitution $z = \sigma w$ and abbreviation $B_{\mathcal{V}}(\sigma) = (\mathcal{V} \cap B(\sigma))/\sigma$,

$$|I(\sigma) - f(0)|$$

$$\leq f(0) \left| \frac{1}{Z_1^\Sigma} \int_{B_{\mathcal{V}}(\sigma)} e^{-\frac{1}{\sigma^2} h(\sigma w)} \, \mathrm{d}w - 1 \right|$$

$$\leq \frac{f(0)}{Z_1^\Sigma} \left( \left| \int_{B_{\mathcal{V}}(\sigma)} e^{-\frac{1}{2} w^\top \Sigma w} (e^{-\chi_h^\sigma(w)} - 1) \, \mathrm{d}w \right| + \left| \int_{\mathbb{R}^k \backslash B_{\mathcal{V}}(\sigma)} e^{-\frac{1}{2} w^\top \Sigma w} \, \mathrm{d}w \right| \right)$$

$$=: \frac{f(0)}{Z_1^\Sigma} \left( |I_1(\sigma)| + |I_2(\sigma)| \right).$$

Estimating $I_1(\sigma)$ we obtain

$$|I_1(\sigma)| \leq Z_1^\Sigma \sup_{w \in B_{\mathcal{V}}(\sigma)} \left| e^{-\chi_h^\sigma(w)} - 1 \right|.$$

To estimate $I_2(\sigma)$, let us note that $\Sigma^{1/2}(\mathbb{R}^k \backslash B_R(0)) \subseteq \mathbb{R}^k \backslash B_{\lambda_{\min}(\Sigma)R}(0)$. Then, via the substitution $u = \Sigma^{1/2} w$, we obtain

$$|I_2(\sigma)| \leq \left| \int_{\mathbb{R}^k \backslash B_{\mathcal{V}}(\sigma)} e^{-\frac{1}{2} w^\top \Sigma w} \, \mathrm{d}w \right| \leq \left| \int_{\mathbb{R}^k \backslash B_{\min\{r, \alpha(\sigma)\}/\sigma}(0)} e^{-\frac{1}{2} w^\top \Sigma w} \, \mathrm{d}w \right|$$

$$\leq Z_1^\Sigma \frac{1}{(2\pi)^{k/2}} \left| \int_{\Sigma^{1/2}(\mathbb{R}^k \backslash B_{\min\{r, \alpha(\sigma)\}/\sigma}(0))} e^{-\frac{1}{2} \|u\|^2} \, \mathrm{d}u \right| \leq Z_1^\Sigma \, \mathrm{G}_0 \left( \lambda_{\min}(\Sigma) \frac{\min\{r, \alpha(\sigma)\}}{\sigma} \right),$$

with the Gaussian tail

$$\mathrm{G}_0(R) = \frac{1}{(2\pi)^{n/2}} \int_{\mathbb{R}^n \backslash B_R(0)} e^{-\frac{1}{2} \|u\|^2} \, \mathrm{d}u.$$

Next we estimate $J(\sigma)$ via

$$|J(\sigma)| \leq \frac{1}{Z_1^\Sigma} \int_{B_{\mathcal{V}}(\sigma)} \zeta_f(\sigma w) e^{-\frac{1}{2} w^\top \Sigma w} e^{-\chi_h^\sigma(w)} \, \mathrm{d}w$$

$$\leq \left( \sup_{w \in B_{\mathcal{V}}(\sigma)} |\zeta_f(\sigma w)| \right) \left( \sup_{w \in B_{\mathcal{V}}(\sigma)} |e^{-\chi_h^\sigma(w)} - 1| + 1 \right).$$

In all, we obtain the following error estimate

$$\left| e^{\frac{1}{\sigma^2} h(0)} \frac{1}{Z_\sigma^\Sigma} \int_{\mathcal{V}} f(z) e^{-\frac{1}{\sigma^2} h(z)} \, \mathrm{d}z - f(0) \right|$$

$$\leq \frac{1}{Z_\sigma^\Sigma} e^{-\frac{1}{\sigma^2} \gamma(\alpha(\sigma))} \|f\|_{L^1(\mathcal{V})} + \frac{f(0)}{Z_1^\Sigma} (|I_1(\sigma)| + |I_2(\sigma)|) + |J(\sigma)|$$

$$\leq \frac{1}{Z_\sigma^\Sigma} e^{-\frac{1}{\sigma^2} \gamma(\alpha(\sigma))} \|f\|_{L^1(\mathcal{V})}$$

$$+ f(0) \left( \sup_{w \in B_{\mathcal{V}}(\sigma)} \left| e^{-\chi_h^\sigma(w)} - 1 \right| + \mathrm{G}_0 \left( \lambda_{\min}(\Sigma) \frac{\min\{r, \alpha(\sigma)\}}{\sigma} \right) \right)$$

$$+ \left( \sup_{w \in B_{\mathcal{V}}(\sigma)} |\zeta_f(\sigma w)| \right) \left( \sup_{w \in B_{\mathcal{V}}(\sigma)} |e^{-\chi_h^\sigma(w)} - 1| + 1 \right).$$

Thus we observe that we need to select $\alpha$ in such a way that for $\sigma \to 0$

(i) $(Z_\sigma^\Sigma)^{-1} \exp(-\gamma_h(\alpha(\sigma))/\sigma^2) \to 0$

(ii) $\mathrm{G}_0 \left( \lambda_{\min}(\Sigma) \frac{\min\{r, \alpha(\sigma)\}}{\sigma} \right) \to 0$

(iii) $\sup_{w \in B_{\mathcal{V}}(\sigma)} |\chi_h^\sigma(w)| \to 0$

(iv) $\sup_{w \in B_{\mathcal{V}}(\sigma)} |\zeta_f(\sigma w)| \to 0$

Let us make the Ansatz $\alpha(\sigma) = \sigma\beta(\sigma)$ for some $\beta : (0, \infty) \to (0, \infty)$ with $\beta(\sigma) \to \infty$ for $\sigma \to 0$. Taking $\beta(\sigma) = |\log(\sigma)|$, noting that $|e^{-x} - 1| \le 2|x|$ for $|x| \le \log 2$ as well as using (31), (28) and the fact that $\sigma|\log(\sigma)|, \sigma|\log(\sigma)|^3 \le \sigma^{1/2}$ when $\sigma \in (0, 1)$, yields the following estimates

$$\exp(-\frac{1}{\sigma^2}\gamma(\alpha(\sigma))) \le \exp(-\min\{c\log(\sigma)^2, \Delta\sigma^{-2}\}),$$

$$G_0\left(\lambda_{\min}(\Sigma)\frac{\min\{r, \alpha(\sigma)\}}{\sigma}\right) \le G_0\left(\lambda_{\min}(\Sigma)\min\{r\sigma^{-1}, |\log(\sigma)|\}\right),$$

$$\sup_{w \in B_{\mathcal{V}}(\sigma)}\left|e^{-\chi_h^\sigma(w)} - 1\right| \le 2\sup_{w \in B_{\mathcal{V}}(\sigma)}|\chi_h^\sigma(w)| \le 2C\sigma\beta(\sigma)^3 = 2C\sigma|\log(\sigma)|^3,$$

$$\sup_{w \in B_{\mathcal{V}}(\sigma)}|\zeta_f(\sigma w)| \le C\sigma\beta(\sigma) = C\sigma|\log(\sigma)|,$$

when $0 < \sigma \le \overline{\sigma} := \min\{\eta^2, (\log 2/(2C))^2\}$. Plugging all these estimates back yields (33) with

$$
\begin{aligned}
E(\sigma) = {}& \frac{1}{Z_\sigma^\Sigma}\exp(-\min\{c\log(\sigma)^2, \Delta\sigma^{-2}\})\|f\|_{L^1(\mathcal{V})} \\
& + f(0)\left(2C\sigma|\log(\sigma)|^3 + G_0(\lambda_{\min}(\Sigma)\min\{r\sigma^{-1}, |\log(\sigma)|\})\right) \\
& + C\sigma|\log(\sigma)|(1 + 2C\sigma|\log(\sigma)|^3) \\
\overset{\sigma \to 0}{=} {}& O(\sigma\log(\sigma)^3) = \widetilde{O}(\sigma).
\end{aligned}
\tag{34}
$$

$\square$

The next result provides an nonasymptotic estimate of the Laplace method for the second order expansion. We will need a version that allows for a non-zero gradient of $f$, as long as it is contained in the subspace $\{(q, -q) \mid q \in \mathbb{R}^n\} \subseteq \mathbb{R}^n \times \mathbb{R}^n$.

**Theorem 16** (Second Order). *Let $f, h \in C^3(\mathcal{V} \times \mathcal{V})$ be such that $f(0) = 0$ and $\nabla h(0) = 0$ for some open $B_r(0) \subseteq \mathcal{V} \subseteq \mathbb{R}^n$ for some $r > 0$. Further, suppose that $h$ is additively separable as $h(z) = \overline{h}(z_1) + \overline{h}(z_2)$ and $\nabla f(0) = \binom{q}{-q}$ for some $q \in \mathbb{R}^n$. Additionally, let $\eta \in (0, r]$ and $C, c, \Delta > 0$ be such that (27), (29) and (30) hold, where this time (27) applies also for $h = f$. Then for $\sigma \in (0, \overline{\sigma})$ with $\overline{\sigma} = \min\{\eta^2, (\log(2)/(2C))^2\}$ it holds*

$$\left|e^{\frac{1}{\sigma^2}h(0)}\frac{\sqrt{\det\Sigma}}{Z_\sigma}\int_{\mathcal{V}\times\mathcal{V}}f(z)e^{-\frac{1}{\sigma^2}h(z)}\,dz - \sigma^2 A(f, h)\right| \le E_2(\sigma; f, h, r),\tag{35}$$

*where $\Sigma = \nabla^2 h(0)$ with $E_2(\sigma; f, h, r) = O(\sigma^3|\log(\sigma)|^3)$ for $\sigma \to 0$. Here the second order coefficient $A_2(f, h)$ is given by*

$$A_2(f, h) = \frac{1}{2}\operatorname{tr}(\Sigma^{-1}\nabla^2 f(0)).\tag{36}$$

*and $E_2(\sigma; f, h, r)$ is given in (37).*

*Proof.* The proof is similar to the proof of Theorem 15 with the modification for the product space $\mathbb{R}^k = \mathbb{R}^n \times \mathbb{R}^n$. Again we can assume $h(0) = 0$. Let $\alpha : [0, \infty) \to [0, \infty)$ be any function and denote $\widehat{B}(\sigma) = B_{\alpha(\sigma)}(0) \times B_{\alpha(\sigma)}(0) \subseteq \mathbb{R}^n \times \mathbb{R}^n$ and $\widehat{\mathcal{V}} = \mathcal{V} \times \mathcal{V}$. Write

$$
\begin{aligned}
& \frac{1}{\sigma^2}\frac{1}{Z_\sigma^\Sigma}\int_{\widehat{\mathcal{V}}}f(z)e^{-\frac{1}{\sigma^2}h(z)}\,dz \\
& = \frac{1}{\sigma^2}\frac{1}{Z_\sigma^\Sigma}\int_{\widehat{\mathcal{V}}\backslash\widehat{B}(\sigma)}f(z)e^{-\frac{1}{\sigma^2}h(z)}\,dz + \frac{1}{\sigma^2}\frac{1}{Z_\sigma^\Sigma}\int_{\widehat{\mathcal{V}}\cap\widehat{B}(\sigma)}f(z)e^{-\frac{1}{\sigma^2}h(z)}\,dz.
\end{aligned}
$$

We will carefully craft $\alpha$ such that $\lim_{\sigma \to 0} \alpha(\sigma) = 0$ and the first and second integral converge to $0$ and $\frac{1}{2} \operatorname{tr}(\Sigma^{-1} \nabla^2 f(0))$, respectively, with explicit error bounds. For the first one we have

$$\left| \frac{1}{\sigma^2} \frac{1}{Z_\sigma^\Sigma} \int_{\widehat{\mathcal{V}} \setminus \widehat{B}(\sigma)} f(z) e^{-\frac{1}{\sigma^2} h(z)} \, \mathrm{d}z \right| \leq \frac{e^{-\frac{1}{\sigma^2} \gamma_h(\alpha(\sigma))}}{Z_\sigma^\Sigma \sigma^2} \|f\|_{L^1(\mathcal{V} \times \mathcal{V})} \, .$$

Now consider the second integral. By a substitution $z = \sigma w$ and abbreviating $\widetilde{B}_{\widehat{\mathcal{V}}}(\sigma) = (\widehat{\mathcal{V}} \cap \widehat{B}(\sigma))/\sigma = \widetilde{B}_{\mathcal{V}}(\sigma) \times \widetilde{B}_{\mathcal{V}}(\sigma)$ with $\widetilde{B}_{\mathcal{V}}(\sigma) = (\mathcal{V} \cap B_{\alpha(\sigma)}(0))/\sigma$ as well as $\chi_g^\sigma(w) = \frac{1}{\sigma^2} \chi_g(\sigma w)$ for $g \in \{f, h\}$, we have

$$\frac{1}{\sigma^2} \frac{1}{Z_\sigma^\Sigma} \int_{\widehat{\mathcal{V}} \cap \widehat{B}(\sigma)} f(z) e^{-\frac{1}{\sigma^2} h(z)} \, \mathrm{d}z$$

$$= \frac{1}{Z_1^\Sigma} \int_{\widetilde{B}_{\widehat{\mathcal{V}}}(\sigma)} \left( \frac{1}{\sigma} \nabla f(0)^\top w + \frac{1}{2} w^\top \nabla^2 f(0) w + \chi_f^\sigma(w) \right) e^{-\frac{1}{\sigma^2} h(\sigma w)} \, \mathrm{d}w$$

$$=: H(\sigma) + I(\sigma) + J(\sigma) \, .$$

Note that $H(\sigma)$ vanishes due to separability of $h$ and the particular form of $\nabla f(0)$:

$$\sigma Z_1^\Sigma H(\sigma) = \int_{\widetilde{B}_{\widehat{\mathcal{V}}}(\sigma)} \nabla f(0)^\top w \cdot e^{-\frac{1}{\sigma^2} h(\sigma w)} \, \mathrm{d}w$$

$$= \int_{\widetilde{B}_{\mathcal{V}}(\sigma)} \int_{\widetilde{B}_{\mathcal{V}}(\sigma)} (q^\top w_1 - q^\top w_2) e^{-\frac{1}{\sigma^2} \overline{h}(\sigma w_1)} e^{-\frac{1}{\sigma^2} \overline{h}(\sigma w_2)} \, \mathrm{d}w_1 \, \mathrm{d}w_2$$

$$= 0 \, .$$

Next, let us consider the difference between $I(\sigma)$ and $\frac{1}{2} \operatorname{tr}(\Sigma^{-1} \nabla^2 f(0))$ given by

$$\left| \frac{1}{Z_1^\Sigma} \int_{\widetilde{B}_{\widehat{\mathcal{V}}}(\sigma)} \frac{1}{2} w^\top \nabla^2 f(0) w \cdot e^{-\frac{1}{2} w^\top \Sigma w} e^{-\chi_h^\sigma(w)} \, \mathrm{d}w - \frac{1}{2} \operatorname{tr}(\Sigma^{-1} \nabla^2 f(0)) \right|$$

$$\leq \left| \frac{1}{Z_1^\Sigma} \int_{\widetilde{B}_{\widehat{\mathcal{V}}}(\sigma)} \frac{1}{2} w^\top \nabla^2 f(0) w \cdot e^{-\frac{1}{2} w^\top \Sigma w} (e^{-\chi_h^\sigma(w)} - 1) \, \mathrm{d}w \right|$$

$$+ \left| \frac{1}{Z_1^\Sigma} \int_{\mathbb{R}^{2n} \setminus \widetilde{B}_{\widehat{\mathcal{V}}}(\sigma)} \frac{1}{2} w^\top \nabla^2 f(0) w \cdot e^{-\frac{1}{2} w^\top \Sigma w} \, \mathrm{d}w \right|$$

$$=: |I_1(\sigma)| + |I_2(\sigma)| \, .$$

Then

$$|I_1(\sigma)| \leq \left( \sup_{w \in \widetilde{B}_{\widehat{\mathcal{V}}}(\sigma)} |e^{-\chi_h^\sigma(w)} - 1| \right) \frac{1}{Z_1^\Sigma} \int_{\mathbb{R}^{2n}} \frac{1}{2} |w^\top \nabla^2 f(0) w| \cdot e^{-\frac{1}{2} w^\top \Sigma w} \, \mathrm{d}w$$

$$\leq \frac{n}{2} \|\Sigma^{-1/2} \nabla^2 f(0) \Sigma^{-1/2}\| \left( \sup_{w \in \widetilde{B}_{\widehat{\mathcal{V}}}(\sigma)} |e^{-\chi_h^\sigma(w)} - 1| \right)$$

Furthermore, by a variable substitution $u = \Sigma^{1/2} w$ we obtain

$$|I_2(\sigma)| \leq \left| \frac{1}{Z_1} \int_{\mathbb{R}^{2n} \setminus \Sigma^{1/2} \widetilde{B}_{\widehat{\mathcal{V}}}(\sigma)} \frac{1}{2} u^\top \Sigma^{-1/2} \nabla^2 f(0) \Sigma^{-1/2} u \cdot e^{-\frac{1}{2} \|u\|^2} \, \mathrm{d}u \right|$$

$$\leq \frac{1}{2} L(\sigma) \|\Sigma^{-1/2} \nabla^2 f(0) \Sigma^{-1/2}\| \, ,$$

where

$$L(\sigma) = \frac{1}{Z_1} \int_{\mathbb{R}^k \setminus \overline{\Sigma}^{1/2} \widetilde{B}_{\mathcal{V}}(\sigma)} \|u\|^2 e^{-\frac{1}{2} \|u\|^2} \, \mathrm{d}u \, .$$

Noting that for $\overline{\Sigma} = \nabla^2 h(0)$ it holds $\Sigma^{1/2} \widetilde{B}_{\widehat{\mathcal{V}}}(\sigma) = \overline{\Sigma}^{1/2} \widetilde{B}_{\mathcal{V}}(\sigma) \times \overline{\Sigma}^{1/2} \widetilde{B}_{\mathcal{V}}(\sigma)$ and

$$\overline{\Sigma}^{1/2} \widetilde{B}_{\mathcal{V}}(\sigma) = \overline{\Sigma}^{1/2}(\mathcal{V} \cap B_{\alpha(\sigma)}(0))/\sigma \supseteq B_{\lambda_{\min}(\Sigma) \min\{r, \alpha(\sigma)\}/\sigma}(0)$$

and hence

$$L(\sigma) \leq 2\, \mathrm{G}_2\left(\lambda_{\min}(\Sigma) \frac{\min\{r, \alpha(\sigma)\}}{\sigma}\right),$$

with the Gaussian second moment tail

$$\mathrm{G}_2(R) := \frac{1}{(2\pi)^{n/2}} \int_{\mathbb{R}^n \setminus B_R(0)} \|u\|^2 e^{-\frac{1}{2}\|u\|^2}\, \mathrm{d}u.$$

Now consider the final expression

$$|J(\sigma)| = \left| \frac{1}{Z_1^{\Sigma}} \int_{\widetilde{B}_{\widehat{\mathcal{V}}}(\sigma)} \chi_f^{\sigma}(w) e^{-h^{\sigma}(w)}\, \mathrm{d}w \right|$$

$$\leq \left( \sup_{w \in \widetilde{B}_{\widehat{\mathcal{V}}}(\sigma)} |\chi_f^{\sigma}(w)| \right) \frac{1}{Z_1^{\Sigma}} \int_{\widetilde{B}_{\widehat{\mathcal{V}}}(\sigma)} e^{-h^{\sigma}(w)}\, \mathrm{d}w$$

Let us further estimate

$$\frac{1}{Z_1^{\Sigma}} \int_{\widetilde{B}_{\widehat{\mathcal{V}}}(\sigma)} e^{-h^{\sigma}(w)}\, \mathrm{d}w = \frac{1}{Z_1^{\Sigma}} \int_{\widetilde{B}_{\widehat{\mathcal{V}}}(\sigma)} e^{-\frac{1}{2}w^{\top}\Sigma w} e^{-\chi_h^{\sigma}(w)}\, \mathrm{d}w$$

$$\leq \left( \sup_{w \in \widetilde{B}_{\widehat{\mathcal{V}}}(\sigma)} |e^{-\chi_h^{\sigma}(w)} - 1| \right) + 1.$$

Hence we see that we need to pick $\alpha$ in such a way that for $\sigma \to 0$

(i) $(Z_{\sigma}^{\Sigma})^{-1} \sigma^{-2} \exp(-\gamma_h(\alpha(\sigma))/\sigma^2) \to 0$.

(ii) $\mathrm{G}_2\left(\lambda_{\min}(\Sigma) \frac{\min\{r, \alpha(\sigma)\}}{\sigma}\right) \to 0$

(iii) $\sup_{w \in \widetilde{B}_{\widehat{\mathcal{V}}}(\sigma)} |\chi_h^{\sigma}(w)| \to 0$

(iv) $\sup_{w \in \widetilde{B}_{\widehat{\mathcal{V}}}(\sigma)} |\chi_f^{\sigma}(w)| \to 0$

Again take $\alpha(\sigma) = \sigma\beta(\sigma)$ for $\beta(\sigma) = |\log(\sigma)|$. This yields

$$\exp\left(-\frac{1}{\sigma^2}\gamma_h(\alpha(\sigma))\right) \leq \exp\left(-\min\{c\log(\sigma)^2, \Delta\sigma^{-2}\}\right),$$

$$\mathrm{G}_2\left(\lambda_{\min}(\Sigma)\frac{\min\{r, \alpha(\sigma)\}}{\sigma}\right) \leq \mathrm{G}_2\left(\lambda_{\min}(\Sigma)\min\{r\sigma^{-1}, |\log(\sigma)|\}\right),$$

$$\sup_{w \in B_{\mathcal{V}}(\sigma)} \left|e^{-\chi_h^{\sigma}(w)} - 1\right| \leq 2 \sup_{w \in B_{\mathcal{V}}(\sigma)} |\chi_h^{\sigma}(w)| \leq 2C\sigma\beta(\sigma)^3 = 2C|\log(\sigma)|^3,$$

$$\sup_{w \in B_{\mathcal{V}}(\sigma)} \left|\chi_f^{\sigma}(w)\right| \leq C\sigma\beta(\sigma)^3 = C|\log(\sigma)|^3,$$

when $0 < \sigma \leq \overline{\sigma} := \min\{\eta^2, (\log(2)/(2C))^2\}$. Combining all these estimates one obtains, as in the proof of Theorem 15, that (35) holds with

$$\frac{1}{\sigma^2} E(\sigma) = \frac{\sqrt{\det \Sigma}}{Z_{\sigma}\sigma^2} \exp\left(-\min\{c\log(\sigma)^2, \Delta\sigma^{-2}\}\right) \|f\|_{L^1(\mathcal{V} \times \mathcal{V})}$$

$$+ \|\Sigma^{-1/2}\nabla^2 f(0)\Sigma^{-1/2}\| \left(nC\sigma|\log(\sigma)|^3 + \mathrm{G}_2\left(\lambda_{\min}(\Sigma)\min\{r\sigma^{-1}, |\log(\sigma)|\}\right)\right)$$

$$+ C\sigma|\log(\sigma)|^3(1 + 2C\sigma|\log(\sigma)|^3)$$

$$\stackrel{\sigma \to 0}{=} O(\sigma|\log(\sigma)|^3) = \widetilde{O}(\sigma).$$

$$(37)$$

$\square$

### E.2 LOCAL VERSION OF THEOREM 1

First we prove a local version of Theorem 1.

**Theorem 17.** *Assume that there exists an subset $\mathcal{U} \subseteq \mathcal{M}$ that is $C^3$-diffeomorphic to an open subset of $\mathcal{V} \subseteq \mathbb{R}^k$ with $B_r(0) \subseteq \mathcal{V}$ via $\psi : \mathcal{V} \to \mathcal{U}$ for some $0 \le k \le d$ and that $\mu \ll \mathrm{Vol}_{\mathcal{U}}$, where $\mathrm{Vol}_{\mathcal{U}}$ is the volume measure on $\mathcal{U}$. Moreover, assume that $\mu(\cdot) = \frac{\mathrm{d}\mu}{\mathrm{d}\,\mathrm{Vol}_{\mathcal{U}}} \in C^3(\mathcal{U})$. Let $x \in \mathbb{R}^d$ be any point with $\psi(0) =: p = \arg\min_{p \in \mathcal{M}} \|x - p\|$ and $\delta = \inf_{y \in \mathcal{M} \setminus \mathcal{U}} \|x - y\|^2 - \|x - p\|^2 > 0$, and such that*

$$\Sigma = \psi'(0)^\top \psi'(0) + \sum_{i=1}^d (p - x)_i \nabla^2 \psi_i(0) > 0 \,. \tag{38}$$

*Then there exists some $\overline{\sigma} > 0$ such that for all $\sigma \in (0, \overline{\sigma})$*

*(i) it holds that*

$$\|\mathbb{E}\nu_{x,\sigma} - p\| \le \frac{2\big(E_1(\sigma; f_0, h, r)\|p - x\| + \sqrt{d}E_1(\sigma; f_1, h, r) + (\|p - x\| + \|\mu\|_1)\Upsilon(\sigma; \Sigma)\big)}{\mu(p)}$$

*(ii) it holds for $P_0 = \psi'(0)\Sigma^{-1}\psi'(0)^\top$ that*

$$\left\| \frac{1}{\sigma^2} \mathrm{Cov}(\nu_{x,\sigma}) - P_0 \right\|$$

$$\le \frac{4d}{\mu(p)^2} \frac{E_2(\sigma; \overline{f}, \overline{h}, r)}{\sigma^2}$$

$$+ \left( \frac{4(\|\mu\|_2 + \|\mu\|_1^2)}{\mu(p)^2} \frac{\Upsilon(\sigma; \Sigma)^2}{\sigma^2} + 12 \frac{E_1(\sigma; f_0, h, r) + \Upsilon(\sigma; \Sigma)}{\mu(p)} \right) \|P_0\|$$

*with $h$, $f_0$ and $f_1$ given in (39), $\overline{f}_2$ and $\overline{h}$ in (40), $\|\mu\|_i = \mathbb{E}_{y \sim \mu}\|y - x\|^i$, $\Upsilon(\sigma; \Sigma) = \frac{\sqrt{\det \Sigma}}{Z_\sigma} \exp(-\frac{1}{2\sigma^2}\delta)$,*

$$E_1(\sigma; f_1, h, r) = \max_{l=1,\dots,d} E_1(\sigma; f_1^l, h, r) \,,$$

$$E_2(\sigma; \overline{f}_2, \overline{h}, r) = \max_{l,j=1,\dots,d} E_2(\sigma; \overline{f}_2^{jl}, \overline{h}, r) \,,$$

*with $E_1(\sigma; f_1^l, h, r)$ given in (34) and $E_2(\sigma; \overline{f}_2^{jl}, \overline{h}, r)$ in (37), respectively.*

**Remark 18.** *This theorem only requires $\mathcal{M}$ to be locally a manifold, namely at $\mathcal{U}$.*

*Proof.* We denote by $\lambda = \psi^{-1} \# \mu$ be corresponding pullback measure on $\mathcal{V}$ and denote its positive density again by $\lambda \in C^3(\mathcal{V})$. Note that $\lambda(0) = \mu(p)$ by (23). Moreover, let $v = x - p$ and $P_0 = \psi'(0)\Sigma^{-1}\psi'(0)^\top$ for the rest of the proof. First we need to investigate the non-asymptotic convergence of following two integrals:

$$p_\sigma(x) = \int_{\mathcal{M}} \varphi_\sigma(x - y) \, \mathrm{d}\mu(y)$$

$$= \underbrace{\int_{\mathcal{V}} \lambda(z) \frac{1}{Z_\sigma} e^{-\frac{1}{2\sigma^2}\|x - \psi(z)\|^2} \, \mathrm{d}z}_{S_0} + \underbrace{\int_{\mathcal{M} \setminus \mathcal{U}} \varphi_\sigma(x - y) \, \mathrm{d}\mu(y)}_{R_0}$$

and

$$p_\sigma(x)(\mathbb{E}\nu_{x,\sigma} - x) = \int_{\mathcal{M}} (y - x)\varphi_\sigma(x - y) \, \mathrm{d}\mu(y)$$

$$= \underbrace{\int_{\mathcal{V}} \lambda(z)(\psi(z) - x) \frac{1}{Z_\sigma} e^{-\frac{1}{2\sigma^2}\|x - \psi(z)\|^2} \, \mathrm{d}z}_{S_1} + \underbrace{\int_{\mathcal{M} \setminus \mathcal{U}} (y - x)\varphi_\sigma(x - y) \, \mathrm{d}\mu(y)}_{R_1}$$

We can write each $S_i$ and $R_i$ as

$$S_i = \frac{1}{Z_\sigma} \int_\mathcal{V} f_i(z) e^{-\frac{1}{\sigma^2} h(z)} \, \mathrm{d}z \,, \quad R_i = \int_{\mathcal{M} \setminus \mathcal{U}} g_i(y) \varphi_\sigma(x - y) \, \mathrm{d}\mu(y) \,,$$

where

$$h(z) = \frac{1}{2} \|x - \psi(z)\|^2 \,, \quad f_0(z) = \lambda(z) \,, \quad f_1(z) = \lambda(z)(\psi(z) - x) \,, \tag{39}$$

as well as $g_0(y) = 1$ and $g_1(y) = y - x$. Then $f_0, f_1 \in C^3(\mathcal{V})$ and

$$h(0) = \frac{1}{2} \|v\|^2 \,, \quad \nabla h(0) = 0 \,, \quad \nabla^2 h(0) = \Sigma \,,$$

where the second equality is due to $p = \psi(0)$ being the closest point from $\mathcal{U}$ to $x$. Also obviously $f_0(0) = \lambda(0)$ and $f_1(0) = \lambda(0)(p - x)$. Applying Theorem 15 component-wise, we obtain for $\sigma \in (0, \overline{\sigma}_i)$, with $\overline{\sigma}_i = \overline{\sigma}_i(\eta, C)$ given as in Theorem 15 with $\eta, C > 0$ dependent on $f_i$ and $h$, the estimate

$$\underbrace{\left\| e^{\frac{1}{2\sigma^2} \|v\|^2} \sqrt{\det \Sigma} S_i - f_i(0) \right\|}_{=: \|F_i\|} \le D_i(\sigma) \,,$$

with

$$D_0(\sigma) = E_1(\sigma; f_0, h, r) \,, \quad D_1(\sigma) = \sqrt{d} E_1(\sigma; f_1, h, r) := \sqrt{d} \max_{l=1,\ldots,d} E_1(\sigma; f_1^l, h, r) \,,$$

and where $E_1$ is given in Theorem 15 and $f_1^l$ is the $l$-th component of $f_1$. Furthermore, we can estimate for $i = 0, 1$

$$\|R_i\| = \left\| \int_{\mathcal{M} \setminus \mathcal{U}} g_i(y) \varphi_\sigma(x - y) \, \mathrm{d}\mu(y) \right\|$$

$$= \frac{1}{Z_\sigma} \left\| \int_{\mathcal{M} \setminus \mathcal{U}} g_i(y) e^{\frac{1}{2\sigma^2} \|x - y\|^2} \, \mathrm{d}\mu(y) \right\|$$

$$\le e^{-\frac{1}{2\sigma^2} \|v\|^2} \frac{e^{-\frac{1}{2\sigma^2} \delta}}{Z_\sigma} \int_{\mathcal{M} \setminus \mathcal{U}} \|g_i(y)\| \, \mathrm{d}\mu(y) \,,$$

where we have used that $\|x - y\|^2 \ge \delta + \|v\|^2$ for all $y \in \mathcal{M} \setminus \mathcal{U}$. Thus we get the estimate

$$\left\| e^{\frac{1}{2\sigma^2} \|v\|^2} R_i \right\| \le \frac{e^{-\frac{1}{2\sigma^2} \delta}}{Z_\sigma} \|\mu\|_i \,,$$

where the quantity $\|\mu\|_i = \int_\mathcal{M} \|y - x\|^i \, \mathrm{d}\mu(y)$ denotes the $i$-th centered moment of $\mu$. Let us denote $J = e^{\frac{1}{2\sigma^2} \|v\|^2} \sqrt{\det \Sigma}$ and express

$$S_i = f_i(0)/J + F_i/J$$

Now we can estimate the distance to $p$ by

$$\mathbb{E}\nu_{x,\sigma} - p = \frac{S_1 + R_1}{S_0 + R_0} - (p - x) = \frac{f_1(0)/J + F_1/J + R_1}{f_0(0)/J + F_0/J + R_0} - (p - x)$$

$$= \frac{f_1(0) + F_1 + JR_1}{f_0(0) + F_0 + JR_0} - (p - x) = \frac{(p - x) + T_1}{1 + T_0} - (p - x) = \frac{T_1 - T_0(p - x)}{1 + T_0} \,.$$

where $T_i = (F_i + JR_i)/\lambda(0)$. Let us bound

$$\|T_i\| \le \frac{1}{\lambda(0)} \left( D_i(\sigma) + \sqrt{\det \Sigma} \frac{e^{-\frac{1}{2\sigma^2} \delta}}{Z_\sigma} \right) \|\mu\|_i \xrightarrow{\sigma \to 0} 0 \,.$$

Then $|T_0| < 1/2$ if $\sigma < \overline{\sigma}(f_0, h, r) := \min\{\overline{\sigma}_0, \widetilde{\sigma}\}$ with $\widetilde{\sigma}$ depending on $D_0$, $\Sigma$ and $\delta$. For this $\sigma \in (0, \overline{\sigma})$ we obtain

$$
\begin{aligned}
&\|\mathbb{E}\nu_{x,\sigma} - p\| \\
&\quad \le 2|T_0|\|p - x\| + 2\|T_1\| \\
&\quad \le \frac{2}{\lambda(0)}\left(E_1(\sigma; f_0, h, r)\|p - x\| + \sqrt{d}E_1(\sigma; f_1, h, r) + (\|p - x\| + \|\mu\|_1)\sqrt{\det \Sigma}\frac{e^{-\frac{1}{2\sigma^2}\delta}}{Z_\sigma}\right).
\end{aligned}
$$

This proves (i). The proof of (ii) is analogous. For $(z, \widetilde{z}) \in \mathcal{V} \times \mathcal{V}$ define

$$
\begin{aligned}
\overline{f}_2(z, \widetilde{z}) &= \lambda(z)\lambda(\widetilde{z})((\psi(z) - x)(\psi(z) - x)^\top - (\psi(z) - x)(\psi(\widetilde{z}) - x)^\top), \\
\overline{h}(z, \widetilde{z}) &= \frac{1}{2}\|x - \psi(z)\|^2 + \frac{1}{2}\|x - \psi(\widetilde{z})\|^2.
\end{aligned}
\tag{40}
$$

Then $f, h \in C^3(\mathcal{V} \times \mathcal{V})$, $\overline{f}_2(0,0) = 0$ and

$$
\overline{h}(0,0) = \|v\|^2, \quad \nabla\overline{h}(0,0) = 0, \quad \nabla^2\overline{h}(0,0) = \overline{\Sigma} := \operatorname{diag}(\Sigma, \Sigma).
$$

Let $\overline{f}_2^{jl}$ be the $(j, l)$-th entry of $\overline{f}_2$. Then it is an elementary, but very tedious exercise to show that $\overline{f}$ and $\overline{h}$ satisfy the conditions of Theorem 16 and that moreover the second order coefficient is given by

$$
A_2(\overline{f}_2^{jl}, \overline{h}) = \lambda(0)^2(\psi'(0)\Sigma^{-1}\psi'(0)^\top)_{lj} = \lambda(0)^2(P_0)_{lj},
$$

i.e. $\overline{A}_2 := (A_2(\overline{f}_2^{jl}, \overline{h}))_{l,j=1}^d = \lambda(0)^2 P_0$. Now split

$$
\begin{aligned}
&p_\sigma(x)^2 \operatorname{Cov}(\nu_{x,\sigma}) \\
&\quad = p_\sigma(x)^2(\operatorname{Cov}(\nu_{x,\sigma}) - \mathbb{E}\nu_{x,\sigma}(\mathbb{E}\nu_{x,\sigma})^\top) \\
&\quad = \int_{\mathcal{M} \times \mathcal{M}} ((y - x)(y - x)^\top - (y - x)(\widetilde{y} - x)^\top)\varphi_\sigma(x - y)\varphi_\sigma(x - \widetilde{y})\,\mathrm{d}(\mu \otimes \mu)(y, \widetilde{y}) \\
&\quad = \underbrace{\int_{\mathcal{V} \times \mathcal{V}} \overline{f}_2(z, \widetilde{z})\frac{1}{Z_\sigma^2}e^{-\frac{1}{2\sigma^2}(\|x - \psi(z)\|^2 + \|x - \psi(\widetilde{z})\|^2)}\,\mathrm{d}(z, \widetilde{z})}_{\overline{S}_2} \\
&\qquad + \underbrace{\int_{\mathcal{M} \times \mathcal{M} \setminus \mathcal{U} \times \mathcal{U}} ((y - x)(y - x)^\top - (y - x)(\widetilde{y} - x)^\top)\varphi_\sigma(x - y)\varphi_\sigma(x - \widetilde{y})\,\mathrm{d}(\mu \otimes \mu)(y, \widetilde{y})}_{\overline{R}_2}.
\end{aligned}
$$

Applying Theorem 16 component-wise to $\overline{f}_2$ we obtain for $\sigma \in (0, \overline{\sigma}_2)$, with $\overline{\sigma}_2 = \overline{\sigma}_2(\eta, C)$ and $\eta, C > 0$ dependent on $\overline{f}_2$ and $\overline{h}$, that

$$
\underbrace{\left\|(e^{\frac{1}{2\sigma^2}\|v\|^2}\sqrt{\det \Sigma})^2\overline{S}_2 - \sigma^2\overline{A}_2\right\|}_{\|\overline{G}_2\|} \le dE_2(\sigma; \overline{f}_2, \overline{h}, r) := d\max_{l,j=1,\dots,d} E_2(\sigma; \overline{f}_2^{jl}, \overline{h}, r),
$$

with $E_2(\sigma; \overline{f}_2^{jl}, \overline{h}, r)$ given in Theorem 37. The term $\overline{R}_2$ on the other hand can be estimated again by

$$
\begin{aligned}
\|\overline{R}_2\| &\le e^{-\frac{1}{\sigma^2}\|v\|^2}\frac{e^{-\frac{1}{\sigma^2}\delta}}{Z_\sigma^2}\int_{\mathcal{M} \times \mathcal{M} \setminus \mathcal{U} \times \mathcal{U}} \|(y - x)(y - x)^\top - (y - x)(\widetilde{y} - x)^\top\|\,\mathrm{d}(\mu \otimes \mu)(y, \widetilde{y}) \\
&\le e^{-\frac{1}{\sigma^2}\|v\|^2}\frac{e^{-\frac{1}{\sigma^2}\delta}}{Z_\sigma^2}(\|\mu\|_2 + \|\mu\|_1^2).
\end{aligned}
$$

For $\overline{S}_2$ it holds

$$
\overline{S}_2 = \sigma^2\lambda(0)^2 P_0(x)/J^2 + \overline{G}_2/J^2
$$

and thus

$$\begin{aligned}
\frac{1}{\sigma^2} \operatorname{Cov}(\nu_{x,\sigma}) - P_0 &= \frac{\overline{S}_2/\sigma^2 + \overline{R}_2/\sigma^2}{(S_0 + R_0)^2} - P_0 \\
&= \frac{\lambda(0)^2 P_0/J^2 + \overline{G}_2/(\sigma^2 J^2) + \overline{R}_2/\sigma^2}{(\lambda(0)/J + F_0/J + R_0)^2} - P_0 \\
&= \frac{P_0(x) + \overline{T}_2}{(1 + T_0)^2} - P_0 \\
&= \frac{\overline{T}_2 - T_0(2 + T_0)P_0}{(1 + T_0)^2}
\end{aligned}$$

with $\overline{T}_2 = (\overline{G}_2 + J^2 \overline{R}_2)/(\lambda(0)^2 \sigma^2)$. Again we have the estimate

$$\|\overline{T}_2\| \le \frac{d}{\lambda(0)^2} \frac{E_2(\sigma)}{\sigma^2} + \frac{(\det \Sigma)(\|\mu\|_2 + \|\mu\|_1^2)}{\lambda(0)^2} \frac{e^{-\frac{1}{\sigma^2}\delta}}{Z_\sigma^2 \sigma^2},$$

and for $\sigma \in (0, \overline{\sigma})$ (with $\overline{\sigma}$ as before, guaranteeing $|T_0| < 1/2$)

$$\left\| \frac{1}{\sigma^2} \operatorname{Cov}(\nu_{x,\sigma}) - P_0 \right\|$$
$$\le 4\|\overline{T}_2\| + 12|T_0|\|P_0\|$$
$$\le \frac{4d}{\lambda(0)^2} \frac{E_2(\sigma)}{\sigma^2} + \left( \frac{4(\det \Sigma)(\|\mu\|_2 + \|\mu\|_1^2)}{\lambda(0)^2} \frac{e^{-\frac{1}{\sigma^2}\delta}}{Z_\sigma^2 \sigma^2} + 12 \frac{D_0(\sigma) + \sqrt{\det \Sigma} \frac{e^{-\frac{1}{2\sigma^2}\delta}}{Z_\sigma}}{\lambda(0)} \right) \|P_0\|$$

$\square$

## E.3 BOUNDS IN TERMS OF DISTRIBUTION AND MANIFOLD PARAMETERS

In this section we bound the constants appearing in Theorem 17 in terms of parameters of the distribution $\mu$ and the manifold $\mathcal{M} \in \mathcal{M}_k(\tau, M)$, when the chart is given by the graph chart $\psi = \psi_p : \mathcal{V} \to \mathcal{M}$ with $\mathcal{V} := B_{\min\{\tau_{\mathcal{M}}, M\}/4}^{\mathrm{T}_p \mathcal{M}}(0)$, $x \in \mathcal{T}(\tau)$, $\tau \in (0, \tau_{\mathcal{M}})$ and $p = \pi(x)$. According to the proof of Theorem 17 need to consider the following maps

$$h : \mathcal{V} \to \mathbb{R} : z \mapsto \frac{1}{2} \|x - \psi_p(z)\|^2, \tag{41}$$

as well as

$$f_0 : \mathcal{V} \to \mathbb{R} : z \mapsto \lambda(z), \tag{42}$$
$$f_1 : \mathcal{V} \to \mathbb{R}^d : z \mapsto \lambda(z)(\psi_p(z) - x), \tag{43}$$
$$\overline{f}_2 : \mathcal{V} \times \mathcal{V} \to \mathbb{R}^{d \times d} : (z, \widetilde{z}) \mapsto \lambda(z)\lambda(\widetilde{z})((\psi_p(z) - x)(\psi_p(z) - x)^\top - (\psi_p(\widetilde{z}) - x)^\top)), \tag{44}$$

where $\lambda = \psi_p^{-1} \# \mu = \operatorname{pr}_p \# \mu$.

### E.3.1 CONDITIONS (27) AND (29) FOR $h$

First we show how to express conditions (27) and (29) in terms of the manifold parameters for $h$ given in (41). To see (27), note that for $z \in \mathcal{V}$ we have by the chain and product rule as well as Lemma 9 (i) that

$$\begin{aligned}
\|\nabla^3 h(z)\| &\le 3\|\psi_p''(z)\|\|\psi_p'(z)\| + \|x - \psi_p(z)\|\|\psi_p'''(z)\| \\
&\le 3M^2 + (\|x - p\| + \|p - \psi_p(z)\|)M \\
&\le 3M^2 + (\tau_{\mathcal{M}} + \frac{8}{7}\|z\|)M \\
&\le (3M + \frac{9}{7}\tau_{\mathcal{M}})M.
\end{aligned}$$

Thus (27) holds with, say $C^{(1)}_{\tau_{\mathcal{M}},M} = \frac{1}{2}(M + \tau_{\mathcal{M}})M$, and $\eta = \min\{\tau_{\mathcal{M}}, M\}/4$. Next, for (29) and $\eta = \min\{\tau_{\mathcal{M}}, M\}/4$ we compute for $z \in B_\eta^{\mathrm{T}_p\mathcal{M}}(0)$

$$h(z) - h(0) \geq \inf_{\widetilde{z} \in \mathcal{V} \setminus B_{\|z\|}^{\mathrm{T}_p\mathcal{M}}(0)} h(\widetilde{z}) - h(0) = \inf_{q \in \psi_p(\mathcal{V} \setminus B_{\|z\|}^{\mathrm{T}_p\mathcal{M}}(0))} \frac{1}{2}\|x - q\|^2 - \frac{1}{2}\|x - p\|^2$$

$$\geq \inf_{q \in \mathcal{M} \setminus B_{\|z\|}(p)} \frac{1}{2}\|x - q\|^2 - \frac{1}{2}\|x - p\|^2$$

$$\geq \frac{1}{2}\frac{\tau_{\mathcal{M}} - \tau}{\tau_{\mathcal{M}} + \tau}\|z\|^2.$$

In the first inequality we have used $\psi_p(\mathcal{V} \setminus B_{\|z\|}^{\mathrm{T}_p\mathcal{M}}(0)) \subseteq \mathcal{M} \setminus B_{\|z\|}(p)$, which follows from Lemma 9 (i), whereas in the last inequality we have used Lemma 13 and $x \in \mathcal{T}(\tau)$. Thus (29) holds on $\mathcal{V}$ with $c = \frac{1}{2}\frac{\tau_{\mathcal{M}} - \tau}{\tau_{\mathcal{M}} + \tau}$ and implies (32).

### E.3.2 Lower bound on $\lambda_{\min}(\Sigma)$ and upper bound on $\sqrt{\det \Sigma}$

We first consider $\lambda_{\min}(\Sigma)$ for $\Sigma = \nabla^2 h(0)$ with $h$ given in (41) and when $x \in \mathcal{T}(\tau)$. By Lemma 10 and (20) we directly obtain

$$\lambda_{\min}(\Sigma) = \|\Sigma^{-1}\|^{-1} \geq 1 - \|x - p\|\kappa_{\mathcal{M}} \geq 1 - \frac{\tau}{\tau_{\mathcal{M}}}.$$

Due to (20) and the definition of the shape operator $S_p$, an upper bound on $\sqrt{\det \Sigma}$ is given by

$$\sqrt{\det \Sigma} = \sqrt{\det(I_{\mathrm{T}_p\mathcal{M}} + S_p^{p-x})} \leq \left(1 + \frac{\|p - x\|}{\rho(p, p - x)}\right)^{k/2} \leq \left(1 + \frac{\tau}{\tau_{\mathcal{M}}}\right)^{k/2} \leq 2^{k/2}.$$

### E.3.3 Condition (27) for $\overline{f}_2^{jl}$

An upper bound for $\|\nabla^3 \overline{f}_2^{jl}(z, \widetilde{z})\|$ in terms of $M$ and the third derivatives of $\frac{\mathrm{d}\mu}{\mathrm{d}\mathrm{Vol}_{\mathcal{M}}}$, where

$$\overline{f}_2^{jl}(z, \widetilde{z}) = \lambda(z)\lambda(\widetilde{z})((\psi_p^j(z) - x)(\psi_p^l(z) - x) - (\psi_p^j(z) - x)(\psi_p^l(\widetilde{z}) - x)),$$

can be obtained by direct differentiation. We don't pursue the complete derivation here and instead say that (27) holds with, say, $C^{(2)}_{\tau_{\mathcal{M}},M,\mu}$.

### E.3.4 Upper bound on $\|f_0\|_{L^1(\mathcal{V})}$, $\|f_1^j\|_{L^1(\mathcal{V})}$ and $\|\overline{f}_2^{jl}\|_{L^1(\mathcal{V} \times \mathcal{V})}$ and $\|\nabla^2 \overline{f}_2^{jl}(0)\|$

We clearly have $\|f_0\|_{L^1(\mathcal{V})} = \lambda(\mathcal{V}) \leq 1$. Next, due to $\|y - x\| \leq \mathrm{diam}(\mathcal{M}) + \|p - x\| \leq \mathrm{diam}(\mathcal{M}) + \tau$ for $y \in \mathcal{M}$, we have

$$\|f_1^j\|_{L^1(\mathcal{V})} \leq \|f_1\|_{L^1(\mathcal{V};\mathbb{R}^d)} = \int_{\mathcal{U}} \|y - x\| \, \mathrm{d}\mu(y) = \|\mu\|_1 \leq \mathrm{diam}(\mathcal{M}) + \tau,$$

and

$$\|\overline{f}_2^{jl}\|_{L^1(\mathcal{V} \times \mathcal{V})} \leq \|\overline{f}_2\|_{L^1(\mathcal{V} \times \mathcal{V};\mathbb{R}^{d \times d})}$$

$$= \int_{\mathcal{U} \times \mathcal{U}} \|(y - x)(y - x)^\top - (y - x)(\widetilde{y} - x)^\top\| \, \mathrm{d}(\mu \otimes \mu)(y, \widetilde{y})$$

$$\leq \|\mu\|_2 + \|\mu\|_1^2$$

$$\leq 2(\mathrm{diam}(\mathcal{M}) + \tau)^2.$$

Furthermore for $v = \binom{v_1}{v_2} \in \mathbb{R}^k \times \mathbb{R}^k$ we have

$$\overline{f}_2''(0, 0)[v, v] = \lambda(0)^2 \cdot (2v_1 v_1^\top - 2v_1 v_2^\top + (p - x)(\mathbb{I}_p(v_1, v_2) - \mathbb{I}_p(v_2, v_2))^\top)$$

$$+ 2\lambda(0)\nabla\lambda(0)^\top(v_1 + v_2) \cdot (p - x)(v_1 - v_2)^\top.$$

Hence, using (23) and the fact that $\|p - x\|\|\mathbb{I}_p(v_1, v_2)\| \leq \frac{\tau}{\tau_{\mathcal{M}}} \leq 1$ for any unit vectors $v_1, v_2 \in \mathrm{T}_p\mathcal{M}$ we obtain

$$\|\nabla^2 \overline{f}_2^{jl}(0)\| \leq \|\overline{f}_2''(0, 0)\| \leq 6\mu(p)^2 + 8\tau\mu(p)\|\mathrm{grad}_{\mathcal{M}}\mu(p)\|.$$

### E.4 PROOF OF THEOREM 1

We apply Theorem 17 to every point $x \in \mathcal{T}(\tau)$ and $p = \pi(x)$ with the graph chart $\psi = \psi_p$. It remains to provide universal constants for the bounds in Theorem 17 (i) and (ii). First we bound $E_1(\sigma; f_0, h, r, \theta)$ with $E_1$ given in (34). By Section E.3 we can take $r_{\tau_{\mathcal{M}}, M} = \eta = \min\{\tau_{\mathcal{M}}, M\}/4$, $c_{\tau_{\mathcal{M}}, M} = \frac{1}{2} \frac{\tau_{\mathcal{M}} - \tau}{\tau_{\mathcal{M}} + \tau}$ and $C_{\tau_{\mathcal{M}}, M, \mu} = \max\{C^{(1)}_{\tau_{\mathcal{M}}, M}, C^{(2)}_{\tau_{\mathcal{M}}, M, \mu}\}$ and hence, due to $\|f_0\|_{L^1(\mathcal{V})} \le 1$,

$$
\begin{aligned}
E_1(\sigma; f_0) \le{} & \frac{2^{k/2}}{Z_\sigma} \exp\left(-c_{\tau_{\mathcal{M}}, M} \log(\sigma)^2\right) \\
&+ \mu(p)\left(2C_{\tau_{\mathcal{M}}, M, \mu}\sigma|\log(\sigma)|^3 + \mathrm{G}_0((1 - \frac{\tau}{\tau_{\mathcal{M}}})\min\{r_{\tau_{\mathcal{M}}, M}\sigma^{-1}, |\log(\sigma)|\})\right) \\
&+ C_{\tau_{\mathcal{M}}, M, \mu}\sigma|\log(\sigma)|(1 + 2C_{\tau_{\mathcal{M}}, M, \mu}\sigma|\log(\sigma)|^3).
\end{aligned}
$$

Similarly, due to $\|f_1^j\|_{L^1(\mathcal{V})} \le \|\mu\|_1$, we have the bound

$$
\begin{aligned}
E_1(\sigma; f_1) \le{} & \frac{2^{k/2}}{Z_\sigma} \exp\left(-c_{\tau_{\mathcal{M}}, M} \log(\sigma)^2\right)(\mathrm{diam}(\mathcal{M}) + \tau) \\
&+ \mu(p)\left(2C_{\tau_{\mathcal{M}}, M, \mu}\sigma|\log(\sigma)|^3 + \mathrm{G}_0((1 - \frac{\tau}{\tau_{\mathcal{M}}})\min\{r_{\tau_{\mathcal{M}}, M}\sigma^{-1}, |\log(\sigma)|\})\right) \\
&+ C_{\tau_{\mathcal{M}}, M, \mu}\sigma|\log(\sigma)|(1 + 2C_{\tau_{\mathcal{M}}, M, \mu}\sigma|\log(\sigma)|^3).
\end{aligned}
$$

Further, due to $\|\overline{f}_2^{jl}\|_{L^1(\mathcal{V}\times\mathcal{V})} \le \|\mu\|_2 + \|\mu\|_1^2$ and the bound on $\|\nabla^2 \overline{f}_2^{jl}(0)\|$ we have

$$
\begin{aligned}
E_2(\sigma; \overline{f}_2) \le{} & 2\frac{2^{k/2}}{Z_\sigma} \exp(-c_{\tau_{\mathcal{M}}, M} \log(\sigma)^2)(\mathrm{diam}(\mathcal{M}) + \tau)^2 \\
&+ \mu(p)\frac{6\mu(p) + 8\tau\|\mathrm{grad}_{\mathcal{M}}\,\mu(p)\|}{1 - \tau/\tau_{\mathcal{M}}}\left(kC_{\tau_{\mathcal{M}}, M, \mu}\sigma|\log(\sigma)|^3\right. \\
&\qquad\qquad\qquad\qquad\left. + \mathrm{G}_2\left((1 - \frac{\tau}{\tau_{\mathcal{M}}})\min\{r_{\tau_{\mathcal{M}}, M}\sigma^{-1}, |\log(\sigma)|\}\right)\right) \\
&+ C_{\tau_{\mathcal{M}}, M, \mu}\sigma|\log(\sigma)|^3(1 + 2C_{\tau_{\mathcal{M}}, M, \mu}\sigma|\log(\sigma)|^3)
\end{aligned}
$$

Now by Lemma 13 we can pick $\delta = \frac{\tau_{\mathcal{M}} - \tau}{\tau_{\mathcal{M}} + \tau}\eta^2 = 2c_{\tau_{\mathcal{M}}, M}r^2_{\tau_{\mathcal{M}}, M}$ and thus

$$
\Upsilon(\sigma; \Sigma) \le \frac{2^{k/2}}{Z_\sigma} \exp\left(-\frac{c_{\tau_{\mathcal{M}}, M}r^2_{\tau_{\mathcal{M}}, M}}{\sigma^2}\right).
$$

Finally, $\|P_0\| \le (1 - \tau/\tau_{\mathcal{M}})^{-1}$. Note that since $\mu(\cdot) \in C^3(\mathcal{M})$ and $\mathrm{supp}\,\mu = \mathcal{M}$, there exist positive lower and upper bounds on $\mu(\cdot)$ and its derivatives on $\mathcal{M}$. This shows that there $K$ and $\overline{\sigma}$ depending only on $\tau$, $M$ and $\mathrm{diam}(\mathcal{M})$ (as $k$) and $\mu$ such that (5) hold and finishes the proof.

### E.5 A USEFUL COROLLARY TO THEOREM 1

The following elementary consequence of Theorem 1 will be useful in the proofs of the results from Section 5.

**Corollary 19.** *Suppose that for some* $\mathrm{v}: \mathbb{R}^d \to \mathbb{R}^d$ *and* $\tau \in (0, \tau_{\mathcal{M}})$ *and* $\epsilon \le \tau$ *we have*

$$
\|\mathrm{v}(x) - \pi(x)\| < \epsilon \text{ for all } x \in \mathcal{T}(\tau). \tag{45}
$$

*Then* $\mathrm{v}(\mathcal{T}(\tau)) \subseteq \mathcal{T}(\epsilon)$. *Moreover, for* $x \in \mathcal{T}(\epsilon)$ *we have* $\|\mathrm{v}(x) - x\| \le 2\epsilon$ *and for* $x \in \mathcal{T}(\tau)$ *we have* $\|\mathrm{v}(x) - x\| \le 2\tau$.

*Proof.* From (45) and the fact that $\pi(x) \in \mathcal{M}$ we clearly have $\mathrm{v}(x) \in \mathcal{T}(\epsilon)$ for any $x \in \mathcal{T}(\tau)$, i.e. $\mathrm{v}(\mathcal{T}(\tau)) \subseteq \mathcal{T}(\epsilon)$. The last claims follows then from the triangle inequality $\|\mathrm{v}(x) - x\| \le \|\mathrm{v}(x) - \pi(x)\| + \|\pi(x) - x\|$. $\square$

## F APPROXIMATE RIEMANNIAN GRADIENT FLOW WITH LANDING

In this section $C = \|\nabla f|_{\mathcal{T}(\tau_{\mathcal{M}})}\|_\infty$ and $L = \text{Lip}(\nabla f|_{\mathcal{T}(\tau_{\mathcal{M}})})$ denote the supremum of $\nabla f$ and the Lipschitz constant of $\nabla f$ on $\mathcal{T}(\tau_{\mathcal{M}})$ for some manifold $\mathcal{M}$. We will often assume the following approximation condition on a function $\mathrm{v} \in C^1(\mathbb{R}^d; \mathbb{R}^d)$:

$$\|\mathrm{v}(x) - \pi(x)\| < \epsilon, \quad \|\mathrm{v}'(x) - P_0(x)\| < \epsilon \text{ for all } x \in \mathcal{T}(\tau). \tag{46}$$

We will also abbreviate the (negative) right hand side of the dynamics (8) by

$$G_{\mathrm{v}}^\eta(x) = \mathrm{v}'(x)\nabla f(x) + \eta(x - \mathrm{v}(x)) \tag{47}$$

### F.1 STATIONARY POINTS OF (8) AND OPTIMALITY CRITERIA

We need to analyze the meaning of approximate stationary points of (8) for the optimization problem (1).

**Lemma 20.** *Suppose that $\tau \in (0, \tau_{\mathcal{M}})$ and $\sigma > 0$ are such that (46) is satisfied. Moreover, suppose that $\mathrm{v}(\mathcal{T}(\tau)) \subseteq \mathcal{T}(\tau)$. Suppose that $\widetilde{\tau} \in (0, \tau]$, $\delta > 0$ and that $x_* \in \mathcal{T}(\widetilde{\tau})$ is a $\delta$-approximate stationary point of (8), i.e.*

$$\|G_{\mathrm{v}}^\eta(x_*)\| \le \delta. \tag{48}$$

*Then for $p_* = \pi(x_*)$ it holds that*

$$\|\text{grad}_{\mathcal{M}} f(p_*)\| \le 2(L + C + \eta)\epsilon + 2\frac{\widetilde{\tau}/\tau_{\mathcal{M}}}{1 - \widetilde{\tau}/\tau_{\mathcal{M}}}C + \delta. \tag{49}$$

*Proof.* We have the estimates

$$\|\eta(\mathrm{v}(x_*) - x_*) - \eta(\pi(x_*) - x_*)\| \le \eta\epsilon,$$

and

$$\|P_0(\pi(x_*))\nabla f(\pi(x_*)) - \mathrm{v}'(x_*)\nabla f(\mathrm{v}(x_*))\|$$
$$\le \|P_0(\pi(x_*))\nabla f(\pi(x_*)) - P_0(\pi(x_*))\nabla f(\mathrm{v}(x_*))\|$$
$$+ \|P_0(\pi(x_*))\nabla f(\mathrm{v}(x_*)) - P_0(x_*)\nabla f(\mathrm{v}(x_*))\|$$
$$+ \|P_0(x_*)\nabla f(\mathrm{v}(x_*)) - \mathrm{v}'(x_*)\nabla f(\mathrm{v}(x_*))\|$$
$$\le (\text{Lip}(\nabla f) + \|\nabla f|_{\mathcal{T}(\tau)}\|_\infty)\epsilon + \|P_0(\pi(x_*)) - P_0(x_*)\|\|\nabla f|_{\mathcal{T}(\tau)}\|_\infty,$$

where we have used that $P_0(\pi(x_*))$ is an orthogonal projection and hence has unit norm. By Lemma 25 applied to $z_1 = \mathrm{v}'(x_*)\nabla f(\mathrm{v}(x_*))$ and $z_2 = \eta(\mathrm{v}(x_*) - x_*)$, Lemma 11 and the fact that $P_0(\pi(x_*))\nabla f(\pi(x_*)) = \text{grad}_{\mathcal{M}} f(\pi(x_*))$, the inequality (49) follows. $\square$

### F.2 LIE DERIVATIVE OF MANIFOLD DISTANCE

**Lemma 21.** *Suppose that $\tau \in (0, \tau_{\mathcal{M}})$ and $\sigma > 0$ is such that (46) is satisfied. Then it holds*

$$-\langle \nabla \mathrm{d}(x), G_{\mathrm{v}}^\eta(x)\rangle \le -2\eta \mathrm{d}(x) + \epsilon(C + \eta)\sqrt{2\,\mathrm{d}(x)} \text{ for all } x \in \mathcal{T}(\tau).$$

*Proof.* We have for $x \in \mathcal{T}(\tau)$

$$\langle x - \pi(x), -G_{\mathrm{v}}^\eta(x)\rangle$$
$$= -2\eta\,\mathrm{d}(x) + \langle \pi(x) - x, (\mathrm{v}'(x) - P_0(x))\nabla f(\mathrm{v}(x))\rangle$$
$$+ \eta\langle x - \pi(x), \mathrm{v}(x) - \pi(x)\rangle$$
$$\le -2\eta\,\mathrm{d}(x) + \|\pi(x) - x\|\|\mathrm{v}'(x) - P_0(x)\|\|\nabla f(\mathrm{v}(x))\|$$
$$+ \eta\|\pi(x) - x\|\|\mathrm{v}(x) - \pi(x)\|$$
$$\le -2\eta\,\mathrm{d}(x) + (\|\nabla f|_{\mathcal{T}(\tau)}\|_\infty\epsilon + \eta\epsilon)\sqrt{2\,\mathrm{d}(x)}.$$

$\square$

F.3   EXACT LANDING WITH $\sigma = 0$

Here we establish the following noiseless version of Theorem 3.

**Theorem 22.** *Consider the flow (8) for $\sigma = 0$ and $\eta \geq 0$ with $x(0) \in \mathcal{T}(\tau)$ for some $\tau \in (0, \tau_\mathcal{M})$. Then the solution $x(t)$ exists for all times $t \geq 0$ and is contained in $\mathcal{T}(\tau)$. Moreover every accumulation point $x_*$ of this flow satisfies $\mathrm{grad}_\mathcal{M} f(x_*) = 0$ and is at most $\tau$ away from the point $p_* = \pi(x_*) \in \mathcal{M}$ with $\|\mathrm{grad}_\mathcal{M} f(p_*)\| \leq L\tau$. Moreover, if $\eta > 0$, then $x_* = p_* \in \mathcal{M}$ and $\mathrm{grad}_\mathcal{M} f(p_*) = 0$, i.e. every accumulation point is critical.*

*Proof.* Note first that

$$\frac{\mathrm{d}}{\mathrm{d}\,t}\,\mathrm{d}(x) = \langle x - \pi(x), \dot{x}\rangle = -\eta \|x - \pi(x)\|^2 = -2\eta\,\mathrm{d}(x)\,,$$

i.e. $\mathrm{d}(x(t)) = e^{-2\eta t}\,\mathrm{d}(x(0))$ and hence $x(t) \in \mathcal{T}$ for all $t \geq 0$. Moreover, if $\eta > 0$ then $\mathrm{d}(x(t)) \to 0$ for $t \to \infty$. Let us now consider $f \circ \pi$ as a Lyapunov function for (9) with $\sigma = 0$. We observe that

$$\begin{aligned}\frac{\mathrm{d}}{\mathrm{d}\,t} f(\pi(x)) &= -\nabla f(\pi(x))^\top P_0(x)^\top P_0(x) \nabla f(\pi(x)) \\ &= -\left\langle \mathrm{P}_{\mathrm{T}_{\pi(x)}\mathcal{M}} \nabla f(x), H_x^{-2} \mathrm{P}_{\mathrm{T}_{\pi(x)}\mathcal{M}} \nabla f(x)\right\rangle \\ &= -\|\mathrm{grad}\,f(x)\|_{H_x}^2\,,\end{aligned}$$

with $\|v\|_{H_x}^2 = \|H_x^{-1}v\|_{\mathrm{T}_{\pi(x)}\mathcal{M}}^2$. This shows that $f(x(t))$ is non-increasing for $t \to \infty$. Since $\mathcal{T}$ is bounded, $f_* = \lim_{t\to\infty} f(x(t))$ is finite. Moreover, we have

$$\int_0^\infty \|\mathrm{grad}\,f(x(t))\|_{H_{x(t)}}^2\,\mathrm{d}t = f(x(0)) - f_* < \infty\,. \tag{50}$$

Notice that $\{x(t) \mid t \geq 0\} \subseteq \overline{\mathcal{T}}(\mathrm{dist}_\mathcal{M}(x(0)))$ being compact. Since $\mathcal{T} \to \mathbb{R} : z \mapsto \|\mathrm{grad}\,f(z)\|_{H_z}^2$ is continuous, it is uniformly continuous on $\overline{\mathcal{T}}(\mathrm{dist}_\mathcal{M}(x(0)))$. This, together with (50) implies, by Barbalat's lemma Farkas & Wegner (2016), that $\lim_{t\to\infty}\|\mathrm{grad}\,f(x(t))\|_{H_{x(t)}}^2 = 0$. Clearly any accumulation point $x_*$ of $\{x(t) \mid t \geq 0\}$ satisfies $H_{x_*}^{-1}\,\mathrm{grad}\,f(x_*) = 0$, i.e. $\mathrm{grad}\,f(x_*) = 0$. Then $\pi(x_*) \in \mathcal{M}$ is an accumulation point of $\{\pi(x(t)) \mid t \geq 0\} \subseteq \mathcal{M}$ and

$$\|\mathrm{grad}\,f(\pi(x_*))\| = \|\mathrm{grad}\,f(\pi(x_*)) - \mathrm{grad}\,f(x_*)\| \leq L\|\pi(x_*) - x_*\| = L\tau\,.$$

$\square$

F.4   PROOF OF THEOREM 3

Note first that, by Theorem 1, if (7) hold with $\epsilon \to \epsilon'$, then 46 are satisfied with $\epsilon = \epsilon' + K\sigma|\log(\sigma)|^3$ for $\sigma \in (0, \overline{\sigma}(\tau, \mathcal{M}, \mu))$. In the following we will write $\epsilon > 0$ for the latter quantity. First let us analyze the manifold distance of (8). We have by Lemma 21 whenever $x(t) \in \mathcal{T}(\tau)$ that

$$\frac{\mathrm{d}}{\mathrm{d}\,t}\,\mathrm{d}(x) = -\langle \nabla\,\mathrm{d}(x), G_\mathrm{v}^\eta(x)\rangle \leq -2\eta\,\mathrm{d}(x) + \epsilon(C + \eta)\sqrt{2\,\mathrm{d}(x)}\,,$$

where the right hand side is non-positive iff

$$\mathrm{dist}_\mathcal{M}(x) \geq \frac{\epsilon}{2}\left(\frac{C}{\eta} + 1\right) =: \tau_0\,.$$

Note that $\tau_0 < \tau$ if $\epsilon < 2\tau/(1 + C/\eta)$. In particular if $\mathrm{dist}_\mathcal{M}(x(0)) \in [\tau_0, \tau)$, then $\overline{\mathcal{T}}(\mathrm{dist}_\mathcal{M}(x(0)))$ is invariant w.r.t. the flow (8) and $x(t)$ exists for all $t \geq 0$. Moreover, by a similar argument as in the standard proof of Lyapunov's direct method Khalil & Grizzle (2002), for each $\delta > 0$ there exists some $T > 0$ such that for all $t \geq T$ it holds that $x(t) \in \mathcal{T}(\tau_0 + \delta)$. If $\mathrm{v}$ is a gradient field, i.e. $\mathrm{v} = \nabla g$ for some function $g \in C^1(\mathbb{R}^d)$, then $V(x) := f(\mathrm{v}(x)) + \eta(\|x\|^2/2 - g(x))$ satisfies $\nabla V(x) = G_\mathrm{v}^\eta(x)$. Similar as in the proof of Theorem 22 we can take $G_\mathrm{v}$ as a Lyapunov function to obtain

$$\frac{\mathrm{d}}{\mathrm{d}\,t}V(x) = -\|G_\mathrm{v}^\eta(x)\|^2$$

along the dynamics (8). By a similar argument $V(x)$ is non-increasing and every accumulation point $x_*$ of $\{x(t) \mid t \geq 0\}$ satisfies $G_{\mathrm{v}}^{\eta}(x)(x_*) = 0$ and belongs to $\overline{\mathcal{T}}(\tau_0)$. Note that the condition $\mathrm{v}(\mathcal{T}(\tau)) \subseteq \mathcal{T}(\tau)$ in Lemma 20 is satisfied when $\epsilon \leq \tau$. Applying Lemma 20 with $\delta = 0$ and $\widetilde{\tau} = \tau_0$ yields (11) and proves Theorem 3 for the case when $s$ is a gradient field. Now consider the more general case when $\mathrm{v}$ is not necessarily a gradient field. In this case consider $V(x) = f(\mathrm{v}(x)) + \eta \, \mathrm{d}(x)$ to obtain

$$\frac{\mathrm{d}}{\mathrm{d}\,t} V(x) = - \langle \mathrm{v}'(x) \nabla f(\mathrm{v}(x)) + \eta(x - \pi(x)), G_{\mathrm{v}}^{\eta}(x) \rangle$$
$$\leq -\|G_{\mathrm{v}}^{\eta}(x)\|^2 + \eta \|\mathrm{v}(x) - \pi(x)\| \|G_{\mathrm{v}}^{\eta}(x)\|$$
$$\leq -\|G_{\mathrm{v}}^{\eta}(x)\|^2 + \eta\epsilon \|G_{\mathrm{v}}^{\eta}(x)\|\,.$$

Thus, if $\|G_{\mathrm{v}}^{\eta}(x)\| \geq \eta\epsilon$ we have $\frac{\mathrm{d}}{\mathrm{d}\,t} V(x) \leq 0$. By a barrier function argument Khalil & Grizzle (2002) this implies that every accumulation point $x_*$ of $\{x(t) \mid t \geq 0\}$ satisfies $\|G_{\mathrm{v}}^{\eta}(x_*)\| \leq \eta\epsilon$ and belongs to $\overline{\mathcal{T}}(\tau_0)$. Again applying Lemma 20 with $\delta = \eta\epsilon$ and $\widetilde{\tau} = \tau_0$ yields (11) and finishes the proof.

# G    DISCRETIZED RIEMANNIAN GRADIENT FLOW AND DESCENT

In this section we provide proof of Theorem 5 as well as analysis of the approximate Riemannian gradient descent and the discretized landing flow. As before $C = \|\nabla f|_{\mathcal{T}(\tau_{\mathcal{M}})}\|_\infty$ and $L = \mathrm{Lip}(\nabla f|_{\mathcal{T}(\tau_{\mathcal{M}})})$ denote the supremum of $\nabla f$ and the Lipschitz constant of $\nabla f$ on $\mathcal{T}(\tau_{\mathcal{M}})$ for some manifold $\mathcal{M}$. Moreover, the following bounds will be useful:

**Lemma 23.** *If $\tau \in (0, \tau_{\mathcal{M}})$ and (46) holds for some $\epsilon > 0$, then*

$$\sup_{x \in \mathcal{T}(\tau)} \|\mathrm{v}'(x)\|_\infty \leq \epsilon + \frac{1}{1 - \tau/\tau_{\mathcal{M}}}\,.$$

*Additionally, if $\epsilon \in (0, \tau]$, then*

$$\|G_{\mathrm{v}}^{\eta}(x)\| \leq C\|\mathrm{v}'(x)\| + \eta\|\mathrm{v}(x) - x\| \leq C(\epsilon + \frac{1}{1 - \tau/\tau_{\mathcal{M}}}) + 2\eta\tau \text{ for } x \in \mathcal{T}(\tau)$$

*Proof.* Via triangle inequality, Lemma 10 and the definition of $C$. $\qquad\square$

## G.1    DISCRETIZED APPROXIMATE RIEMANNIAN GRADIENT FLOW

In this section we analyze the discretized version of (9), specifically the corresponding gradient descent

$$x_{k+1} = x_k - \gamma_k \nabla F_{\sigma}^{\eta}(x_k)\,, \tag{51}$$

for some sequence of step sizes $\{\gamma_k\}_{k=1}^{\infty}$. The selection of $\gamma_k$ can be inferred from any standard analysis of gradient descent for $F_{\sigma}^{\eta}$ to guarantee that all accumulation points of the resulting sequence $\{x_k\}_{k=0}^{\infty}$ are stationary points of $F_{\sigma}^{\eta}$. Since we can only interpret stationary points of $F_{\sigma}^{\eta}$ in terms of our original problem (1) when they are contained in a tubular neighborhood $\mathcal{T}(\tau)$ for some $\tau \in (0, \tau_{\mathcal{M}})$ (see Lemma 20), we drive conditions on the step-size to ensure $\{x_k\}_{k=0}^{\infty} \subseteq \mathcal{T}(\tau)$.

**Theorem 24.** *Let $\tau \in (0, \tau_{\mathcal{M}}/2)$ and $\sigma > 0$ be such that (46) holds for some $\epsilon \in (0, \frac{\eta\tau}{2(C+\eta)}]$. Then if $\gamma_k \in [0, \gamma_{\mathrm{tubular}}]$ with*

$$\gamma_{\mathrm{tubular}}(\epsilon, \tau, \eta) = \tau \cdot \min \left\{ \frac{1}{2(C(\epsilon + 2) + 2\eta\tau)}, \frac{\frac{1}{4}\eta\tau - \frac{1}{2}(C + \eta)\epsilon}{4(C(\epsilon + 4) + 3\eta\tau)^2} \right\}$$

*and $x_0 \in \mathcal{T}(\tau)$, the iterates $x_k$ of the discretized flow (51) belong to $\mathcal{T}(\tau)$.*

*Proof.* First we show that $x_k \in \mathcal{T}(\tau)$ implies $x_{k+1} \in \mathcal{T}(\tau)$. If $x_k \in \mathcal{T}(\tau/3)$, then, since $\gamma_k \|\nabla F_{\sigma}^{\eta}(x_k)\| \leq \tau/2$ by Lemma 23, it follows $x_{k+1} \in \mathcal{T}(\tau)$. Hence assume $x_k \in \mathcal{T}(\tau) \setminus \mathcal{T}(\tau/3)$, i.e. $\sqrt{2\,\mathrm{d}(x_k)} \geq \tau/2$. Let us write

$$\mathrm{d}(x_{k+1}) = \mathrm{d}(x_k) - \gamma_k \langle \nabla \, \mathrm{d}(x_k), \nabla F_{\sigma}^{\eta}(x_k) \rangle + \frac{1}{2}\gamma_k^2 \|(I - P_0)\,|_{\mathcal{T}(3\tau/2)}\|_\infty \|\nabla F_{\sigma}^{\eta}(x_k)\|^2\,,$$

where we have used $\nabla^2 \, \mathrm{d} = I - P_0$ and $x_{k+1} \in \mathcal{T}(3\tau/2)$, since again $\gamma_k \|\nabla F_\sigma^\eta(x_k)\| \leq \tau/2$. Then by Lemma 21

$$-\langle \nabla \, \mathrm{d}(x), \nabla F_\sigma^\eta(x)\rangle \leq -2\eta \, \mathrm{d}(x) + \epsilon(C+\eta)\sqrt{2 \, \mathrm{d}(x)}\,.$$

Now, Lemma 23 and the fact that $(1 - 3\tau/(2\tau_\mathcal{M}))^{-1} \leq 4$ imply

$$\delta = \frac{1}{2}\|(I - P_0)|_{\mathcal{T}(3\tau/2)}\|_\infty \|\nabla F_\sigma^\eta|_{\mathcal{T}(\tau)}\|_\infty^2 \leq 4(C(\epsilon+4) + 3\eta\tau)^2$$

Therefore we have

$$\mathrm{d}(x_{k+1}) - \mathrm{d}(x_k) \leq \gamma_k(-2\eta \, \mathrm{d}(x_k) + \epsilon(C+\eta)\sqrt{2 \, \mathrm{d}(x_k)} + \gamma_k \delta)\,,$$

where right hand side is non-positive for all $\sqrt{2 \, \mathrm{d}(x_k)} \geq \tau/2$ if

$$-\frac{1}{4}\eta\tau^2 + \frac{1}{2}(C+\eta)\tau\epsilon + \gamma_k\delta \leq 0\,.$$

or, equivalently,

$$\gamma_k \leq \frac{1}{\delta}\left(\frac{1}{4}\eta\tau^2 - \frac{1}{2}(C+\eta)\tau\epsilon\right)\,.$$

If these are satisfied, then $\mathrm{d}(x_{k+1}) \leq \mathrm{d}(x_k) \leq \tau^2/8$ and therefore $x_{k+1} \in \mathcal{T}(\tau)$. In all, $\{x_k\}_{k=0}^\infty \subseteq \mathcal{T}(\tau)$ provided that $x_0 \in \mathcal{T}(\tau)$. $\qquad\square$

## G.2  PROOF OF THEOREM 5

Let us write $\epsilon' = \epsilon + K(\tau, \mathcal{M}, \mu)\sigma|\log(\sigma)|^3 \leq \tau/2$. First we note that $\{x_k\}_{k=0}^\infty \subseteq \mathcal{T}(\epsilon')$ as soon as $x_0 \in \mathcal{T}(\tau/2)$, because $\mathrm{v}(\mathcal{T}(\tau)) \subseteq \mathcal{T}(\epsilon')$ and $x_k - \gamma_k \mathrm{v}'(x_k)\nabla f(x_k) \in \mathcal{T}(\tau)$, since

$$\gamma_k \leq \frac{\tau}{2C(\epsilon' + (1 - \tau/\tau_\mathcal{M})^{-1})} \leq \frac{\tau}{2\|\mathrm{v}'(x_k)\nabla f(x_k)\|}\,,$$

where first inequality is due to $(1 - \tau/\tau_\mathcal{M})^{-1} \leq 2$ and the second inequality due to Lemma 23. For brevity let us denote $y_k = x_k - \gamma_k \mathrm{v}'(x_k)\nabla f(x_k)$ and $z_k = x_k - \gamma_k P_0(x_k)\nabla f(\pi(x_k))$. Also let $L_0 = \mathrm{Lip}(\nabla(f \circ \pi)|_{\mathcal{T}(\tau)})$. By Lemma 12 and $(1 - \tau/\tau_\mathcal{M})^{-1} \leq 2$ it follows that

$$L_0 \leq C\|P_0'|_{\mathcal{T}(\tau)}\| + (1 - \tau/\tau_\mathcal{M})^{-1}L \leq 8C\left(2(\frac{3}{\tau_\mathcal{M}} + \tau M) + \frac{1}{\tau_\mathcal{M}}\right) + 2L\,.$$

Then, since $\|x_k - \pi(x_k)\| \leq \epsilon'$ for $k \geq 1$, we have

$$
\begin{aligned}
f(x_{k+1}) - f(x_k) &= f(\mathrm{v}(y_k)) - f(x_k) \\
&= f(\pi(z_k)) - f(\pi(x_k)) + f(\pi(x_k)) - f(x_k) \\
&\quad + f(\mathrm{v}(y_k)) - f(\pi(y_k)) + f(\pi(y_k)) - f(\pi(z_k)) \\
&\leq -\gamma_k \langle P_0(x_k)\nabla f(\pi(x_k)), P_0(x_k)\nabla f(\pi(x_k))\rangle + \gamma_k^2 \frac{L_0}{2}\|P_0(x_k)\nabla f(\pi(x_k))\|^2 \\
&\quad + C\|x_k - \pi(x_k)\| + C\epsilon + L_0\gamma_k(C\epsilon + L\|x_k - \pi(x_k)\|) \\
&\leq -\gamma_k(1 - \frac{L_0}{2}\gamma_k)\|P_0(x_k)\nabla f(\pi(x_k))\|^2 + (2C + L_0\gamma_k(C+L))\epsilon'\,.
\end{aligned}
$$

Now let $\gamma_k \in [\gamma_{\min}, \gamma_{\max}] \subseteq (0, \frac{2}{L_0})$. Then summing over $k = 1, \ldots, N$ yields

$$\gamma_{\min}(1 - \frac{1}{2}\gamma_{\max}L_0)\frac{1}{N}\sum_{k=1}^N \|P_0(x_k)\nabla f(\pi(x_k))\|^2 \leq \frac{f(x_1) - f(x_{N+1})}{N} + (2C + L_0\gamma_k(C+L))\epsilon'\,.$$

Now note that by Lemma 11 and the fact that $(1 - \epsilon'/\tau_\mathcal{M})^{-1} \leq 2$ it holds

$$
\begin{aligned}
\|\mathrm{grad}_\mathcal{M} f(\pi(x_k))\|^2 &\leq 2\|(P_0(\pi(x_k)) - P_0(x_k))\nabla f(\pi(x_k))\|^2 + 2\|P_0(x_k)\nabla f(\pi(x_k))\|^2 \\
&\leq 8C^2(\epsilon'/\tau_\mathcal{M})^2 + 2\|P_0(x_k)\nabla f(\pi(x_k))\|^2\,,
\end{aligned}
$$

which implies

$$\gamma_{\min}(1 - \frac{1}{2}\gamma_{\max}L_0)\frac{1}{N}\sum_{k=0}^N \|\mathrm{grad}_\mathcal{M} f(\pi(x_k))\|^2 \leq \frac{4D}{N} + 8C^2(\epsilon'/\tau_\mathcal{M})^2 + 2(2C + L_0\gamma_k(C+L))\epsilon'\,.$$

This finishes the proof.

### G.3 AUXILIARY RESULTS

#### G.3.1 A LEMMA ON NORMS OF ORTHOGONAL VECTORS

**Lemma 25.** *Let $x, y \in \mathbb{R}^n$ be orthogonal and $\epsilon > 0$. If there exists some $z \in \mathbb{R}^n$ with $\|x - z\| \leq \epsilon$ and $\|y - z\| \leq \epsilon$, then $\|x\| \leq 2\epsilon$, $\|y\| \leq 2\epsilon$ and $\|z\| \leq \sqrt{2}\epsilon$. If on the other hand for some $\delta > 0$ there exist some $z_1, z_2 \in \mathbb{R}^n$ with $\|x - z_1\| \leq \epsilon$, $\|y - z_2\| \leq \epsilon$ and $\|z_1 - z_2\| \leq \delta$, then $\|x\| \leq 2\epsilon + \delta$, $\|y\| \leq 2\epsilon + \delta$ and $\|z\| \leq \sqrt{2}\epsilon + \delta/\sqrt{2}$.*

*Proof.* The first claim follows by geometric considerations. The second follows from the first by noting that the midpoint $z = (z_1 + z_2)/2$ satisfies $\|z - z_1\| \leq \delta/2$ and $\|z - z_2\| \leq \delta/2$ and hence $\|x - z\| \leq \epsilon + \delta/2$ and $\|y - z\| \leq \epsilon + \delta/2$, i.e. $\|x\| \leq 2\epsilon + \delta$, $\|y\| \leq 2\epsilon + \delta$ and $\|z\| \leq \sqrt{2}\epsilon + \delta/\sqrt{2}$. $\square$

#### G.3.2 GAUSSIAN TAIL BOUNDS

The following elementary lemmas give simplifications for some of the constants appearing in the proof of Theorem 15 and Theorem 16.

**Lemma 26.** *It holds for $R > 0$ that*

$$\mathrm{G}_0(R) := \frac{1}{(2\pi)^{n/2}} \int_{\mathbb{R}^n \setminus B_R(0)} e^{-\frac{1}{2}\|u\|^2} \, du \leq 2e^{-\frac{1}{2n}R^2}$$

*and for $R \geq 2n$ that*

$$\mathrm{G}_2(R) := \frac{1}{(2\pi)^{n/2}} \int_{\mathbb{R}^n \setminus B_R(0)} \|u\|^2 e^{-\frac{1}{2}\|u\|^2} \, du \leq \frac{2^{2-n/2}}{\Gamma(n/2)} e^{-\frac{1}{4}R^2} \,.$$

*In particular $\mathrm{G}_0(|\log(\sigma)|) = O(\sigma^l)$ and $\mathrm{G}_2(|\log(\sigma)|) = O(\sigma^l)$ for any $l \geq 1$ as $\sigma \to 0$.*

*Proof.* See Majerski (2015). $\square$

## H FURTHER NUMERICAL EXPERIMENTS AND IMPLEMENTATION DETAILS

### H.1 OPTIMIZATION OVER $O(n)$

#### H.1.1 IMPLEMENTATION DETAILS

In all of our experiments, we discretize the flow equation 8 using the Euler scheme with a step size of $t_{\text{step}} = 1 \cdot 10^{-4}$ and set the landing gain $\eta = 3 \cdot 10^3$.

**Data generation:** In our experiments, we take $Q = \mathrm{diag}(1, \ldots, n)$ and a randomly sampled symmetric $A \in \mathbb{S}^{n \times n}$ with $\mathcal{N}(0, 1)$-entries.

**Score architecture:** We use the following score architecture:

$$s_\sigma(X) = \frac{1}{\sigma} \widetilde{s}_\sigma(X) \,,$$

with $X = \begin{pmatrix} x_1 & \cdots & x_n \end{pmatrix} \in \mathbb{R}^{n \times n}$ and $Y = \begin{pmatrix} y_1 & \cdots & y_n \end{pmatrix} = \widetilde{s}_\sigma(X)$, where

$$y_i = \mathrm{MLP}_{l,w}([r; x_i; \sigma]) \text{ for } i = 1, \ldots, n \,,$$

and MLP a fully connected multi-layer perceptron with ReLU activation function, $l$ layers of width $w$. The features $r = r(X) \in \mathbb{R}^m$ are

$$r_j(X) = \mathrm{tr}(Q_j X K_j X^\top) \,, \quad j = 1, \ldots, m$$

where $Q_j, K_j \in \mathbb{R}^{n \times n}$ are learnable weight matrices (shared for all $i = 1, \ldots, n$). For $n = 10$ we take $l = 4$, $w = 512$, $m = 128$ and for $n = 20$ we take $l = 4$, $w = 2048$, $m = 512$.

**Diffusion and training parameters:** We train minimizing equation 26 with the Adam optimizer, early stopping (i.e. $t \sim \mathrm{Unif}[\epsilon, T]$) with $T = 3$ and $\epsilon = 10^{-4}$ and a cosine learning rate scheduling from $\texttt{lr} = 10^{-3}$ to $\texttt{lr} = 5 \cdot 10^{-5}$. We use $N_{\text{epochs}} = 10000$ and $N_{\text{epochs}} = 50000$ epochs for $n = 10$ and $n = 20$, respectively.

## H.2 DATA-DRIVEN REFERENCE TRACKING

### H.2.1 BENCHMARK SYSTEMS AND TRACKING GOALS

**Benchmark systems:** We consider two classical benchmark systems: The double pendulum and the unicycle car model LaValle (2006). For the double pendulum the state is $x = (\theta_1, \omega_1, \theta_2, \omega_2)$ with $\omega_i = \dot{\theta}_i$, gravity $g$, masses $m_1, m_2$, lengths $l_1, l_2$, dampings $d_1, d_2$, control torque $u$ applied at joint 1 (first pendulum), and $\Delta\theta := \theta_2 - \theta_1$.

$$\dot{\theta}_1 = \omega_1, \qquad\qquad\qquad \dot{\theta}_2 = \omega_2,$$

$$\mathbf{M}(\theta)\,\ddot{\boldsymbol{\theta}} + \mathbf{C}(\theta, \dot{\theta}) + \mathbf{G}(\theta) + \mathbf{D}\,\dot{\boldsymbol{\theta}} = \boldsymbol{\tau}, \quad \boldsymbol{\theta} = \begin{pmatrix} \theta_1 \\ \theta_2 \end{pmatrix}, \quad \boldsymbol{\tau} = \begin{pmatrix} u \\ 0 \end{pmatrix},$$

with

$$\mathbf{M}(\theta) = \begin{pmatrix} (m_1 + m_2)l_1^2 & m_2 l_1 l_2 \cos\Delta \\ m_2 l_1 l_2 \cos\Delta\theta & m_2 l_2^2 \end{pmatrix}, \qquad \mathbf{D} = \begin{pmatrix} d_1 & 0 \\ 0 & d_2 \end{pmatrix},$$

$$\mathbf{C}(\theta, \dot{\theta}) = \begin{pmatrix} -m_2 l_1 l_2 \sin\Delta\theta\,\omega_2^2 \\ m_2 l_1 l_2 \sin\Delta\theta\,\omega_1^2 \end{pmatrix}, \qquad \mathbf{G}(\theta) = \begin{pmatrix} (m_1 + m_2)g l_1 \sin\theta_1 \\ m_2 g l_2 \sin\theta_2 \end{pmatrix}.$$

We pick the output $y = (\theta_1, \theta_2)$ and set $m_1 = l_1 = g = 1$, $m_2, l_2 = 0.5$ and $d_1, d_2 = 0.1$. For the unicycle car model the dynamics is given by $x = (\mathrm{x}, \mathrm{y}, \theta)$ with

$$\dot{\mathrm{x}} = v\cos(\theta), \quad \dot{\mathrm{y}} = v\sin(\theta), \quad \dot{\theta} = \omega$$

and the input $u = (v, \omega)$ and output $y = x = (\mathrm{x}, \mathrm{y}, \theta)$. Here $(\mathrm{x}, \mathrm{y})$, $v$, $\theta$, $\omega$ is the car's position, velocity, angle and angular velocity, respectively.

**Tracking goals:** For the double pendulum system the goal is to track a reference trajectory $r$ via the first joint angle $\theta_1$ and we pick the optimal control objective $f$ to be 15 with

$$Q = \begin{pmatrix} 10 & 0 \\ 0 & 10 \end{pmatrix}, \quad R = 0.01,$$

while for the unicycle car model the goal is to track a positional reference $r = (r_{\mathrm{x}}, r_{\mathrm{y}})$, i.e. we pick

$$Q = \begin{pmatrix} 10 & 0 & 0 \\ 0 & 10 & 0 \\ 0 & 0 & 0 \end{pmatrix}, \quad R = \begin{pmatrix} 0.01 & 0 \\ 0 & 0.01 \end{pmatrix}.$$

Here $R$ is some small penalty on the input $\boldsymbol{u}$ to keep it bounded during the optimization.

### H.2.2 IMPLEMENTATION DETAILS

**Discretization:** We discretize both continuous-time dynamics $\dot{x} = \boldsymbol{f}_{\mathrm{cont}}(x, u)$ via the RK4-method and discretization step $\Delta t$ to obtain the discrete-time dynamics (13) as

$$\boldsymbol{f}(x, u) = x + \frac{\Delta t}{6}(k_1 + 2k_2 + 2k_3 + k_4) \quad \text{where} \quad \begin{cases} k_1 &= \boldsymbol{f}_{\mathrm{cont}}(x, u) \\ k_2 &= \boldsymbol{f}_{\mathrm{cont}}(x + \frac{1}{2}\Delta t k_1, u) \\ k_3 &= \boldsymbol{f}_{\mathrm{cont}}(x + \frac{1}{2}\Delta t k_2, u) \\ k_4 &= \boldsymbol{f}_{\mathrm{cont}}(x + \Delta t k_3, u) \end{cases}$$

We use $\Delta t = 0.1$ for the double pendulum and $\Delta t = 0.05$ for the unicycle model.

**Data generation:** To generate trajectories, we use i.i.d. random inputs $u_k \sim \mathrm{Unif}[-5, 5]$ for the double pendulum and $u_k = (v_k, \omega_k) \sim \mathrm{Unif}[0, 1] \otimes \mathcal{N}(0, 25)$ and a horizon of $N_h = 100$ for both system. We use $N_{\mathrm{data}} = 50000$ trajectories for the double pendulum and $N_{\mathrm{data}} = 20000$ trajectories for the unicycle model.

**Score architecture:** The score architecture is a 1-dimensional version of the standard UNet architecture Ronneberger et al. (2015) with a sin-cos time-embedding Song et al. (2020) and

residual connections, where the different input-, state- and output dimensions are concatenated and treated as additional channels. The down- and upsampling convolutions are done w.r.t. the temporal dimension and channels.

**Diffusion and training parameters:** Same as in Appendix H.1.1 with this time $N_{\text{epochs}} = 50000$ training epochs. For the DRGD step-size we pick a fixed step-size of $\gamma = 0.001$.

### H.2.3 EXPERIMENTS

In Figure 4 we present the objective value evolution for the experiment from Section 6.2.

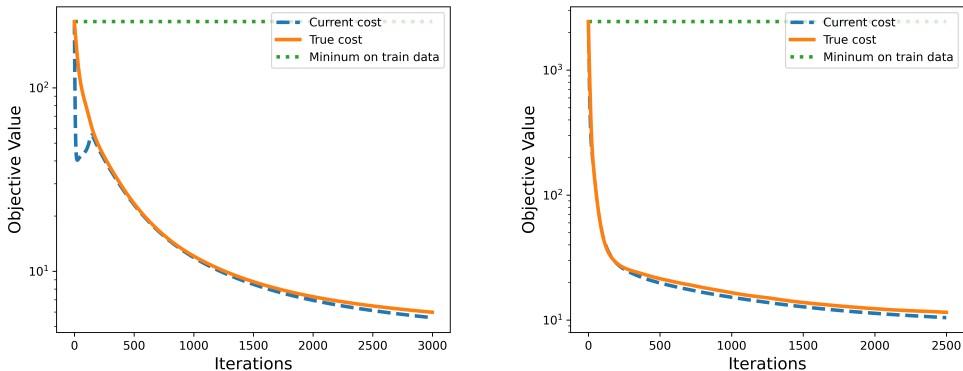

Figure 4: Denoising Riemannian gradient descent: Objective value $f$ vs the iteration count $j$ for double pendulum (left) and unicycle car model (right). Note the logarithmic scale on the y-axis. The current cost (blue, dashed) is the objective value $f(\boldsymbol{u}_j, \boldsymbol{y}_j)$ at the current (in general infeasible $(\boldsymbol{u}_j, \boldsymbol{y}_j) \notin \mathcal{M}_{\text{IO}}$) iterate, while the true cost (orange) is the value $f(\boldsymbol{u}_j, \boldsymbol{y}_j^{\text{true}})$, with $\boldsymbol{y}_j^{\text{true}}$ obtained by simulating (13) with input $\boldsymbol{u}_j$.

In Figure 5 we show optimized trajectories for two other reference trajectories. Note that we have set our iteration budget at $N = 4000$, while the objective is still decreasing. How to accelerate the denoising Riemannian gradient descent without losing feasibility is a core question for future work.

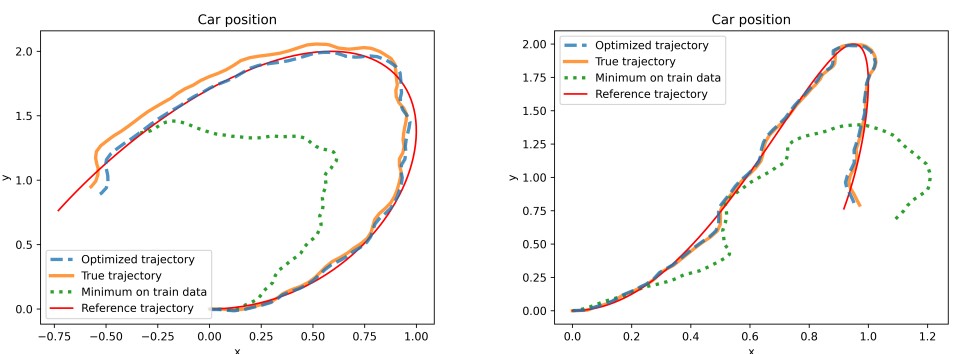

Figure 5: Denoising Riemannian gradient descent: Unicycle car position (right) with the optimized output trajectory $\boldsymbol{y}^*$ (blue, dashed), the true system trajectory $\boldsymbol{y}^{\text{true}}$ (orange), the initial trajectory $\boldsymbol{y}_0$ (green, dotted) and the reference trajectory $\boldsymbol{r}$ (red)

