# OpenReview forum: "Landing with the Score: Riemannian Optimization through Denoising"
_ICLR.cc/2026/Conference — ICLR 2026 Poster_

### Official Review · Reviewer_rhkJ · 2025-10-30

**Soundness:** 3
**Presentation:** 4
**Contribution:** 3
**Rating:** 8
**Confidence:** 3

**Summary:**

In this paper, the author's introduce a new way to harness data manifolds described using score-based diffusion models. By introducing a so-called link function, whose gradient is related to the score function and thus to the projection onto the data manifold, they derive Riemannian optimization algorithms that work directly on the score-based description of the manifold. The authors analyze the expected performance of their algorithms under accuracy assumptions on the underlying score functions. Finally, they demonstrate the applicability of their method in two synthetic examples.

**Strengths:**

To me, the main strength of the paper is that it uses a representation of data manifold that is widely used in current machine learning (i.e. score-based models) to introduce practically usable algorithms on said manifolds. There is, of course, a lot of papers currently that try to achieve something to this end. Here, I enjoyed the clear combination of geometry and probabilistic components in the construction of the method. The resulting algorithms seem natural and actually usable in practice.

Furthermore, I appreciate the provided theoretical results, which give an insight into what to expect of the method. At the same time, the authors do not oversell the impact of their results

Finally, I found the paper to be well written. I could read the paper in one go without any road blocks even though I am not an expert on optimization. After this, I believe to have understood the method to a point where I could now implement it myself, which is huge plus in my book. I also enjoyed the discussion of related work and clear positioning of the paper.

**Weaknesses:**

The primary weakness of the presented, from my perspective, is the experimental validation. It is limited to two synthetic examples, one on O(n) and one discrete data-driven control. Since the authors themselves make the point that score-based models have become ubiquitous in modern machine learning, I would have hope that one of these would also be considered here. For example, for image generation models one could try to modify a given image to mislead a given classifier. There are certainly other creative examples one could come up with. However, I understand that this requires a different expertise and computational resources, and this is not a deal-breaker for me.

**Questions:**

- the authors should please check again their usage of \citep and \citet. In some places, it seems to be off.
- I would maybe switch (8) and (9), so that the method is first introduced with the more accurate \pi before going to the score-based approximation.
- One reference I missed is "What regularized auto-encoders learn from the data-generating distribution"  by Guillaume Alain and Yoshua Bengio. It also analyses denoising autoencoders (although not considering an underlying manifold) and it would be nice to set it into relation with the present work.

---

> ### Author Response · Authors · 2025-11-25
>
> Thank you for the positive feedback on our work.
> We will include the typographical suggestions in the revised manuscript and proceed to address your main comment, i.e.
>
> > One reference I missed is "What regularized auto-encoders learn from the data-generating distribution" by Guillaume Alain and Yoshua Bengio. It also analyses denoising autoencoders (although not considering an underlying manifold) and it would be nice to set it into relation with the present work.
>
> In the following we explain how this work is related to ours and will include this discussion in the updated manuscript as well.
> In [1] it is shown [1, Theorem 1] that when a denoising auto-encoder $r = d \circ e$ is trained using the expected quadratic loss and noise level $\sigma > 0$, i.e. by minimizing
>
> $$
> L(r) = \mathbb{E}_{x \sim p,\ \epsilon \sim \mathcal{N}(0,I)} \Vert x - r(x+\sigma \epsilon)\Vert^2,
> $$
>
> then the optimal reconstruction function $r_\sigma^*$ satisfies
> ```
> $$
> r_\sigma^*(x)
> = \frac{\mathbb{E}_{\epsilon \sim \mathcal{N}(0,I)} p(x-\sigma \epsilon)(x-\sigma \epsilon)}{\mathbb{E}_{\epsilon \sim \mathcal{N}(0,I)} p(x-\sigma \epsilon)}
> \overset{\sigma \to 0}{=} x + \sigma^2 \nabla \log p(x) + o(\sigma^2)
> $$
> ```
> whenever $p(x) > 0$, where $p$ is the (positive) Lebesgue-density of the data distribution $\mu$.
> The function $r_\sigma^*$ here is precisely our “approximate projection” $\pi_\sigma$, since the first above expression corresponds to the posterior of observing $x$ under the convolution of the prior $p$ with $\mathcal{N}(0,\sigma^2 I)$, i.e. in our notation
>
> $$
> r_\sigma^*(x)
> = \mathbb{E} \nu_{x,\sigma}
> = \pi_\sigma(x).
> $$
>
> Therefore, if $\mu$ has the Lebesgue-density $p$ (i.e. the data manifold would have full dimension $k=d$), then **[1] shows**
>
> $$
> x + \sigma^2 \nabla \log p_\sigma(x)
> \overset{\sigma \to 0}{=} x + \sigma^2 \nabla \log p(x) + o(\sigma^2).
> $$
>
> In particular, if $p(x) > 0$, then the last equation states that
>
> $$
> \lim_{\sigma \to 0} \bigl(x + \sigma^2 \nabla \log p_\sigma(x)\bigr)
> = x.
> $$
>
> Our work deals with the case $k < d$, i.e. when the manifold is truly lower-dimensional. In this case, $\mu$ has no density $p$ w.r.t. the Lebesgue measure on $\mathbb{R}^d$, and **we show**
>
> $$
> \lim_{\sigma \to 0} \bigl(x + \sigma^2 \nabla \log p_\sigma(x)\bigr)
> = \pi(x),
> $$
>
> where $\pi$ denotes the projection operation onto the data manifold $\mathcal{M}$.
> While [1] does consider manifold support in their discussion (see [1, Section 3.4, Figure 5]), note that the population data measure in this case is **not** absolutely continuous w.r.t. the Lebesgue measure on $\mathbb{R}^d$, and hence $\log p(x)$ has no meaning. Consequently, results of [1], e.g. the expansion above regarding $r_\sigma^*$, are not applicable.
>
> In conclusion, our work can be seen as a true generalization of [1], establishing the asymptotic convergence of $r_\sigma^*$ and its Jacobian for $\sigma \to 0$ as
> ```
> $$
> r_\sigma^*(x) = \pi(x) + O(\sigma \lvert \log \sigma \rvert^3),, \quad
> (r_\sigma^*)'(x) = \pi'(x) + O(\sigma \lvert \log \sigma \rvert^3),.
> $$
> ```
> [1] G. Alain and Y. Bengio, *What regularized auto-encoders learn from the data-generating distribution*.

---

### Official Review · Reviewer_xQd1 · 2025-10-30

**Soundness:** 2
**Presentation:** 2
**Contribution:** 2
**Rating:** 2
**Confidence:** 3

**Summary:**

The paper targets the problem of optimizing objective function on data manifolds, i.e., manifolds that are given only implicitly through the data distribution. The paper proposes to do so via Riemannian optimization, by obtaining the required quantities (Riemannian gradient and Hessian, projection onto the tangent space) through a link function associated with the data distribution which is associated with a diffusion score function.

**Strengths:**

The paper targets an interesting problem, which is relevant to the field of manifold learning. The approach nicely combines manifold learning with elements from recent works in diffusion models and score matching. The paper is generally clearly written.

**Weaknesses:**

The paper does not mention an important category of related works in manifold learning, where the data manifold is characterized via the pullback metric of the decoder of a (deep) generative model (Tosi et al, Arvanitidis et al, Syrota et al, Diepeveen et al). In such cases, the data manifold is endowed with a Riemannian pullback metric, which can directly be used for computing explicit manifold operations for Riemannian optimization. In this case, the motivation of the paper (“In this data-driven regime, methods from classical Riemannian optimization cannot be applied directly, since they rely on explicit manifold operations such as tangent-space projection, retraction, or exponential maps”) does not hold anymore, as such quantities can be computed based on the Riemannian metric. I think that these approaches must be discussed, and some of the claims of the paper revised according to this perspective.

In addition, it is unclear to me if the proposed approach really qualifies as Riemannian optimization. In Riemannian optimization (Absil et al, Boumal, as cited in the paper), the objective function is defined on the manifold of interest and the optimization strictly takes place on the manifold, i.e., all iterate x lie on M. In contrast, in equation 10, the second term belongs to the orthogonal space to the manifold, and pushes the iterate towards the manifold, i.e., the iterates do not necessarily belong to the manifold of interest.

The experiments provided in the paper are simple and conducted only on low-dimensional spaces. I would like to see how the approach performs on more complex and higher dimensional optimization problems. The fact that score functions and diffusion models scale well to high dimensions could allow the approach to scale well, and could be presented as a strength if shown experimentally.

All in all, the performance of the approach seems very dependent on the value of $\sigma$. In Fig. 3, only $\sigma=1e^{-5}$ leads to a good optimization, with significant differences compared to $\sigma=1e^{-4}$. It is unclear how the approach would perform for different values of $\sigma$ in more complex optimization problems.

Given that the link function seems to be available based on the data distribution, which is assumed to be given, it is unclear to me why it is necessary to additionally learn a network $s$ to represent the gradient of the link function.


References:
- Tosi, A., Hauberg, S., Vellido, A. & Lawrence, N. D. Metrics for Probabilistic Geometries. in UAI (2014).
- Arvanitidis, G., Hansen, L. K. & Hauberg, S. Latent Space Oddity: on the Curvature of Deep Generative Models. in International Conference on Learning Representations (ICLR) (2018).
- Syrota, S., Zainchkovskyy, Y., Xi, J., Bloem-Reddy, B. & Hauberg, S. Identifying metric structures of deep latent variable models. in ICML (2025).
- Diepeveen, W., Batzolis, G., Shumaylov, Z. & Schönlieb, C.-B. Score-based pullback Riemannian geometry: Extracting the Data Manifold Geometry using Anisotropic Flows. in ICML (2025).

**Questions:**

- It is unclear to me why it is necessary to learn a network $s$. Could the gradient $\nabla \ell_{\sigma}$ and Hessian be computed solely from equation 3 instead?
- How is the network $s$ trained? Which training data are needed?
- How does the proposed approach compare to Riemannian optimization on learned data manifold endowed with a Riemannian metric?
- How is the probability measure obtained in general? How is it obtained in the problems considered in the experiments?
- How does the approach perform for more complex and higher dimensional optimization problems?
- How should $\sigma$ be chosen in practice?
- What does “exact landing” refers to in Figure 3?
- How does the proposed approach compare to gradient descent on the O(n) manifold? I know that the paper considers the manifold as unknown, but providing such a comparison would still be valuable to understand the influence of the manifold learning component.
- It would be better if figures corresponding to main results, e.g., Figure 3, would appear in the main text. If space is an issue, I would suggest to reduce the introduction (which takes 4 pages out of 9).

---

> ### Author Response · Authors · 2025-11-20
>
> We thank the reviewer for the feedback.
>
> First, we would like to emphasize that our primary goal goes beyond manifold learning: we aim to solve Riemannian optimization problems over the data manifold. Consequently, the design of our algorithm is guided not only by the ability to recover manifold operations, but also by the requirement that these recovered operations yield a provably correct and computationally efficient optimization procedure.
>
> We would also like to clarify a potential misunderstanding about our setting. In our paper, $\mu_{\mathrm{data}}$ denotes an **unknown underlying** population distribution, which we use as a conceptual object for the theoretical analysis in Section 3. In practice, we only have access to a finite **empirical dataset**, and a score function $\texttt{s}$ is learned to approximate the ground-truth score $\nabla \log p_\sigma$ via denoising score matching, as is standard in the training of diffusion models. The learned score is then plugged into our proposed algorithms, DLF and DRGD.
> In Sections 4 and 5, under the assumption that $\texttt{s}$ approximates $\nabla \log p_\sigma$ sufficiently well, we establish convergence guarantees for these methods. By contrast, determining how many data points are needed to achieve the required score accuracy is a statistical question, which we view as an important direction for future work.
> We believe making this distinction explicit helps address several of the Reviewer’s concerns, and we will incorporate this clarification in the revised manuscript.
>
> Additionally, for the reviewer's questions regarding further numerical experiments (such as comparison with existing work (Comment 1, Question 3, 8), performance on more high-dimensional data (Comment 3, Question 5), empirical choice of noise $\sigma$ (Question 6)), we will provide the new results in the coming days after some additional clarifications.
>
> Now we address the reviewer's individual comments.
>
> **Comment 1:**
>
> > The paper does not mention an important category of related works in manifold learning, ... I think that these approaches must be discussed, and some of the claims of the paper revised according to this perspective.
>
> We thank the Reviewer for pointing out this important line of work on manifold learning, in particular in identifying the pullback metrics on the latent space. Our understanding of the comment is that the Reviewer is referring to approaches where a decoder $\psi : \mathcal{Z} \to \mathbb{R}^d$ induces a pullback Riemannian metric on the latent space $\mathcal{Z}$, and this metric is then used to implement the **pullback gradient flows** in the latent space for Riemannian optimization. In a separate response, we compare this method with our proposed approach (see "Compare with the pull-back gradient flow" comment below).
> Once we have confirmed that this is the method the reviewer had in mind, we will definitely cite the works mentioned by the reviewer and discuss them in our revision. We are also happy to provide some empirical comparison with this method.
>
> However, if we have misunderstood the Reviewer’s intention and the comment was referring to a different mechanism for obtaining explicit manifold operations from the pullback metric, we would be very grateful for more specific pointers. We will gladly incorporate these references and clarify the relationship to our framework in the revised version.

---

> ### Author Response · Authors · 2025-11-20
>
> **Comment 2:**
>
> > In addition, it is unclear to me if the proposed approach really qualifies as Riemannian optimization. In Riemannian optimization (Absil et al, Boumal, as cited in the paper), the objective function is defined on the manifold of interest and the optimization strictly takes place on the manifold, i.e., all iterate x lie on M. In contrast, in equation 10, the second term belongs to the orthogonal space to the manifold, and pushes the iterate towards the manifold, i.e., the iterates do not necessarily belong to the manifold of interest.
>
> While it is true that the original meaning of "Riemannian optimization" concerned the development of algorithms that are intrinsically constrained to the manifold only, most recent work in the domain of Riemannian optimization also considers infeasible algorithms such as (discretized) landing flows [5], [10].
> For instance, in [5] the flow defined by [5,(7)] ventures out of the explicitly defined manifold [5,(1)] (see also the discussion in [5, Section 1]).
> The advantage of landing algorithms there is that they potentially do not necessarily need to compute a retraction at each step, thus having a lower per-iterate computational complexity.
>
> Further, even in the feasible on-manifold setting, classical references [3] and [4] do consider and develop algorithms that utilize various retractions from the ambient space, as opposed to using the exponential map to take a descent step in the direction of a geodesic.
> For instance, if the manifold is embedded in $\mathbb{R}^n$, Riemannian Gradient Descent takes steps in the tangent space (which can be seen as being embedded in $\mathbb{R}^n$ as well) and retracts them via a retraction defined on the ambient $\mathbb{R}^n$ (see [3, Section 4.1.1, Examples 4.1.1–4.1.3]).
> Our Denoising Riemannian Gradient Descent follows the same methodology by retracting the iterates back to the manifold.
>
> In our setting, due to the inexactness of the manifold operations, we can only guarantee feasibility in the small-noise limit $\sigma \to 0$.
> However, this is to be expected if the underlying manifold is not known explicitly and is learned from data (such as diffusion models).
>
> ---
>
> **Comment 3:**
>
> > The experiments provided in the paper are simple and conducted only on low-dimensional spaces. I would like to see how the approach performs on more complex and higher dimensional optimization problems. The fact that score functions and diffusion models scale well to high dimensions could allow the approach to scale well, and could be presented as a strength if shown experimentally.
>
> We want to highlight that our current experiments are already operating in the ambient dimension of $(d_u + d_y) N_h$, which is $500$ and $400$ for the unicycle car and double pendulum, respectively.
> When comparing with classical manifold learning algorithms [9], we believe this is already not low-dimensional.
>
> ---
>
> **Comment 4:**
>
> > All in all, the performance of the approach seems very dependent on the value of $\sigma$. In Fig. 3, only $\sigma=1e\text{-}5$ leads to a good optimization, with significant differences compared to $\sigma=1e\text{-}4$. It is unclear how the approach would perform for different values of in more complex optimization problems.
>
> We fully agree on this.
> There is a trade-off in picking $\sigma > 0$ between being a good approximation to the manifold operations (see Theorem 1) and numerical stability:
> Since our approximate projection is computed via
> $$
> \pi_\sigma(x) = x + \sigma^2 ,\texttt{s}(x,\sigma),
> $$
> with $\texttt{s}(x,\sigma)$ being the approximate score, picking $\sigma$ too small will lead to difficulties learning the score $\nabla \log p_\sigma(x)$ with a neural network $\texttt{s}(x,\sigma)$, because $\lVert \nabla \log p_\sigma(x)\rVert \to \infty$ for $\sigma \to 0$.
> In practice we have treated $\sigma$ as a hyperparameter dependent on the concrete score architecture.
>
> > Given that the link function seems to be available based on the data distribution, which is assumed to be given, it is unclear to me why it is necessary to additionally learn a network to represent the gradient of the link function.
>
> Since we assume that the population data distribution $\mu_{\operatorname{data}}$ is **unknown** and **only samples $x_1,\ldots,x_N$ are available**, we choose not to learn the link function directly; instead, we learn a score neural network from the samples to approximate **the gradient of the link function**. Moreover, we highlight that we use the denoising score matching approach to train the score network, which perfectly aligns with our goal and is well-developed in the literature on diffusion models.

---

> ### Author Response · Authors · 2025-11-20
>
> Now we come to answer the questions posed by the reviewer:
>
> > Question 1: It is unclear to me why it is necessary to learn a network $s$. Could the gradient $\nabla \ell_{\sigma}$ and Hessian be computed solely from equation 3 instead
>
> See the answer to the previous comment: While the gradient can be computed solely from the data points of the empirical distribution, it will require an exponential amount of data to be practically accurate.
> Consequently, instead of directly plugging in the empirical version of $\nabla \ell_{\sigma}$ and $\nabla^2 \ell_{\sigma}$, we use the samples to learn a score network and use that for our algorithms.
>
>
> > Question 2: How is the network $s$ trained? Which training data are needed?
>
> The network is trained as a diffusion score model minimizing the conditional score matching loss.
> See Appendix C in our paper for more details on diffusion models.
>
> The conditional score matching loss is defined in (23) by
> ```
> $$
> L_{\operatorname{CSM}}(s)
>   = \mathbb{E}_{t \sim \operatorname{Unif}[0,T]}
>     \mathbb{E}_{x_0 \sim \mu}
>     \mathbb{E}_{x \sim p_{\sigma(t)}(\cdot\mid x_0)}
>     \,\sigma(t)^2
>     \left\Vert s_{\sigma(t)}(x) - \nabla \log p_{\sigma(t)}(x\mid x_0)\right\Vert^2 \,.
> $$
> ```
> Given a noise schedule $\sigma(t)$, it minimizes the mismatch of $s$ and the Stein score function of the convolved distribution (see [6] for more details), i.e.
> ```
> $$
> \arg\min_s L_{\mathrm{CSM}}(s)
> = \arg\min_s \mathbb{E}\Big[ \sigma(t)^2 \,\big\| s_{\sigma(t)}(x)
>     - \nabla \log p_{\sigma(t)}(x) \big\|^2 \Big]_{t \sim \mathrm{Unif}(0,T),\, x \sim p_{\sigma(t)}}.
> $$
> ```
> In Appendix G we describe the architecture and the hyperparameters used for each numerical example.
>
> * For the orthogonal group $O(n)$ we use an MLP with trace features of the input matrix (see Appendix G.1).
> * For the dynamical systems we use the 1D-version of UNet architecture (see [8] for the UNet architecture, and also Appendix G.2).
>
> The data comes from empirical samples obtained from a probability measure that we assume to be supported on the manifold $\mathcal{M}$.
>
> * In the case of $O(n)$ we obtain the training data by sampling random orthogonal matrices uniformly. This is done by randomly sampling square matrices with standard normal entries and performing a QR-decomposition. Then $Q \operatorname{diag}(\operatorname{sign}(\operatorname{diag}(R)))$ will be uniformly distributed on $O(n)$.
> * In the case of dynamical systems, we sample random inputs $u$ and simulate the corresponding $y$ via the underlying system. Here we don't get a uniform distribution over the data manifold, but this is not needed for our theory (we only need $\operatorname{supp}(\mu) = \mathcal{M}$).
>
> In both cases the actual manifold/system is **unknown** to our model network $s$. The model $s$ is only supplied with the generated data.
>
> > Question 3: How does the proposed approach compare to Riemannian optimization on learned data manifold endowed with a Riemannian metric?
>
> See the answer to the first comment on our interpretation of the Riemannian metric approach. Since we do not know if this is the approach that is mentioned by the reviewer, we would appreciate a clarification regarding this.
>
> > Question 4: How is the probability measure obtained in general? How is it obtained in the problems considered in the experiments?
>
> We assume that the manifold is **unknown**.
> The only information about it is given by measured data, which comes from the unknown population distribution $\mu_{\mathrm{data}}$ supported on the manifold.
> This is consistent with the manifold hypothesis that posits that many distributions of real-world data are supported on lower-dimensional manifolds, which are **not explicitly known**.
>
> In our experiments we consider synthetic manifolds where we already have access to a well-defined distribution.
> In case of $O(n)$ this is the uniform measure and in case of dynamical systems this is the distribution obtained by taking the uniform distribution on the inputs $u$ and pushing it (by the dynamical map) to a distribution on the output $y$.
> The latter procedure (getting input-output trajectories from dynamical systems) is consistent with usual practice in system identification experiments [7].
>
> > Question 7: What does “exact landing” refers to in Figure 3?
>
> It refers to the landing flow (8) with $s$ constructed from the exact operations, i.e. $s(x) = \pi(x)$, where $\pi$ is the (exact) closest point projection onto $\mathcal{M}$, since we want to compare how our error (i.e. due to non-zero noise $\sigma > 0$) affects the flow dynamics.
> We will clarify this in the caption of the figure.
>
> > Question 9: It would be better if figures corresponding to main results, e.g., Figure 3, would appear in the main text. If space is an issue, I would suggest to reduce the introduction (which takes 4 pages out of 9).
>
> We thank the reviewer for the suggestion. We will modify the introduction and put the main result Figures in the main body

---

> ### Author Response · Authors · 2025-11-20
> **A discussion on the pull-back gradient flow**
>
> ### Compare with the pull-back gradient flow
>
> To facilitate a clear comparison, we first summarize the pull-back gradient flow in a succinct but formal manner and then contrast it with our proposed method.
>
> Let $\mathcal{M} \subseteq \mathbb{R}^d$ denote the data manifold.
> Let $(\phi^{-1}, U)$ be a chart in the atlas of $\mathcal{M}$ where $U\subseteq \mathcal{M}$ is the chart domain, $\phi^{-1}: U\rightarrow \mathcal{Z} \subseteq \mathbb{R}^k$ is the coordinate map, and $k$ is the dimension of $\mathcal{M}$. (In the context of machine learning, $\mathcal{Z}$ is typically referred to as the latent space and $\phi$ is often approximated by a decoder $\psi$, parameterized by a neural network.)
>
> Consider the minimization problem with a manifold constraint:
> $$
> \min f(x), \quad \mathrm{s.t.}\ x\in\mathcal{M}.
> $$
>
> The Riemannian gradient flow in the embedding space $\mathbb{R}^d$ (Embedding RGF)
> $$
> \dot{x} = - \mathrm{grad}_\mathcal{M} f(x), \quad x(0)= x_0,
> $$
>
> can be equivalently written as the following flow in the latent space $\mathcal{Z}$ (Pull-back GF)
> $$
> \dot{z} = - G(z)^{-1} J_\phi(z)^\top \nabla f(x)\big\vert_{x = \phi(z)}, \quad z(0) = \phi^{-1}(x_0) \in \mathcal{Z},
> $$
> where by equivalent, we mean $x(t) \equiv \phi(z(t))$ as long as $\phi(z(t)) \in U$.
>
> Here $\mathrm{grad}_\mathcal{M} f$ denotes the Riemannian gradient of $f$ on $\mathcal{M}$ and
>
> $J_\phi(z)$ denotes the Jacobian matrix of $\phi$ at $z$
>
> $G:\mathcal{Z}\rightarrow\mathbb{R}^{k\times k}$ denotes the metric tensor of the pull-back metric on $\mathcal{Z}$ induced by the map $\phi$, where
> $$
> G(z) = J_\phi(z)^\top J_\phi(z) \in \mathbb{R}^{k\times k}.
> $$
>
> We note that the Euler discretization of the continuous dynamics of $z$ is typically known as natural gradient descent (see Theorem 1 of [1]) in the classical Riemannian optimization literature. Clearly, to carry out the above Pull-back GF, one needs explicit access to the local chart representation of the manifold $\mathcal{M}$, which is not available under the data manifold setting.
>
> To generalize the Pull-back GF to the data manifold setting, as suggested by the reviewer, one can use a deep generative model to learn a decoder $\psi$ that is close to the ground truth function $\phi$.
> While this is an interesting research direction, we point out that there are a few challenges that should not be neglected.
>
>
> * **Parameterizing the full atlas of a manifold is hard.** The atlas of a manifold typically consists of multiple charts, and we discuss two natural strategies to parameterize it.
>   (1) One approach is to parameterize the inverse coordinate map $\phi_i, i=1,2,\ldots$ in each chart with a neural network $\psi_i$. However, this requires explicit knowledge of how the data manifold is partitioned into charts, which is not available in practice. Moreover, this multi-chart parameterization is rarely used in common generative models such as GANs, VAEs, normalizing flows, or diffusion models.
>   (2) An alternative strategy, as a common practice, is to use a single global chart and set $\mathcal{Z} = \mathbb{R}^k$. In this case, since $\psi$ is required to be a diffeomorphism, the latent space $\mathcal{Z}$ must share all topological invariants with $\mathcal{M}$ (e.g., the fundamental group). However it is typically unrealistic to assume a complex data manifold is homeomorphic to $\mathbb{R}^k$.
>   Consequently, it is highly challenging for such a global-chart parameterization to fully capture the manifold $\mathcal{M}$ in practice.
>
>   Last but not least, for both options, one needs to know the manifold dimension $k$ a priori so that the dimension of $\mathcal{Z}$ can be properly chosen, which is another highly non-trivial task.
>
> * **Compatibility with generative model training.** The implementation of Pull-back GF requires the inverse coordinate map $\phi$ to be well approximated by a neural network $\psi$. However, it is not straightforward to align this requirement with the standard training objectives of most commonly used generative models, such as the likelihood-based loss in VAEs, the denoising score-matching loss in diffusion models, or the adversarial loss in GANs.
>   The only training objective we are aware of that is directly related to learning an inverse coordinate map is the loss used in M-Flow [2], which penalizes the reconstruction error of $\psi$ via
>
>   $$
>   \mathbb{E}_{x\sim p*\mathrm{data}}|| \psi^{-1}(\psi(x)) - x||^2,
>   $$
>
>   where we use $p_\mathrm{data}$ to denote some empirical data distribution.
>   However, since this objective requires explicit access to the inverse $\psi^{-1}$, it effectively restricts $\psi$ to be a normalizing flow, and is, to the best of our knowledge, fundamentally incompatible with architectures such as U-Nets or transformers.

---

> ### Author Response · Authors · 2025-11-20
>
> * **Expensive update rule.** The update rule of Pull-back GF requires (1) computing the matrix $G(z)$, which in turn involves the explicit evaluation of the Jacobian $J_{\phi}$, and (2) inverting $G(z)$. Both operations become computationally expensive when $\phi$ is implemented by a neural network $\psi$, even for a moderate intrinsic dimension $k$.
>
> * **Difficult convergence analysis.** In practice, one cannot expect perfect recovery of the ground-truth coordinate map $\phi$, so it is crucial to establish convergence guarantees for the optimization procedure even when only an approximate map $\psi$ is available. However, the complicated update rule of Pull-back GF makes such an analysis challenging. Moreover, the existing guarantees for Pull-back GF (which at best ensure convergence to a stationary point under reasonable computational budgets) are formulated in the latent space. Ideally, we would like guarantees in the embedding space, since only convergence there is semantically meaningful for the original data manifold.
>
> We now discuss the advantages of our method, which can be viewed as an approximate embedding RGF, along the above dimensions:
>
> * **Easy to parameterize with a neural network.** The central object in our approach is the projection operator onto the data manifold $\mathcal{M}$, which is provably $C^2$ in the tubular neighborhood of any $C^3$ manifold. Assuming $\mathcal{M}$ is compact, standard universal approximation results ensure that the target map $\pi$ can be well-approximated by a single neural network.
>
>   Importantly, an additional advantage of our approach is that it requires no knowledge about the manifold dimension $k$.
>
> * **Alignment with denoising score matching.** We approximate $\pi$ via the score function of $p_\sigma$ (up to scaling and a residual term of $x$), so the denoising score-matching loss used to train diffusion models is directly aligned with our target quantity. Our approach imposes no architectural constraints: in our experiments, for example, we successfully use both MLPs and U-Nets.
>
> * **Simple and efficient update rule.** Our update rule is highly compatible with modern hardware and software stacks and requires only a single forward–backward pass of the network per iteration (see Remark 4 in our paper).
>
> * **Convergence guarantees in the embedding space.** We provide explicit convergence analysis with guarantees on both optimality and feasibility, stated directly in the embedding space, where convergence is semantically meaningful for the original data manifold.
>
> [1] Amari, Shun-Ichi. "Natural gradient works efficiently in learning." Neural computation 10.2 (1998): 251-276.
>
> [2] Brehmer, Johann, and Kyle Cranmer. "Flows for simultaneous manifold learning and density estimation." Advances in neural information processing systems 33 (2020): 442-453.
>
> [3] Absil, P-A., Robert Mahony, and Rodolphe Sepulchre. Optimization algorithms on matrix manifolds. Princeton University Press, 2008.
>
> [4] Boumal, Nicolas. An introduction to optimization on smooth manifolds. Cambridge University Press, 2023.
>
> [5] Schechtman, Sholom, et al. "Orthogonal Directions Constrained Gradient Method: from non-linear equality constraints to Stiefel manifold." The Thirty Sixth Annual Conference on Learning Theory. PMLR, 2023.
>
> [6] Song, Yang, et al. "Score-Based Generative Modeling through Stochastic Differential Equations." International Conference on Learning Representations.
>
> [7] Chiuso, Alessandro, and Gianluigi Pillonetto. "System identification: A machine learning perspective." Annual Review of Control, Robotics, and Autonomous Systems 2.1 (2019): 281-304.
>
> [8] Ronneberger, Olaf, Philipp Fischer, and Thomas Brox. "U-net: Convolutional networks for biomedical image segmentation." International Conference on Medical image computing and computer-assisted intervention. Cham: Springer international publishing, 2015.
>
> [9] Meila, Marina, and Hanyu Zhang. "Manifold learning: What, how, and why." Annual Review of Statistics and Its Application
>
> [10] Florentin Goyens, P-A Absil, Florian Feppon. "Geometric design of the tangent term in landing algorithms for orthogonality constraints", International Conference on Geometric Science of Information

---

### Official Review · Reviewer_eZg1 · 2025-11-01

**Soundness:** 3
**Presentation:** 3
**Contribution:** 3
**Rating:** 6
**Confidence:** 2

**Summary:**

The paper studies constrained optimization on data manifolds when the manifold is only available implicitly via samples. It smooths the data distribution with a Gaussian of scale σ and defines a “link function” $\ell_\sigma$ whose gradient and Hessian can be related to manifold operations. Theorem 1 shows that as $\sigma \to 0$, $x + \sigma^2\nabla log p_\sigma(x)$ uniformly approximates the closest-point projection $\pi(x)$ to the manifold, and $I + \sigma^2\nabla^2 log p_\sigma(x)$ approximates the Jacobian of that projection with error $O(\sigma|log \sigma|^3)$ on a tubular neighborhood. Corollary 2 then states that, on the manifold, $I + \sigma^2\nabla^2 log p_\sigma(x) \to P_{TM}$. By this process, the denoising score approximates manifold projection/retraction.

Building on this, the paper proposes two algorithms that only require inference-time access to a trained score network sσθ and its Jacobian–vector products: (i) Denoising Landing Flow (DLF), a continuous-time flow combining a Riemannian-like step with a “landing” term toward the manifold (eq. 8), and (ii) Denoising Riemannian Gradient Descent (DRGD), a discrete method that applies an approximate tangent step followed by an approximate retraction given by the Tweedie mean (eq. 12). The paper provides non‑asymptotic guarantees: for DLF, accumulation points are near the manifold and have small Riemannian gradient norm with bias proportional to the geometry-approximation error (Theorem 3); for DRGD, an average Riemannian gradient-norm bound with similar bias terms (Theorem 5). Theorem 20 shows that with exact manifold operators ($\sigma=0$) the flow reduces to classical exact Riemannian gradient flow. Experiments on illustrate the approach. Figures 1 and 2 show optimized trajectories tracking references more closely than the best sample from the training set.

**Strengths:**

* The link function viewsystematically connect $\nabla \log p_\sigma$ and $\nabla^2 \log p_\sigma$ to retraction and tangent projection, respectively, with explicit uniform error bounds (eq. 5). This is a useful and reusable idea.

* Theorem 1 leverages Laplace-method estimates to obtain uniform $O(\sigma|log \sigma|^3)$ control on a tubular neighborhood; the constants are traced to curvature/tubular radius and density smoothness (Appendix D). This is technically solid and extends beyond heuristic “score is normal to the manifold” statements.

* Both DLF and DRGD require s(x) and Jacobian–vector products s′(x)v (Remark 4), which makes them practical with modern autodiff.

* The analysis is structured so that (a) convergence of optimization with approximate manifold operators is established assuming an $L_{\infty}$ approximation error, and (b) Theorem 1 supplies such an error by letting $\sigma$ be small. This cleanly decomposes the problem.

* The paper clearly distinguishes constrained optimization from posterior-guided sampling and explains why small “temperatures” in the latter can drift off the true manifold—motivating the explicit feasibility machinery here (Section 1.2).

**Weaknesses:**

* The dependence on $\sigma \to 0$ is strong in practice. Theorem 1 and Corollary 2 fundamentally rely on $\sigma$ being small, and all optimization guarantees carry a bias term $\sigma|log \sigma|^2$. Training accurate scores precisely in the small‑$\sigma$ regime is hard, so the theoretically clean limit may be brittle in practice. The paper acknowledges this by proposing a landing flow, but still the guarantees rely on $\sigma$ shrinking (Theorem 3).

* Uniform $L_\infty$ approximation is a strong assumption. Convergence theorems (7) require s to approximate both $\pi_\sigma$ and its Jacobian uniformly over a neighborhood. Score models are typically trained in L2 (denoising score matching). The authors do note this (Remark 6) and defer L2‑type analyses to future work.

* Comparison to classical Riemannian GD is limited. With $\sigma = 0$ and exact operators, DLF reduces to Riemannian gradient flow with exact landing (Theorem 20), which is consistent with classical results; however, for $\sigma > 0$ the paper provides average gradient‑norm bounds with additive bias rather than full complexity guarantees comparable to known rates for feasible Riemannian methods. A more direct side‑by‑side empirical and theoretical comparison to “Riemannian GD with a known retraction” would be  useful.

* The experiments demonstrate improvement over the best training sample, but comparisons to explicit‑manifold baselines (where available) are limited to an “exact landing flow”. There are no comparisons to standard Riemannian optimizers using the true retraction on the same objective, nor ablations for the effect of $\sigma$ and Jacobian accuracy in higher‑dimensional settings.

**Questions:**

1. The theorems assume a fixed small $\sigma$ to control the geometry error. In practice, do you anneal $\sigma$ during optimization? Or do you pick a $\sigma$ that trades off stability and bias? Some concrete empirical guidance would be helpful.
2. DRGD performs “tangent step + retraction” using s and s′. If we hypothetically replace $s$ with the exact projection $\pi$ and $s'$ with $P_{T_x M}$, do your bounds recover the standard complexity guarantees for Riemannian GD (up to constants), or is there an intrinsic loss because you work in ambient coordinates with the landing penalty? It would be helpful to position Theorem 5 against known rates.
3. For real networks, $s$ may not be conservative. Apart from the barrier‑type argument in Appendix E.4, are there practical regularizers or architectural choices (e.g., gradient‑field parameterizations) that make s approximately integrable?

---

> ### Author Response · Authors · 2025-11-24
>
> Thank you for the careful and constructive review.
> We are glad that you found the link-function viewpoint, the Laplace-based analysis of $\nabla \log p_\sigma$ and $\nabla^2 \log p_\sigma$, and the practicality of using only score and Jacobian–vector products to be useful strengths of the work.
>
> Before addressing the questions raised by the reviewer, we would like to comment on the first point of the weakness section.
> We will also provide some additional experiments relating to the fourth comment (comparison with the baseline RGD) in the coming days.
>
> > The dependence on $\sigma\to0$ is strong in practice. Theorem 1 and Corollary 2 fundamentally rely on  being small, and all optimization guarantees carry a bias term $\sigma |\log \sigma|^2$ . Training accurate scores precisely in the small-$\sigma$ regime is hard, so the theoretically clean limit may be brittle in practice. The paper acknowledges this by proposing a landing flow, but still the guarantees rely on $\sigma$ shrinking (Theorem 3).
>
> We appreciate the Reviewer’s concern about the small–$\sigma$ regime and would like to emphasize that our **requirements on score accuracy are different from** the standard ones used in diffusion-model convergence analyses.
>
> Concretely, to satisfy condition (7) in our submission, the learned score `s` needs to satisfy (please see (*1) for a clarification of the notation)
>
> $$
> |\texttt{s}(\cdot, \sigma) - \nabla \log p_\sigma|_{\mathcal{L}^\infty} \leq \frac{\epsilon}{\sigma^2}
> $$
>
> and
>
> $$
> |\nabla \texttt{s}(\cdot, \sigma) - \nabla^2 \log p_\sigma|_{\mathcal{L}^\infty} \leq \frac{\epsilon}{\sigma^2},
> $$
>
> i.e. the error for the **score and its Jacobian** is of order $o(\sigma^{-2})$ as $\sigma \to 0$.
> By contrast, typical convergence guarantees for diffusion models require a score learning accuracy of the form [1, Assumption 3]
>
> $$
> |\texttt{s}(\cdot, \sigma) - \nabla \log p_\sigma|_{\mathcal{L}^2} \leq \epsilon
> $$
>
> in order to recover the underlying data distribution, i.e. the error for the **score** is of order $o(1)$ as $\sigma \to 0$.
> Ignoring for the moment the difference between $\mathcal{L}^\infty$ and $\mathcal{L}^2$ norms, the key point is that our scaling in $\sigma$ is much more permissive when it comes to the requirement on the score: we only need the score to be accurate enough to recover certain **geometric operations on (the support of) $\mu_{\mathrm{data}}$**, whereas diffusion models aim to recover the **full density**. Clearly the former is a prerequisite of the latter, and therefore this translates into a less stringent condition on score accuracy.
>
> We agree that training accurate scores becomes harder as $\sigma \to 0$.
> However, our experiments suggest that this weaker requirement for the accuracy of the score is realistically achievable in practice.
> This is also why we believe that the current $\mathcal{L}^\infty$-type assumptions could be further relaxed to suitable $\mathcal{L}^2$-type conditions in future work.
>
> (*1) We apologize for the confusing notation. The function $s$ in eq.(7) is **not** the score function. Instead, the score function is denoted by `\texttt{s}` above eq.(7) (i.e. `\texttt{s}`). A simple calculation shows that eq.(7) is equivalent to the above requirement on `\texttt{s}`. We adjust the notation and clarify this in our revision.
>
> [1] Chen, Sitan, et al. "Sampling is as easy as learning the score: theory for diffusion models with minimal data assumptions." International Conference on Learning Representations. 2023.

---

> ### Author Response · Authors · 2025-11-24
>
> In the following, we proceed to address each of the comments.
>
> > Question 1: The theorems assume a fixed small $\sigma$ to control the geometry error. In practice, do you anneal $\sigma$ during optimization? Or do you pick a $\sigma$ that trades off stability and bias? Some concrete empirical guidance would be helpful.
>
> In all the numerical experiments we pick a constant $\sigma > 0$ and have treated $\sigma$ as a hyperparameter dependent on the concrete score architecture.
> In general, as the Reviewer notes, there is an inherent trade-off in selecting $\sigma$: smaller $\sigma$ yields a better approximation of the ideal manifold operations (as in Theorem 1), but makes the score-learning problem more challenging.
>
> Using an annealing schedule for $\sigma$ is indeed a natural idea—and is precisely the strategy employed in diffusion models. We expect such an annealing scheme to bring further practical benefits in our setting as well, but a careful investigation of this requires a more extensive empirical study, which we view as an interesting direction for future work.
>
> > If we hypothetically replace $s$ with the exact projection $\pi$ and $s'$ with $P_{T_x M}$, do your bounds recover the standard complexity guarantees for Riemannian GD (up to constants), or is there an intrinsic loss because you work in ambient coordinates with the landing penalty? It would be helpful to position Theorem 5 against known rates.
>
> Yes, we do recover the standard asymptotic bound up to constants: a $O(1/\sqrt{N})$ bound on the minimum norm of the first $N$ gradients.
> Note that the average-squared-norm-of-gradients (and hence minimum-norm-of-gradients) bound is due to the general non-convexity of the function $f$.
> The difference to the classical result is in the constants due to the manifold curvature and charts (which are only dependent on the manifold).
>
> To see this, let us recall the standard complexity guarantee for Riemannian gradient descent under a general smooth non-convex objective $f$:
> In [1, Corollary 4.9] it is stated that (taking $R_x(v) = \pi(x+v)$ as the retraction defined on some ball $B_r(x) \cap T_x \mathcal{M}$, where for us $r > 0$ can be picked uniformly over $x \in \mathcal{M}$) if $f \circ \pi$ satisfies the pullback $L$-Lipschitz condition
>
> $$
> f(\pi(x+v)) - f(x) \leq v^\top \nabla f(x) + \frac{L}{2}\Vert v\Vert^2 \text{ for all } v \in B_r(x) \cap T_x \mathcal{M},.
> $$
>
> Then for step-size $\gamma_k = \frac{1}{L}$ the RGD generates iterates $(x_k, v_k = -\gamma_k \pi'(x_k) \nabla f(x_k))$ that satisfy
>
> $$
> \min_{k=0,\ldots,N-1} \Vert \operatorname{grad} f(x_k) \Vert \leq \frac{\sqrt{2L(f(x_0) - f_*)}}{\sqrt{N}},,
> $$
>
> provided that $v_k \in B_r(x) \cap T_x \mathcal{M}$ for all $k=0,\ldots,N-1$.
>
> Now we set the following in Theorem 5 in our paper: $s = \pi$, $\sigma = 0$.
> The pullback $L_0$-Lipschitz condition is satisfied under our assumption $L = \operatorname{Lip}(\nabla f)$ for a different constant $L_0 = \operatorname{Lip}(\nabla (f \circ \pi))$, which can be estimated in terms of $L$, $C$, $\tau$ and $M$ (see the proof of Theorem 5).
> The rate obtained in Theorem 5 reads then
>
> $$
> \min_{k=0,\ldots,N} \Vert \operatorname{grad} f(x_k) \Vert^2
> \leq \frac{4D}{N \gamma_{\min}(1-L_0 \gamma_{\max}/2)},,
> $$
>
> where $\gamma_{\min}(1-L_0 \gamma_{\max}/2)$ can be lower-bounded by a constant constructed from $\tau$ and $L_0$.
> Moreover, $2D$ is simply an estimation of $f(x_0) - f_*$ via the triangle inequality (see the proof of Theorem 5).
> Thus in this case both theorems guarantee an $O(1/\sqrt{N})$ best-norm-of-gradient convergence rate with similar constants.
> We will provide a comparison in the updated appendix of the paper.
>
> > Question 3: For real networks, $s$ may not be conservative. Apart from the barrier-type argument in Appendix E.4, are there practical regularizers or architectural choices (e.g., gradient-field parameterizations) that make s approximately integrable?
>
> We first emphasize that our theoretical analysis does **not** require $s$ to be a gradient field, and in our experiments we do not constrain the learned score to have this form. Prior works have proposed architectures that explicitly enforce the score to be a conservative vector field (e.g., [2], [3]). However, to the best of our knowledge, empirical evidence on whether such parametrizations systematically improve the generative performance of diffusion models remains inconclusive. For this reason, we have not explored enforcing conservativity of the score in our setting, nor how such a constraint would affect its quality as an approximate projection.
>
> [1] Boumal, Nicolas. An introduction to optimization on smooth manifolds. Cambridge University Press, 2023.
>
> [2] C. Chao et. al. On Investigating the Conservative Property of Score-Based Generative Models
>
> [3] C. Horvat, J. P. Pfister. On gauge freedom, conservativity and intrinsic dimensionality estimation in diffusion models

---

> > ### Comment · Reviewer_eZg1 · 2025-11-26
> >
> > I thank the authors for the response. I will maintain my score.

---

### Official Review · Reviewer_NnVy · 2025-11-04

**Soundness:** 4
**Presentation:** 4
**Contribution:** 4
**Rating:** 10
**Confidence:** 4

**Summary:**

The goal of the paper is creating a novel framework that ties theoretical Riemannian optimization tools (retraction, computing the Riemannian gradient, projection onto a tangent space, etc) to practical methods that rely solely on the data distribution (e.g., diffusion models). The proposed work is impactful and impressive in numerous ways. I highly recommend this paper to be accepted.

**Strengths:**

Strong points: the paper is well-written, with clearly made arguments, well-structured problem, meaningful theoretical as well as experimental results. The paper provides a comprehensive literature review and acknowledges its own limitations.

The paper supports results with rigorous mathematical proofs, as well as intuition-based explanations, which I appreciated very much. This work has many, many strengths. To name a few: this work provides a rigorous theoretical analysis of how Riemannian optimization tools can in fact approximate practical operations such as projection (for denoising purposes), and how this can be done by learning from data. This works provides a massive step forward in this literature by connecting two fields that have been struggling to converge -- theory of high-dimensional geometry and diffusion models.

The significance of this work is huge. It makes a rigorous connection between theoretical and practical fields (Riemannian optimization and diffusion models), which is very high-impact and rare to come across. Creating such impactful and meaningful work requires a vast and deep knowledge of the field, and one can dream to be at that level as a researcher!

Authors make several innovations. Impressively, authors prove that the quantities in equation 4 converge to manifold operations as $\sigma \rightarrow 0$. For example, the proposed link function provides a way to approximately project points onto the unknown manifold wihtout explicitly knowing its equation, using the score function learned from data. Authors also clearly demonstrate how to use these operations for optimization by proposing two concrete algorithms that tell us how exactly their work can be implemented. Lastly, one non-obvious strength is in experiments, where the authors show that even though theoretical results are strong when $\sigma \rightarrow 0$, experiments work even when $\sigma$ is nonzero.

**Weaknesses:**

The authors heavily rely on the claim that classical algorithms require knowledge of the manifold to be able to use Riemannian optimization tools, however, this claim is not necessarily true. Plenty of works, especially [1], develop provable, feasible, accurate, learnable, and inference-time efficient algorithms that create machinery to use Riemannian optimization tools without knowledge about the manifold, while having access to noisy samples only. In fact, the noise level can also be relatively high. Authors are also missing another key work [2] which provides a provable and accurate denoising method without knowing the underlying manifold. While I find the results of the paper impressive and useful, if one were to make the paper even stronger, I would recommend re-positioning / re-phrasing the argument that "classical Riemannian optimization cannot be applied directly, since they rely on explicit knowledge of the manifold."

Lastly, this work heavily relies on the fact that a diffusion model will learn the score function well. If this is not the case, the proposed framework will not be useful.

[1] Wang, Shiyu, Mariam Avagyan, Yihan Shen, Arnaud Lamy, Tingran Wang, Szabolcs Márka, Zsuzsa Márka, and John Wright. "Fast, Accurate Manifold Denoising by Tunneling Riemannian Optimization." Proceedings of the 42 nd International Conference on Machine
Learning (ICML), Vancouver, Canada. PMLR 267, 2025.

[2] Yao, Zhigang, Jiaji Su, Bingjie Li, and Shing-Tung Yau. "Manifold fitting." arXiv preprint arXiv:2304.07680 (2023).

**Questions:**

1. It makes sense that authors need a $C^2$-submanifold. Do you have any intuition on what this means in terms of limitations, both practical and theoretical?

2. How does the proposed approach compare to well-known baselines, such as VAEs and GANs for modeling manifolds? This would help position the work in the literature.

3. Could the proposed approach converge slowly (in terms of convergence rate) in certain scenarios?

4. Is the approach easily extendable to incorporating methods like heavyball (momentum) into the optimization process? If so, what are some potential challenges that you foresee?

5. Are there certain manifolds where the proposed framework could struggle?

---

> ### Author Response · Authors · 2025-11-27
>
> We thank the reviewer for the very positive and encouraging feedback. We are glad that the contributions of the work were clearly conveyed.
>
> We also thank the reviewer for pointing out these interesting works. We will incorporate them in the revised version to better situate our contribution within the broader manifold-learning literature and will include a comparison with non-parametric methods such as [1] and the references therein. We emphasize that the algorithms in [1] do not perform Riemannian optimization over a smooth manifold as in our setting; instead, their optimization is effectively carried out over a discrete graph, even though they still obtain novel global convergence guarantees.
>
> The extension of our work to settings with noisy manifold samples would be very interesting. For instance, suppose the noise is uniformly distributed in a ball of radius $r$ smaller than the reach of the manifold. In this case, our preliminary research shows that the corresponding link function is given by the distance to the tubular neighborhood of $M$ with radius $r$. It would be an intriguing direction for future work to investigate the implications of this characterization for optimization.
>
> In the following, we address the questions raised by the reviewer. All the following discussion will be incorporated in our revision.
>
>
> **Q1.** Yes, in the context of the current work, we need the $C^2$ regularity for the tangent projection matrix to be well-defined. Please note that the $C^2$ regularity is also implicitly required in [1]: Theorem 7.1 therein require the extrinsic geodesic curvature to be finite, but the existence of the extrinsic geodesic curvature already requires the underlying manifold to be $C^2$.
>
> The practical limitation of this assumption heavily depends on the application in mind. When the true underlying data manifold is highly irregular, e.g. only a topological manifold, optimization over such a set is typically difficult. Theoretically, when the regularity of the underlying data manifold is below $C^2$, our approach needs to be refined to accommodate for the fact that the closest point projection need not to be differentiable (even at points where it is unique). Also, another issue of lower regularity, e.g. $C^1$ manifold, is that the tubular neighborhood theorem breaks and there may not be a tubular neighborhood in a geometric sense. Hence the applicable region of our work needs to be clarified in this setup.
>
>
> **Q2.** This is an excellent question. Once the inverse coordinate map of a chart is available (i.e. the chart representation of the manifold is known), one can equivalently formulate the pull-back gradient flow in the chart domain (the “latent space” in machine learning terminology). If the decoder of a VAE or the generator of a GAN accurately learns this inverse coordinate map, then an approximate pull-back gradient flow can in principle be implemented locally using the learned decoder/generator. However, there are several nontrivial challenges associated with this approach that should not be overlooked. For a more detailed discussion, we refer the reviewer to our response titled `A discussion on the pull-back gradient flow` to Reviewer xQd1.
>
>
> **Q3.** As suggested by our theory, the efficiency of the proposed method depends on several factors, including the accuracy of the learned score and the regularity of the underlying manifold. However, we are not entirely sure what specific aspect the reviewer is referring to in this question. We would greatly appreciate it if the reviewer could elaborate, so that we can provide a more precise and helpful response.
>
>
> **Q4.** This is an interesting idea. A key step in momentum-based methods is to apply a momentum vector, computed at a point different from the current iterate, to update the current iterate. In practice, by direct analogy with the Euclidean momentum method, one can use tangent projections to approximate parallel transport, which our theory provides. However, for establishing rigorous convergence guarantees, one typically needs access to the exponential map [A, B, C]. Unfortunately, we are not aware of any efficient strategy to reconstruct this operation directly from the projection operator onto the manifold.
>
>
> **Q5.** Our results rely on a local analysis of the gradient and Hessian of the link function. For manifolds with very small reach, this forces us to train the score network with a correspondingly small $\sigma$ in order for our theory to remain valid. However, learning scores at such small noise levels is typically known to be challenging in the diffusion-model literature. We therefore expect this regime to constitute a particularly hard instance for our method.

---

> > ### Author Response · Authors · 2025-11-27
> >
> > [A] Han, Andi, et al. "Riemannian accelerated gradient methods via extrapolation." International Conference on Artificial Intelligence and Statistics. PMLR, 2023.
> >
> > [B] Feng, Shuailing, et al. "A Riemannian Accelerated Proximal Gradient Method." arXiv preprint arXiv:2509.21897 (2025).
> >
> > [C] Jin, Jikai, and Suvrit Sra. "Understanding riemannian acceleration via a proximal extragradient framework." Conference on Learning Theory. PMLR, 2022.

---

### Author Response · Authors · 2025-12-03
**To AC: Major concern in the reviews and how we address it in our revision.**

We thank the reviewers for the careful reading of our paper and for the constructive feedback, which has helped us to substantially improve the manuscript.
For the convenience of the area chair, we shortly summarize the major concern raised by the reviewers and describe how the concerns are addressed in our revision.

A common concern raised by the reviewers is how our work compares to the existing literature on manifold learning as well as some alternative options for solving optimization over data manifold.
In the revised version, we have implemented the following changes in direct response to the comments:

1. Introduction. We have rewritten and streamlined the introduction to more clearly explain the problem setting, our assumptions, and the scope of the proposed framework, so that the contribution and positioning of the work are easier to follow.

2. Related work. We have expanded the related work section to better situate our contribution within the broader literature on parametric manifold learning, now explicitly discussing approaches based on GANs, VAEs, and M-flows, and clarifying how our setting and goals differ from these lines of work.
We also explicitly compare with the work *"Fast, Accurate Manifold Denoising by Tunneling Riemannian Optimization"* as requested by reviewer NnVy.

3. Connections to prior work. We have added an appendix section (Appendix A) that elaborates on the connection and comparison of our framework to specific prior work highlighted by the reviewers, including
(i) *“What regularized auto-encoders learn from the data-generating distribution”* (Alain \& Bengio),
(ii) work on optimization in latent space and pull-back Riemannian gradient flow (requested by reviewer xQd1), and
(iii) classical iteration-complexity results for Riemannian gradient descent (requested by reviewer eZg1).
This new section makes explicit how our link-function viewpoint and feasibility/optimality guarantees relate to these existing results.

4. Typographical and notational improvements.
We have corrected various typographical issues and refined some notation in line with the reviewers’ suggestions, aiming to improve readability and reduce possible sources of confusion.

We hope that these revisions address the main concerns raised in the reviews and help clarify the contribution, positioning, and technical content of the paper. We are grateful for the insightful comments and suggestions, which have significantly strengthened the manuscript.

---

### Meta-Review · Area_Chair_9Nok · 2026-01-04

**Summary:**

This paper presents a novel framework that ties theoretical Riemannian optimization tools (retraction, computing the Riemannian gradient, projection onto a tangent space, etc) to practical methods that rely solely on the data distribution (e.g., diffusion models).

3/4 reviewers gave highly positive recommendations for acceptance, but 1/4 reviewer pointed out that this paper did not consider an important line of related works in manifold learning, where the data manifold is characterized via the pullback metric of the decoder of a (deep) generative model. I agree with Reviewer xQd1 that manifold learning should be considered to broaden the proposed framework's capabilities. The authors did a good job responding to all concerns in the rebuttal, and I believe this may somehow relieve the concern of Reviewer xQd1. Despite these concerns, I think this paper is generally good enough to meet the ICLR standard.

**Reviewer Concerns:**

Most concerns are well addressed by the authors, and they provide detailed responses to these rebuttals.

**Reviewer Scores:**

Reviewer xQd1 might increase his/her score if he/she had been able to participate fully in the discussion.

---

### Decision · Program_Chairs · 2026-01-26

Accept (Poster)